# On the Ability of Transformers to Verify Plans

**Yash Sarrof**[⅃ 1] **Yupei Du**[↗ 1] **Katharina Stein**[↗ 1] **Alexander Koller**[1] **Sylvie Thiébaux**[2 3] **Michael Hahn**[1]

## Abstract

Transformers have shown inconsistent success in AI planning tasks, and theoretical understanding of when generalization should be expected has been limited. We take important steps towards addressing this gap by analyzing the ability of decoder-only models to verify whether a given plan correctly solves a given planning instance. To analyse the general setting where the number of objects – and thus the effective input alphabet – grows at test time, we introduce **C\*-RASP**, an extension of **C-RASP** designed to establish length generalization guarantees for transformers under the simultaneous growth in sequence length and vocabulary size. Our results identify a large class of classical planning domains for which transformers can provably learn to verify long plans, and structural properties that significantly affects the learnability of length generalizable solutions. Empirical experiments corroborate our theory.

## 1. Introduction

Transformer based Large Language Models (LLMs) have demonstrated remarkable capabilities, but their success in the field of *AI planning* has been mixed. Planning involves computing a sequence of actions to transform an initial state into a desired goal state. To circumvent combinatorial explosion, planning state spaces are represented symbolically, typically using variants of the STRIPS formalism as standardized in the PDDL planning definition language (Fikes & Nilsson, 1971; Haslum et al., 2019). LLMs have made considerable strides in solving planning problems through apt prompting, fine tuning, and programmatic plan or heuristic generation (Yao et al., 2023; Stein et al., 2025; Pallagani et al., 2023; Silver et al., 2024; Chen et al., 2025a; Aghzal et al., 2025), but there is also evidence pointing to their

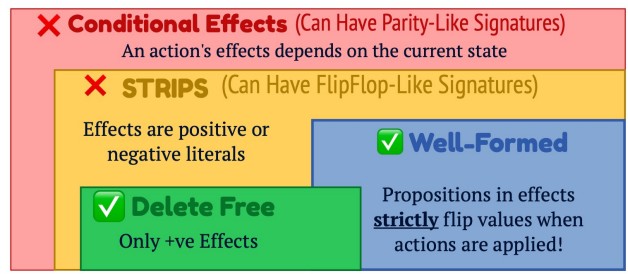

*Figure 1.* We study plan verification in the *fixed-universe* setting, where the object set is fixed and the plan length grows, and the *variable-universe* setting, where longer test plans may contain more objects than seen during training. We study four classes: delete-free domains that allow only positive effects; well-formed domains where propositions listed in the effects strictly change value when the action is applied; STRIPS domains without any restriction on propositions appearing in effects or preconditions; and domains with conditional effects which permit actions to have differing effects conditioned on the current state. We prove that transformers can learn to generalizably verify plans for delete-free and well-formed domains. We also show that there exist STRIPS and conditional-effect domains for which transformers are not expected to learn to generalizably verify plans. We use **C-RASP**, a length generalization framework for transformers (Yang & Chiang, 2024; Huang et al., 2025b) for proving these results for the fixed universe setting. We introduce **C\*-RASP**, an extension of **C-RASP** as a tool to give length generalization guarantees for Transformers in cases where the test time vocabulary grows, and use it to prove our results in the variable universe setting.

*inability* to *reliably* solve planning problems, e.g. under the slogan "LLMs still can't plan" (Valmeekam et al., 2022; 2024). On many benchmarks, transformers tend to lose track of the world state, propose inapplicable actions, or stop before reaching the goal – especially as plans grow longer (Huang et al., 2025a; Fritzsche et al., 2026). The analysis of LLM planning failures in the literature is almost exclusively empirical and a clear theoretical analysis is lacking.

This paper contributes to the *theoretical understanding of why transformers succeed on certain planning tasks and fail on others*. We focus specifically on *plan verification*, which decides for a given planning instance and a given plan whether the plan correctly solves the instance. Furthermore, we focus on the question of *length generalization*: can a transformer learn to verify long plans if it is trained only on short plans? If not, then they cannot fully learn

---
[⅃]Led theoretical work  [↗]Led empirical work
[1]Saarland University [2]Australian National University [3]University of Toulouse / LAAS-CNRS. Correspondence to: Yash Sarrof <ysarrof@lst.uni-saarland.de>.

*Proceedings of the 43rd International Conference on Machine Learning*, Seoul, South Korea. PMLR 306, 2026. Copyright 2026 by the author(s).

to verify plans from any finite amount of training data. A key technical challenge in understanding how transformers generalize on plan verification is that existing techniques for length generalization are limited to the case where the token vocabulary at training and inference time is the same. In this case, a recent line of work has established a theoretically principled framework for predicting length generalization of transformers based on the expressiveness of the **C-RASP** language (Yang & Chiang, 2024; Huang et al., 2025b; Yang et al., 2025; Jobanputra et al., 2025; Jiang et al., 2026; Chen et al., 2025b) and its variant **C-RASP**[Pos][1] (Huang et al., 2025b; Jobanputra et al., 2025). It is known that transformers can length-generalize on problems that can be expressed in **C-RASP**[Pos], and that they empirically tend not to length-generalize otherwise. We use the **C-RASP** framework to analyze plan verification for the case where the planning instances at training and test time contain the same objects (Fixed Universe); but of course an even more interesting question is whether transformer can generalize to instances with a *larger* set of objects (Variable Universe). This case is out of reach of a **C-RASP**[Pos]-based analysis.

We therefore introduce **C\*-RASP**, which extends **C-RASP** with the ability to handle increasing alphabets during test time, and provides length generalization guarantees under a formal learning model inspired by Huang et al. (2025b). This is a nontrivial technical result with ramifications beyond plan verification. Using our frameworks, we establish the following results (Overview in Figure 1). Plan verification is in **C\*-RASP** if it is *delete-free* (i.e. actions only have positive effects) or *well-formed* (i.e. each of their effects necessarily changes the state). Full STRIPS plan verification is not in **C\*-RASP**, nor is plan verification with conditional effects. This implies that certain structural properties – such as well-formedness which holds of a large class of classical planning domains – can enable the learning of length generalizable solutions, whereas the latter is intractable for unconstrained plan verification with transformers. We confirm our theoretical findings experimentally on a number of planning domains.

## 2. Background & Task Definition

### 2.1. Planning

Planning involves finding a path from an initial to a goal state in a huge transition system where we transition between possible states of the world with actions. This problem is compactly represented in terms of a *domain* (predicates representing states and action schemas describing state transitions) and an *instance* that describes the particular set of objects present in the world, the initial state, and goal.

A planning *domain* is a pair $\mathcal{D} = \langle \mathcal{P}, \mathcal{A} \rangle$ where

---

---

> **Predicates:** $\mathrm{room}(r), \mathrm{ball}(b), \mathrm{gripper}(g), \mathrm{free}(g), \mathrm{heavy}(b),$ charged, $\mathrm{atRobby}(r), \mathrm{at}(b, r), \mathrm{carry}(b, g)$
>
> **Action** $\mathrm{move}(r_1, r_2)$:
>   **Pre:** $\{\mathrm{room}(r_1), \mathrm{room}(r_2), \mathrm{atRobby}(r_1)\}$
>   **Eff:** $\{\mathrm{charged}, \mathrm{atRobby}(r_2), \neg\mathrm{atRobby}(r_1)\}$
>
> **Action** $\mathrm{pick}(b, r, g)$:
>   **Pre:** $\{\mathrm{ball}(b), \mathrm{room}(r), \mathrm{gripper}(g), \mathrm{atRobby}(r), \mathrm{at}(b, r),$
>     $\mathrm{free}(g), \mathrm{charged}\}$
>   **Eff:** $\{\neg\mathrm{heavy}(b)\} \triangleright \{\mathrm{carry}(b, g), \neg\mathrm{free}(g), \neg\mathrm{at}(b, r)\}$
>     $\{\mathrm{heavy}(b)\} \triangleright \{\mathrm{carry}(b, g), \neg\mathrm{free}(g), \neg\mathrm{at}(b, r), \neg\mathrm{charged}\}$
>
> **Action** $\mathrm{drop}(b, r, g)$:
>   **Pre:** $\{\mathrm{ball}(b), \mathrm{room}(r), \mathrm{gripper}(g), \mathrm{atRobby}(r), \mathrm{carry}(b, g)\}$
>   **Eff:** $\{\mathrm{at}(b, r), \mathrm{free}(g), \neg\mathrm{carry}(b, g), \neg\mathrm{charged}\}$

*Figure 2.* Heavy Grippers domain. We omit the empty condition sets for the effects in *move* and *drop* which are unconditional.

1. $\mathcal{P}$ is a finite set of *predicates*. A predicate $p \in \mathcal{P}$ has the form $p(x_1, \ldots, x_{n_p})$ where $x_i$ are arguments $p$ takes.

2. $\mathcal{A}$ is a finite set of *action schemas*. Each $a \in \mathcal{A}$ is a triplet $\langle \mathrm{args}(a), \mathrm{pre}(a), \mathrm{eff}(a) \rangle$ where $\mathrm{args}(a)$ are arguments, $\mathrm{eff}(a) = [\mathrm{C}_i(a) \triangleright \mathrm{eff}_i(a)]_{i=1}^{k(a)}$ are *conditional effects*, and $\mathrm{pre}(a)$, $\mathrm{eff}_i(a)$, and $\mathrm{C}_i(a)$ are sets of literals over $\mathcal{P}$ with arguments taken from $\mathrm{args}(a)$. They represent the *preconditions* of $a$, and its *effects* under various mutually exclusive and exhaustive *conditions*, respectively.

Figure 2 presents an example, a variant of a well-known domain which we call Heavy Grippers, where a robot must relocate balls within a set of rooms. The robot can move between rooms, pick, drop and carry balls using its grippers. Some balls are heavy. Dropping balls and picking heavy balls discharges the battery, whereas moving recharges it.

A planning *instance* $\Pi = \langle \mathcal{D}, O, I, G \rangle$, has a domain $\mathcal{D}$, set of *objects* $O$, *initial state* $I$, and *goal* $G$.

A *proposition* $p(o_1, \ldots, o_{\mathrm{arity}(p)})$ is a ground instance of a predicate $p \in \mathcal{P}$ obtained by substituting the arguments of $p$ with objects from $O$; we write $P$ for the sets of all propositions. A *state* is a set of propositions under the closed world assumption, i.e. this set contains exactly those propositions that hold in the state. The goal $G$ is represented by a set of ground *literals*. In the following, we partition sets of literals $L$ into: $L^+$ containing all propositions appearing positively in $L$, and $L^-$ containing propositions whose negation appears in $L$. $L$ is satisfied in state $S$, written $S \models L$, iff $L^+ \subseteq S$ and $L^- \cap S = \emptyset$. $S$ is a goal state iff $S \models G$.

Actions represent one-step transitions in the state space of the planning instance. An *action* $a(o_1, \ldots, o_{\mathrm{arity}(a)})$ is a ground instance of an action schema $a \in \mathcal{A}$, where arguments of $a$ are substituted with objects from $O$ in its preconditions, conditions and effects; we write $A$ for the set of actions. An action $a \in A$ is *applicable* in a state $S$ iff its preconditions are satisfied (i.e. $S \models \mathrm{pre}(a)$), in

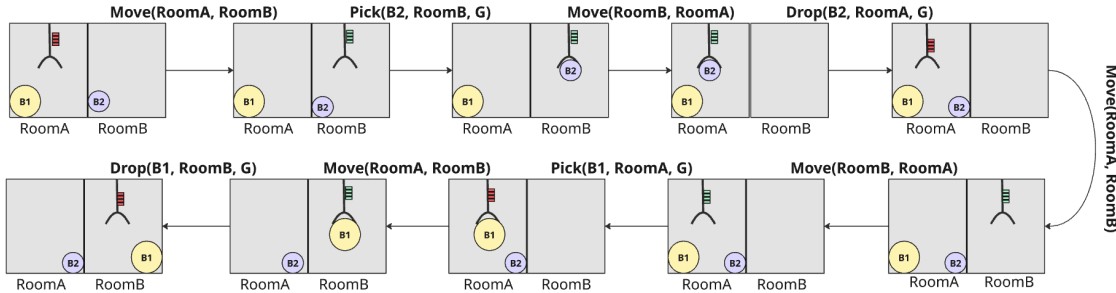

*Figure 3.* Valid plan for a Heavy Grippers instance with 1 gripper, 2 balls and 2 rooms. The goal $G = \{\text{at}(B1, \text{RoomB}), \text{at}(B2, \text{RoomA})\}$ is to swap the balls in our rooms. Ball B1 is heavy and requires charge, making moving the only legal action in $I$. The robot then picks B2 from RoomB, moves, drops it in RoomA. It needs charge so it moves back and forth before picking B1, moving and dropping it in RoomB

which case the *successor* state is obtained by removing (resp. adding) the propositions made false (resp. true) by $a$: i.e. $\text{succ}(S, a) = (S \setminus \text{eff}_i(a)^-) \cup \text{eff}_i(a)^+$ where $S \models \text{C}_i(a)$.

A *plan* is a sequence of actions $\pi = [a_1, \ldots, a_n]$, and is *valid* for the planning instance $\Pi$ iff all actions are applicable and it reaches the goal. That is, $\pi$ induces a sequence of states $s_1, \ldots, s_{n+1}$ such that $s_1 = I$, $s_{i+1} = \text{succ}(s_i, a_i)$ and $s_{n+1} \models G$. Figure 3 shows a valid plan for a small instance of the Heavy Grippers domain, where the robot must swap the balls in the rooms. In the following, we say that a state $S$ is *reachable* iff there exists a valid plan for the planning instance $\langle \mathcal{D}, O, I, S \rangle$.

### 2.2. Planning Subclasses & Our Generalization Setting

Our results distinguish the following subclasses of planning instances, listed from the least restrictive to the most:

**Conditional Effects:** no restrictions are applied.

**STRIPS:** no action has conditional effects. Thus, for all $a \in A$, $k(a) = 1$ and $\text{C}_1(a) = \emptyset$. Hence $\text{eff}(a)$ simplifies to a single literal set and we write STRIPS actions in this simplified form in the rest of the paper (e.g. see move and drop in Figure 2).

**Well-Formed:** a STRIPS instance where each proposition listed in the effects of an action strictly changes value whenever the action is applied. Formally, for all reachable states $S$, all actions $a \in A$, and all $l \in \text{eff}(a)$ we have that $S \models \text{pre}(a) \implies S \models \neg l$.

**Delete-Free:** a STRIPS instance where actions *only* have positive effects, i.e. for all $a \in A$, $\text{eff}(a)^- = \emptyset$.

We say that a planning *domain* falls into one of the above classes if all instances of interest for this domain do.[2]

STRIPS is the standard formalism to describe planning problems. Conditional effects are a useful feature when modeling real-world problems, and are strictly more expressive than STRIPS (Nebel, 2000). Many classical planning benchmarks are well-formed according to our definition. The Delete-Free class captures the most well-known relaxation of STRIPS, and is commonly used when computing cost-to-goal heuristics estimates to guide the search of planners (Helmert & Domshlak, 2009).

Our primary focus is to study a transformer's capability to **length generalize**: training on valid plans of a limited length and testing on longer plans. We study two generalization settings, providing both theoretical and experimental results for each. The domain $\mathcal{D} = \langle \mathcal{P}, \mathcal{A} \rangle$ is fixed throughout, and the task is to decide if a given plan $\pi$ is valid.

**1. Fixed Universe:** The set of objects $O$ is fixed, but $I$ and $G$ can vary per input sample. The model must learn to validate longer plans for unseen pairs of $(I, G)$.

**2. Variable Universe:** The set of objects $O, I, G$ can all vary across samples. Thus at test time, longer plans might also have many more objects than ever seen during training.

Note that both the fixed and variable universe settings are widely used in the planning literature, see e.g. (Arfaee et al., 2011; Ferber et al., 2022; Rossetti et al., 2024; Toyer et al., 2020; Ståhlberg et al., 2022; Huang et al., 2025a).

### 2.3. C-RASP

Understanding and predicting transformers' generalization behavior has been of great interest (e.g. Zhou et al., 2024; Golowich et al., 2025; Izzo et al., 2026). Recently, expressiveness in the **C-RASP** (Counting RASP) language (Yang & Chiang, 2024) and its variant **C-RASP**[Pos] (Huang et al., 2025b; Jobanputra et al., 2025) has been used to establish

---

[2]Planning domain formulations normally assume a set of implicit constraints that all states must satisfy, and which hold in the initial state and are preserved by actions; the instances of interest are those for which $I$ and $G$ satisfy these constraints. This is only

relevant to Well-Formedness, since the other three properties only depend on the domain.

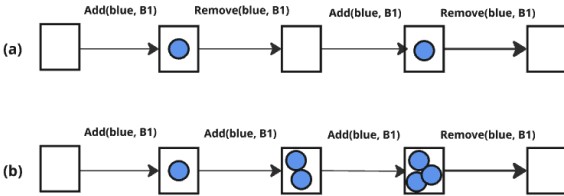

*Figure 4.* Example from the Colors domain (Fig. 9), where actions add balls of a given color to a bag or remove all balls of a given color from a bag. (a) Well-Formed: balls of the same color cannot be added if already present, and can't be removed if not in the bag. Such alternation allows the truth of "Is a color in the bag?" to be determined by simply counting add versus remove actions from the initial state. (b) STRIPS: In general STRIPS, determining the truth value requires identifying the last action that actually changed the state, making it equivalent to solving the FlipFlop language.

a theoretically principled and empirically predictive framework for predicting length generalization of transformers (Huang et al., 2025b; Yang et al., 2025; Jobanputra et al., 2025; Jiang et al., 2026; Chen et al., 2025b). If a task can be defined by a program in **C-RASP**[Pos], then decoder-only transformers with absolute positional encodings (APE) provably generalize to longer inputs under a specific formal model of training (Theorem 7 in Huang et al. (2025b)). Huang et al. (2025b) show that this result unifies a range of prior empirical observations. Tasks exhibiting length generalization can be programmed in **C-RASP**[Pos], whereas tasks provably outside this class empirically fail to generalize. For instance, Huang et al. (2025b) proved that the Flipflop language $L = \Sigma^* b e^*$ (over $\Sigma = \{a, b, e\}$) and the PARITY language $L = b^*(ab^*ab^*)^*$ cannot be expressed in **C-RASP**[Pos]. Languages reducible to either Flipflop or PARITY, are therefore predicted to and have been empirically observed to not length generalize (Liu et al., 2023; Hahn & Rofin, 2024; Huang et al., 2025b; Jobanputra et al., 2025). Thus, we obtain positive generalization results via membership in **C-RASP**[Pos], and negative results via reduction to PARITY or Flipflop.

## 3. Theoretical Results

### 3.1. Generalization over a Fixed Universe

**Theorem 3.1** (Fixed Universe). *Let $\mathcal{D}$ be a planning domain and $O$ a fixed set of objects. Consider the language $L_{\mathcal{D},O}$ consisting of all sequences $\langle I, \pi, G \rangle$ where $\pi$ is a valid plan for the planning instance $\langle \mathcal{D}, O, I, G \rangle$.*

*1. (**Delete-Free & Well-Formed**) If $\mathcal{D}$ is delete-free or well-formed, then $L_{\mathcal{D},O} \in$ **C-RASP**[Pos].*

*2. (**STRIPS**) There exist STRIPS domains and object sets such that $L_{\mathcal{D},O} \notin$ **C-RASP**[Pos].*

*3. (**Conditional Effects**) There exist domains with con-*

ditional effects such that $L_{\mathcal{D},O}$ is neither star-free nor in **C-RASP**[Pos].

*Proof Sketch.* (Full proof in Appendix A.1) These results imply that transformers can learn to verify delete-free and well-formed plans in the fixed universe setting with perfect length generalization. They also show that there are more general instances where, even with a fixed set of objects, transformers cannot length generalize.

Statement 1: **C-RASP** highlights how a transformer using uniform attention can count over its context window. For delete-free, a proposition is true if it was initially present or the count of actions listing it as an effect is $> 0$. For well-formed, the truth value of a proposition toggles (See Figure 4a). We can count the number of actions adding a proposition (+1 if initially present) and the ones deleting it. Thus all validity checks (action preconditions and goal) can be done through such counts, which allows us to write **C-RASP**[Pos] programs to prove our positive results.

For our negative results, we construct a planning instance $\Pi$ for a given $\mathcal{D}, O$, and show that even the language of valid plans of a fixed $\Pi$: $L(\Pi) \notin$ **C-RASP**[Pos].

Statement 2: Lin & Bercher (2022) already showed that $L(\Pi)$ when $\Pi$ is a STRIPS instance, is a strict subset of star free regular languages. We construct a STRIPS instance $\Pi$ such that $L(\Pi) = \Sigma^* b e^*$, where $\Sigma = \{a, b, e\}$, the FlipFlop language. Thus $L(\Pi) \notin$ **C-RASP**[Pos]. As shown in Figure 4, ascertaining the truth value of a proposition in STRIPS might require figuring out the last action that affected it.

Statement 3: We define an instance in the Lights Out planning domain with conditional effects. We show that $L(\Pi)$ in such a case can be reduced to PARITY, making $L(\Pi) \notin$ **C-RASP**[Pos] (Huang et al., 2025b). $\square$

### 3.2. Generalization over a Variable Universe

*Motivation for a new Framework:* Planning might require reasoning over a larger number of objects at test time than seen at train time. For transformers, this corresponds not only to generalization in sequence length, but also to generalization over an effectively larger alphabet. Since **C-RASP** assumes a fixed and finite alphabet $\Sigma$, it is not suitable to analyze such settings. To address this limitation, we expand on several key ideas from Huang et al. (2025b) and introduce **C\*-RASP**, an extension of **C-RASP** designed to establish length generalization guarantees for transformers under *simultaneous* growth in sequence length and vocabulary size. Huang et al. (2025b) framed length generalization as the asymptotic convergence of a sequence of transformers – trained on increasing lengths – to the same underlying algorithm, formalized as a Limit Transformer, that would

thus work at arbitrarily longer lengths than training (Huang et al., 2025b; Izzo et al., 2026). We adopt the same view, but consider asymptotic convergence of a sequence of transformers – trained on not just increasing lengths but also on increasing vocabulary – to the same underlying algorithm, which we refer to as a Symbolic Limit Transformer (Formal definition in Appendix C.6). Huang et al. (2025b) used **C-RASP** as a more human readable interface to prove tasks representable in a Limit Transformer. We will do the same with **C\*-RASP** and Symbolic Limit Transformers.

*Split Alphabet:* The key observation underlying our extension is that, in planning domains, not all symbols play the same role. Action names and predicate names are drawn from a fixed, finite set, while object identifiers are not. We therefore view the input alphabet (say $\Omega$) as partitioned into two parts: $\Sigma \cup \mathcal{C}$, $\Sigma$ is a fixed alphabet of domain-level symbols (actions, predicates, delimiters), and $\mathcal{C}$ is a countably increasing alphabet (isomorphic to natural numbers) used to represent object identities.

*Product Functions:* To formalize this, we parameterize transformers using *product functions* (Huang et al., 2025b). These functions express the internal computations of a transformer—such as attention scores and MLP activations—as inner products of learnable parameters mediated by the model's matrices (keys, queries, values, and feedforward layers; see Definition C.2 for the full formalization). They can either reflect pairwise interactions or products that depend on only a single token or position. Pairwise interactions are those between different tokens, arising for instance from self attention or with the unembedding matrix. An example is a token-token interaction $c_i^T W c_j$, where $W$ is the product of several parameter matrices throughout the transformer, and $c_i, c_j$ are embeddings for tokens $i, j \in \mathcal{C}$. Product functions affected by single token/position capture products formed because of the MLPs. Along the lines of Huang et al. (2025b), for a transformer to generalize to increasing lengths and alphabets, its internal 'algorithm' cannot depend on absolute position values or object identifiers, which is formalized via the following constraints:

1. *Translation Invariance*: Interactions must depend only on relative differences, not absolute indices. Crucially, we apply this to the alphabet $\mathcal{C}$ and not just positions: a token-token interaction must satisfy $f(c_i, c_j) = f(c_{i+\delta}, c_{j+\delta})$. Interactions between unrelated types (e.g., Position-Token) and those involving just one token or one position must collapse to constants.

2. *Locality*: Interactions must vanish when the relative distance exceeds a finite bandwidth $\Delta$, ensuring the algorithm relies only on local context.

Practically, invariance can be brought about during training via the *offset trick*: For APE, offsets are used to train

positional embeddings beyond the maximum train length. For example, if the maximum training length was 100, but generalization to length 200 was desired, during training, positions are randomly shifted to $[1 + o, \dots 100 + o]$ such that $0 \leq o \leq 100$. This discourages reliance on absolute positions, promoting invariance (Huang et al., 2025b). We can apply the same idea to our extended characters, and despite having access to a limited vocabulary during training, we can shift object identities with offsets to train embeddings beyond what is seen during training.

*The Learning Model:* Symbolic Limit Transformer is essentially an algorithm of a transformer that satisfies these constraints over an infinite context and alphabet. To prove that finite training can actually lead to this solution, we define a learning model along the lines of Huang et al. (2025b):

1. *Hypothesis Class ($\Theta_n$):* We restrict the search space to transformers that already satisfy translation invariance, and where any parameter is specified at fixed precision (finite number of bits used to represent any parameter). [3].

2. *Regularizer ($\mathcal{R}$):* A regularizer penalizes model complexity (depth, norms) and non-local interactions, forcing the inference procedure to select 'algorithms' that are local. Our regularizer simplifies the one used by Huang et al. (2025b).

3. *Inference Procedure:* We model learning as an idealized selection of the $T \in \Theta_n$ minimizing $\mathcal{R}(T)$ while matching the target function on all inputs of length $n/2$ (Huang et al., 2025b). Crucially, these inputs are restricted to use only a contiguous subset of the extended alphabet of size at most $n/2$. This forces the model to infer rules that are invariant to token identities. Our main result shows that this procedure converges: by selecting the simplest local algorithm on short data, the model effectively identifies the unique Symbolic Limit Transformer, guaranteeing generalization to length $n$.[4]

*Guaranteed Length Generalization:* We show that expressibility by a Symbolic Limit Transformer is the necessary and sufficient condition for length generalization in our framework. We note that this guarantee, similar to Huang et al. (2025b), relies on the use of specific activation functions that in practice are only approximately represented by real-world MLPs (Heaviside in Huang et al. (2025b), we also use 1/x). See Remark C.1 Informally our result is the following,

**Theorem 3.2** (Informal version of Theorem C.10). *A task is expressible by a Symbolic Limit Transformer iff our Inference Procedure generates a sequence of transformers $T_n$ that eventually generalize: there exists a threshold $N_0$*

---

[3] In infinite precision setups one cannot hope to identify algorithms implemented by transformers from finite data, notwithstanding the choice of the regularizer (Huang et al., 2025b)

[4] In doing so, we improve on the treatment of the Unique Copy task compared to Huang et al. (2025b). See Remark C.6.

*such that for all $m > N_0$, the model $T_m$ (selected on length $m/2$) correctly computes $f$ on all inputs up to length $m$.*

*Proof Sketch (Detailed Proof in Appendix C.10)* Our inference procedure generates an infinite sequence of distinct transformers $T_1, T_2, \ldots$, but translation invariance and locality ensure that they traverse only a *finite set of underlying algorithms*. The regularizer bounds the structural complexity (depth, precision), while the invariance constraints ensure that the number of product functions is finite. Thus, as $n \to \infty$, the procedure sequentially rules out algorithms that would not have generalized. Since the set of candidate algorithms is finite, there must be a $N_0$ after which only the correct one remains. □

***C\*-RASP***: Huang et al. (2025b) used **C-RASP** as a human-readable interface to establish expressibility results for Limit Transformers. We will do the same for Symbolic Limit Transformers and **C\*-RASP**. We define the following logical predicate that acts over our extended alphabet $\mathcal{C}$, which is used to refer to object IDs. Since generally these objects IDs are numbered, we assume that $\mathcal{C} \cong \mathbb{N}$.

**Definition 3.3** (Match Predicate). A **Match Predicate** $\chi(i, j)$ is a conjunction of equality checks between tokens in the neighborhoods of $i$ and $j$: $\chi(i, j) := \bigwedge_{k=1}^{K}(c_{j-\delta_k} = c_{i-\gamma_k} + \tau_k)$, where $\delta_k, \gamma_k \in \mathbb{N}$ and $\tau_k \in \mathbb{Z}$ are constants and $c_i$ refers to the token at position $i$. This predicate evaluates to True if the token at position $j - \delta_k$ has the same value as the token ((shifted by $\tau_k$)) at $i - \gamma_k$ for all $k$.

Such a match predicate can match tokens based on local constants $\delta, \gamma, \tau$, without memorizing identities of any token from $\mathcal{C}$. The algorithms represented by this predicate correspond to simple pattern matching operations in local neighborhoods throughout our context window. As an example, if we wish to check whether for an action $\text{pick}(b, r, g)$, its precondition $\text{at}(b, r)$ was in the initial state, the following match predicate can help us check that they refer to the same set of objects ($i, j$ denote the end of the action and the proposition respectively) $\chi(i, j) := (c_{i-2} = c_{j-1}) \wedge (c_{i-3} = c_{j-2})$. This predicate is our sole addition (in green) to **C-RASP**.

**Definition 3.4** (**C\*-RASP**). Let $\Sigma \cup \mathcal{C}$ be an alphabet, where $\Sigma$ is finite, and $\mathcal{C}$ is an increasing alphabet. Let $\Psi$ be a set of *binary relations* $\psi : \mathbb{N} \times \mathbb{N} \to \{0, 1\}$. A **C\*-RASP** program $P$ is defined as a sequence $P_1, \ldots, P_k$ of operations, where each operation is either Boolean-valued or count-valued, and can be formed according to the rules listed in Table 1.

It should be noted that only symbols from the fixed alphabet $\Sigma$ may be accessed via unary queries $Q_\sigma$, while symbols from the extended alphabet $\mathcal{C}$ can only be compared through the match predicate.[5] Counting operations return

| **Boolean-Valued Operations** | |
|---|---|
| **Initial** | $P(i) := Q_\sigma(i) \quad$ for $\sigma \in \Sigma$ |
| **Boolean** | $P(i) := \neg P_1(i)$ $P(i) := P_1(i) \wedge P_2(i)$ |
| **Constant** | $P(i) := \top$ |
| **Comparison** | $P(i) := C_1(i) \leq C_2(i)$ |
| **Count-Valued Operations** | |
| **Counting** | $C(i) := \#[j \leq i, \psi(i, j)] \ P(j)$ for $\psi \in \Psi \cup \{\top\}$ |
| **Match** | $C(i) := \#[j \leq i, \chi(i, j)]$ |
| **Conditional** | $C(i) := P(i) ? C_1(i) : C_2(i)$ |
| **Addition** | $C(i) := C_1(i) + C_2(i)$ |
| **Subtraction** | $C(i) := C_1(i) - C_2(i)$ |
| **Constant** | $C(i) := 1$ |

*Table 1.* Operations allowed in a **C\*-RASP** program.

the number of positions $j \leq i$ where $P(j)$ and $\psi(i, j)$ which represents functions such as $i = j + \delta$, for a fixed $\delta$, thus acting as a local relation ($\delta$ cannot be arbitrarily large) for checking things in local neighborhoods. As in **C-RASP**, if a **C\*-RASP** program is run on input $w$ with final operation $L$, then we accept $w$ if and only if $L(|w|)$ is true. We also write **C\*-RASP** for the class of all languages accepted by some **C\*-RASP** program. Any **C\*-RASP** program can be simulated by a Symbolic Limit Transformer. Hence, by Theorem 3.2, the existence of a **C\*-RASP** program for a task implies a guarantee of length generalization for APE transformers on that task, allowing us to prove positive length generalization results. If no $\psi \in \Psi$ is used and all constants of the type $\sigma_k, \delta_k$ are 0, then length generalization would be guaranteed even for NoPE transformers. Thus we make a distinction between **C\*-RASP** and **C\*-RASP**[Pos] to denote the necessity of positional encodings. On the other hand, we show that the difficulty of FlipFlop and PARITY carries over even to **C\*-RASP**:

**Theorem 3.5.** *Consider the alphabet $\Sigma = \{a, b, e\}$. Then, $PARITY := b^*(ab^*ab^*)^* \notin$ **C\*-RASP**. and Flip flop $:= \Sigma^*be^* \notin$ **C\*-RASP**.*

*Proof sketch.* If the input alphabet $\Omega = \Sigma \cup \mathcal{C}$ is finite, then **C\*-RASP** reduces to **C-RASP**[Pos], as the match predicate can then be simulated by the syntax of **C-RASP**[Pos] itself (Detailed Proof in Appendix, Lemma C.23). Thus, existence of a program for FlipFlop or PARITY in **C\*-RASP** would imply existence of a program in **C-RASP**[Pos], which Huang et al. (2025b) showed to be impossible. □

**Theorem 3.6.** *Let $\mathcal{D}$ be a planning domain. Consider*

---

[5]Compared to **C-RASP**[Pos] as defined by (Huang et al., 2025b), we omit operations performing modular counting on posi-

tions. These play no role in our results; we discuss technical issues about their representation in Appendix C.6

the language $L_{\mathcal{D}}$ consisting of all possible sequences $\langle I, \pi, G \rangle$ where $\pi$ is a valid plan for some planning instance $\langle \mathcal{D}, O, I, G \rangle$.

1. **(Delete-Free / Well-Formed)** *If $\mathcal{D}$ is delete-free or well-formed, then $L_{\mathcal{D}} \in$ **C\*-RASP**$[Pos]$. In particular, generalization from both limited number of objects and plan lengths during training to a increased number of objects and plan lengths is expected.*

2. **(STRIPS / Conditional Effects)** *There exists STRIPS domains as well as domains with conditional effects for which $L_{\mathcal{D}} \notin$ **C\*-RASP**$[Pos]$. Generalization to instances with longer plans or increased number of objects is thus not expected. In particular, there are domains where $L_{\mathcal{D}}$ subsumes FlipFlop or PARITY.*

*Detailed Proof is in Appendix B.1.* To show our positive results, we construct **C\*-RASP** programs along the same lines as for Theorem 3.1. The crucial difference is the use of the match predicate to match actions to propositions, without relying on specific object identities. Since STRIPS and domains with conditional effects have instances with Flip-Flop and PARITY like signatures, Theorem 3.5 applies. $\qquad\square$

## 4. Experiments

### 4.1. Planning Domains and Datasets Construction

We develop three domains to empirically validate our predictions regarding the learnability of different planning sub-classes. For each dataset we generate correct and incorrect plans, where the incorrect plans are constructed to be either *incomplete* (where the goal is not fully satisfied) or *non-executable* (where an action's preconditions are violated).[6]

**Heavy Grippers.** We modify the Heavy Grippers domain (Figure 2), and create well-formed and delete-free variants. First, we eliminate the conditional effects by separating `pick` into two actions and remove charged from the preconditions of picking up not heavy balls. Second, we create the well-formed variant by extending the preconditions and the delete-free variant by removing the delete effects.

**Colors.** This domain involves placing colored balls into bags, where actions add or remove a color from a specific bag, and goals specify which colors each bag should contain. We design well-formed and standard STRIPS variants that differ in handling redundant operations: the well-formed variant enforces that a color can be added or removed only if it is absent or present in the bag, respectively, while the standard STRIPS formulation treats these as no-ops.

---

[6]For some domains, like the standard Colors, since all actions are applicable, invalid plans can only be incomplete. Details of our domain design, data generation, and setups are in Appendix D.

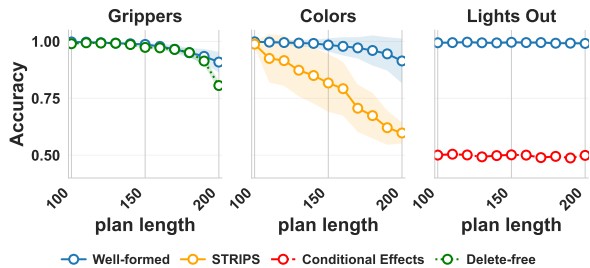

*Figure 5.* ID (100-bucket) OOD (110-200, 10-bucket) accuracy for our domains. We compare the well-formed variant to a minimal pair: delete-free for Grippers, STRIPS for Colors, and conditional effects for Lights Out. Well-formed and delete-free has good accuracy even at higher lengths, accuracy for STRIPS drops with plan length and performance for conditional effects is chance level. Our detailed results are in Figure 13.

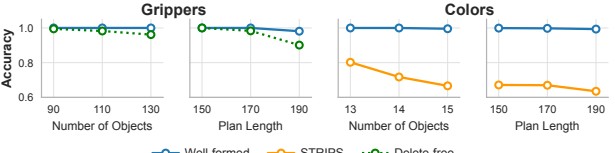

*Figure 6.* Controlled OOD evaluations separating generalization to more objects from generalization to longer plans. We either fix plan length as 100 and increase number of objects, or fix number of objects (15 for Colors, 140 for Heavy Grippers) and increase plan length. Well-formed variants generalize along both dimensions, while STRIPS Colors declines; delete-free Heavy Grippers remains strong with mild degradation. More details in Appendix E.

**Lights Out.** We formalize the Lights Out game as our final domain. It consists of lights (on/off) on a $5 \times 5$ grid where pressing a light toggles it and its neighbors. The goal is to turn all lights off. We consider two variants: one with conditional effects and a well-formed variant. In the conditional effects variant, the post-press states of the pressed light and its neighbors depend on their states before pressing; the well-formed variant uses separate actions for pressing each light under every possible combinations of its and its neighbors' states. For example, pressing the corner light $\mathsf{L}_{00}$ yields $2^3$ different actions, corresponding to the states combinations of $\mathsf{L}_{00}$ and its neighbors $\mathsf{L}_{10}$ and $\mathsf{L}_{01}$.

### 4.2. Experimental Setup & Results

**Setup:** We use the GPT-2 architecture with APE and pre-layer normalization, and train it with the standard causal language modeling objective. The initial state, plan, goal, and a verdict token indicating plan validity ('correct'/'incorrect'), each separated by a custom delimiter form our inputs:

`<init>`$I$ `<plan>`$\pi$ `<goal>`$G$ `<verdict>`$V$,

where $V$ refers to the verdict label. We train on plans of 11–100 actions and validate on 101–200 actions (length here refers to the number of actions in $\pi$, not tokens). The

numbers of objects in Colors and Heavy Grippers grow with plan length, whereas Lights Out has a fixed number of objects. We use a fixed number of objects for Lights Out to show that our negative result regarding conditional effects holds even in the setting described in Theorem 3.1, whereas the increased number of objects in the other domains helps validate the positive results regarding well-formedness and being delete-free, as described in Theorem 3.6. We use four random seeds and select the best checkpoint on a held out validation set. At test time, the verdict token is omitted for the model to generate it. We evaluate on both in-distribution (ID, lengths 11–100) and out-of-distribution (OOD, lengths 101–200) test sets.

**Results:** Our experiments (Figure 5) corroborate our theory. For Lights Out, we observe a sharp contrast: the conditional effects formulation yields chance-level performance even on a *fixed* set of objects, whereas the well-formed variant generalizes perfectly despite having an exponential action space, corroborating Theorem 3.1. Our models successfully generalize to *longer plans* and *more objects* in Heavy Grippers and Colors as predicted by Theorem 3.6. Specifically, the well-formed variants of both—as well as the delete-free variant of Heavy Grippers (all in ∈ **C\*-RASP**)—achieve high accuracy, while for the STRIPS variant of Colors (∉ **C\*-RASP**), the performance dips. We disentangle the effects of object-count growth and plan-length growth in Figure 6, by varying either the number of objects or the plan length and keeping the other fixed. The same pattern holds: well-formed and delete free variants do show generalization for Colors and Heavy Grippers, where as the STRIPS variant for Colors declines when either of the objects or plan length increases. Overall, generalization succeeds where **C\*-RASP** membership holds and fails otherwise. [7]

# 5. Discussion

## 5.1. Implications

Our results establish a sharp divide in the learnability of plan verification by transformers: generalization in delete-free and well-formed instances is possible; but for unconstrained STRIPS domains it is not, with conditional effects making it even harder. Our negative results highlights why transformers have had inconsistent success in planning tasks. Conversely, our positive results are significant given that a broad class of practical planning problems are well-formed. For instance, 70% of the domains from the Learning Track 2023 (Taitler et al., 2024), and a lot of domains in the International Planning Competition (IPC) (McDermott, 2000) come with benchmark problem generators that can only produce well-formed instances. We achieve this breadth

through a "semantic" definition of well-formedness in terms of the actual dynamics of the actions and states; much more permissive than the syntax-based definition of Gösgens et al. (2025), which is rarely satisfied by planning benchmarks. We also find that the exact formulation of the planning task matters: for instance, the Lights Out game when formulated with conditional effects does not generalize, but a compilation of this domain into a well-formed one does. This confirms that the *structural complexity* of the planning problem, rather than just the number of actions or objects, drives the ability of transformers to learn it. Thus, if one can reformulate the problem structure, transformers could have marked improvement in performance.

It is also worth noting that the ramifications of Theorem 3.2 (More Formal Statement C.10), and **C\*-RASP** which establishes a framework to give length generalization guarantees for transformers with growing alphabets at test time, are far beyond plan verification. For instance, they open up the possibility to reason about how powerful can transformers with a growing index hint vocabulary be, as even a limited set of index hints have shown to help improve generalization in transformers (Zhou et al., 2024).

## 5.2. Related Work

Transformers have been applied extensively to planning tasks (Yao et al., 2023; Silver et al., 2024; Stein et al., 2025; Hirsch et al., 2024; Pallagani et al., 2023; Fritzsche et al., 2026; Chen et al., 2025a). While LLMs can generate plans when test data instances are of the same size as training (Rossetti et al., 2024), they struggle on larger instances (Huang et al., 2025a; Fritzsche et al., 2026). A recurrent observation is a sharp performance degradation on longer plans (Stein et al., 2025; Valmeekam et al., 2025; Hirsch et al., 2024; Pallagani et al., 2023; Hazra et al., 2024), underscoring the need to better evaluate length generalization in planning (Chiari et al., 2025). Comparatively, *plan verification* has only been studied by a few (Stechly et al., 2025; Valmeekam et al., 2023). Stechly et al. (2025) found that LLMs struggle to verify self generated plans, while Valmeekam et al. (2023) found LLMs to be better at verification than generation for a STRIPS domain, but overall performance remained low.

Prior work has also tried to connect computational complexity of classical planning (Bylander, 1994) to formal language theory (Höller et al., 2016; Lin & Bercher, 2022), albeit not in the context of transformer expressiveness. In contrast, a rich line of research has looked into transformers expressiveness through the lens of formal language theory and computational complexity. This includes the design of formalisms that capture transformer computations, such as RASP (Weiss et al., 2021), B-RASP (Yang et al., 2024), C-RASP (Yang & Chiang, 2024) and results linking models to complexity classes such as $TC^0$ (Merrill & Sabharwal,

---

[7]Our code is available at: https://github.com/coli-saar/transformers_plan_verification

2023; Strobl, 2023). Zhou et al. (2024) conjectured that only programs expressible in RASP-L exhibit length generalization, a notion formalized by Huang et al. (2025b) using the C-RASP formalism (Yang & Chiang, 2024). The C-RASP length generalization framework has since seen substantial success, with results demonstrated at scale (Jobanputra et al., 2025), quantitative bounds on training length (Izzo et al., 2026), and even predictions of the model depth required to solve certain problems (Yang et al., 2025). More broadly, length generalization remains an active area of research, with approaches exploring scratchpads, positional embeddings, and synthetic data generation (Anil et al., 2022; Abbe et al., 2023; Zhou et al., 2024; Golowich et al., 2025; Hou et al., 2025; Xiao & Liu, 2025; Press et al., 2022; He et al., 2024; Cho et al., 2024; Lee et al., 2025; Chen et al., 2025b).

There have been attempts to enable vocabulary extensions in Transformers, especially when the intended extension of vocabulary involves generalizing to distinct symbols that are semantically equivalent, for instance in Işık et al. (2025). Unlike Işık et al. (2025), who advocate for randomized permutations of the interchangeable tokens (here the alphabet $\mathcal{C}$) we enforce a character ordering in $\mathcal{C}$. *Removing* this ordering will *not* affect our planning theorems. Its role is to make the match predicate more general. In particular, the term $\tau_k$ in Definition 3.3 lets a program match not only two identical new symbols, but also symbols that are a fixed small shift apart in the extended embedding/index space. Without an ordering on $\mathcal{C}$, this operation would not be possible, and **C\*-RASP**[Pos] would become a more restrictive interface to Symbolic Limit Transformers. We believe this additional structure is precisely what would help formalize why index hints improve length generalization in prior work (Zhou et al., 2024): a model could use the consistent local ordering of index tokens to move from one index to a nearby one, while the hints provide local context for disambiguation (Kraus et al. (2026) takes some first steps in doing so). Thus, the approaches of Işık et al. (2025) and our ordered-alphabet framework address complementary aspects of vocabulary generalization, and could potentially be combined as well.

Our work is most closely related to Núñez-Molina et al. (2025), who show that plan verification for propositional STRIPS is expressible in B-RASP, and thus realizable by a restricted class of transformers (Yang et al., 2024) that uses hard attention. This result is complementary to ours, but differs in key respects. First, their analysis is situated in a fixed-vocabulary, grounded setting, whereas a central motivation of our work is generalization to longer plans and instances with more objects, where the effective input alphabet grows. This distinction is important, since the desired behavior is typically to train on small grounded instances and generalize to larger ones. Secondly, we study *generalization in learning* rather than in principle expressivity, which is what B-RASP guarantees. Our experiments support the relevance of this distinction: although B-RASP-style expressivity suggests that STRIPS verification traces are representable in principle, standard transformers fail on the STRIPS/FlipFlop-style cases in the way predicted by the C-RASP-based framework. Thus, our results are not implied by theirs; rather, they identify a length-generalization limitation for standard transformers that is invisible from the B-RASP expressivity perspective.

### 5.3. Limitations & Future Work

Our theoretical results on length generalization apply to standard transformers without scratchpads (no chain of thought). However, our results are still relevant, as chain of thought is expensive and knowing what is possible without Chain of thought is important. Much like **C-RASP**, our theory also only applies to APE and NoPE encodings, extending our theory to cover Rotary Positional Encodings is a natural next step. Although **C-RASP** is empirically predictive, length generalization guarantees given through it rely on idealized learning procedures. Our learning framework is inspired by Huang et al. (2025b) and so we also have the limitation of using a convenient tool for proving results but one that is not equipped to reflect subtleties of training dynamics. The framework makes several notable idealizations, such as assuming translation invariance, matching the target function on all training-distribution inputs (rather than only a bounded training set), and idealizing activation functions. Finally, our focus was on verification and not plan generation. As Stechly et al. (2025) mentions, their respective computational complexity (P vs PSPACE-complete for propositional STRIPS) should not be expected to translate to relative transformer performance. Thus generalizing our findings to plan generation is non-trivial. Our proofs rely on the existence of valid plans containing loops, whereas optimal plans could be much shorter. Thus, while our results serve as a foundational step, characterizing the learnability of plan generation remains exciting grounds for future work.

## 6. Conclusion

We presented a formal analysis of the ability of transformers to verify plans, focusing on the challenge of length generalization. To address the setting where the number of objects grows at test time, we introduced **C\*-RASP**, a novel extension of the **C-RASP** framework that characterizes learnability under simultaneous growth of sequence length and vocabulary size. Our results indicate that the specific formulation of a planning problem—rather than only the number of objects or actions—is a decisive factor in learnability. These findings take steps toward explaining inconsistent performance of transformers on planning tasks, opening new avenues for characterizing their capabilities in planning.

## Impact Statement

This paper presents work whose goal is to advance the field of Machine Learning. There are many potential societal consequences of our work, none of which we feel must be specifically highlighted here.

## Acknowledgments

Katharina Stein was funded by the Deutsche Forschungsgemeinschaft (DFG, German Research Foundation) under the project number 232722074 – SFB 1102. We gratefully acknowledge the stimulating research environment of the GRK 2853/1 "Neuroexplicit Models of Language, Vision, and Action", funded by the Deutsche Forschungsgemeinschaft under project number 471607914. Sylvie Thiébaux's work was funded by the Australian Research Council (ARC) under the Discovery Project grant DP220103815 and by the Artificial and Natural Intelligence Toulouse Institute (ANITI) under the grant agreement ANR-23-IACL-0002. We thank Chaahat Jain for insightful discussions.

## Contributions

YS drafted the paper, contributed Theorem 3.1, 3.6 and their corresponding proofs (Appendix A, B) with inputs from ST and MH, and coordinated the experiments with KS and YD. YS and MH jointly developed the learning framework, symbolic limit transformer translation, C*-RASP formulation (Section 3.2 and Appendix C). YD developed the training and evaluation pipeline, trained all models, implemented the Colors domain dataset in use, and contributed substantially to the design and development of the other two domain datasets. He also contributed to writing. KS led the development of the datasets and contributed substantially to their designs. She conducted the implementation and generation of the Grippers Heavy and Lightsout datasets, and contributed to the writing. AK advised on the research direction and narrative framing, supervised the experimental work, and contributed to the writing of the paper. ST contributed to the research framing, to the identification of the generalisable planning classes, to the formulation of theorems 3.1 and 3.6, and proposed the Color domain. She drafted section 2 and provided feedback and corrections on every section except Appendix C. MH contributed to the research framing and to the identification of the generalisable planning classes, and jointly developed the learning framework, limit transformers, and C*-RASP formulation with YS.

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

# A. Proofs on Plan Verification

## A.1. Proofs about Generalization over a Fixed Object Universe

We restate Theorem 3.1 and then prove it.

**Theorem A.1.** *Let $\mathcal{D}$ be a planning domain and $O$ a fixed set of objects. Consider the language $L_{\mathcal{D},O}$ consisting of all sequences $\langle I, \pi, G \rangle$ where $\pi$ is a valid plan for the planning instance $\langle \mathcal{D}, O, I, G \rangle$.*

1. *(**Delete-Free & Well-Formed**) If $\mathcal{D}$ is delete-free or well-formed, then $L_{\mathcal{D},O} \in$ **C-RASP**[Pos].*

2. *(**STRIPS**) There exist STRIPS domains and object sets such that $L_{\mathcal{D},O} \notin$ **C-RASP**[Pos].*

3. *(**Conditional Effects**) There exist domains with conditional effects such that $L_{\mathcal{D},O}$ is neither star-free nor in **C-RASP**[Pos].*

*Proof of Statement 1.* In this setting, the domain $\mathcal{D}$ and the set of objects $O$ are fixed and finite. However, the initial state $I$ and the goal state $G$ are provided as part of the input sequence, and can vary per sample. Given this setting, all possible instances will have the set of all possible ground propositions $P$, and the same set of all possible ground actions $A$, where both $P, A$ are finite as well. Because of this we can directly refer to the necessary $P$, which is all the set of possible propositions, and $A$ which is the entire set of ground actions (i.e. after the object names have been substituted into the predicates and action schemas taken from $\mathcal{P}$ and $\mathcal{A}$ respectively). The initial state $I \subseteq P$, and the goal state $G$ are as defined in Section 2.1.

We now give a recipe for constructing a **C-RASP**[Pos] program for any such input, where the given plan $\pi$ is a valid action sequence and upon starting from $I$ and taking all the actions in $\pi$ all the goal conditions listed in $G$ are met.

We assume that the input sequence is as follows $w = \$I@\pi@G@.$ where,

$$I = p_1, po_{1,1}, \ldots, po_{1,pk_1}, \quad p_2, po_{2,1}, \ldots, po_{2,pk_2}, \quad \ldots$$

$$\pi = \alpha_1, o_{1,1}, \ldots, o_{1,k_1}, \quad \alpha_2, o_{2,1}, \ldots, o_{2,k_2}, \quad \ldots$$

$$G = g_1, go_{1,1}, \ldots, go_{1,gk_1}, \quad g_2, go_{2,1}, \ldots, go_{2,gk_2}, \quad \ldots$$

Thus, in the $I$ block, the token $p_t$ represents a predicate name, followed by $pk_t$ (arity of the predicate) objects representing its arguments, Thus $p_t, po_{t,1} \ldots po_{t,pk_t} \in P$. Similarly, in the the token $\pi$ block, $\alpha_t$ is an action schema name, followed by $k_t$ object tokens representing its arguments, and $\alpha_t, o_{t,1} \ldots o_{t,k_t} \in A$ Finally the $G$ block is also similar to the $I$ block and therefore $g_t, go_{t,1} \ldots go_{t,gk_t} \in P$. We need to verify the plan by tracking the state of every ground proposition in $P$, making sure that all preconditions of a given action was always satisfied and also checking that all the goal conditions were indeed met by the end of the $w$.

1. We must identify which part of the input a token belongs to, using cumulative counts of separator tokens.

$$\text{CountSep}(i) := \#\,[j \leq i]\ Q_@(j)$$

Using this, we define the section selectors:

- $\text{InInit}(i) := (\text{CountSep}(i) == 0)$
- $\text{InPlan}(i) := (\text{CountSep}(i) == 1)$
- $\text{InGoal}(i) := (\text{CountSep}(i) == 2)$
- $\text{End}(i) := (\text{CountSep}(i) == 3)$

The $==$ syntax is equivalent to checking $(\text{CountSep}(i) \leq k) \wedge (\text{CountSep}(i) \geq k)$.

2. We first identify, at any index $i$, whether an action has just completed and what that action is. Let $a \in A$ be a specific ground action, defined by schema $\alpha$ and argument objects $u_1, \ldots, u_k$. The action $a$ ends at index $i$ if the sequence of tokens ending at $i$ matches the signature of $a$:

$$\text{token at } i = u_k, \quad \text{token at } i-1 = u_{k-1}, \quad \ldots, \quad \text{token at } i-k = \alpha$$

In C-RASP, we can implement lookbacks using the query relation $\psi(i, j) \equiv (i = j + \Delta)$. For each ground action $a$, we define a boolean selector sequence $Occurs_a(i)$ that is true iff the action $a$ ends at $i$:

$$\text{OccursAction}_a(i) := \left( (\#\,[j \leq i, i = j + k]\ Q_\alpha(j) == 1) \wedge \bigwedge_{m=1}^{k} (\#\,[j \leq i, i = j + m]\ Q_{u_m}(j) == 1) \wedge \text{InPlan}(i) \right)$$

This expression verifies the action schema name at offset $k$ and every object argument $u_m$ at the correct offset. Since there are finitely many ground actions, we define one such line for every $a \in A$.

Thus at every position i, at most only 1 ground action can ever be true. We do the same thing for every $p \in P$ to match propositions listed in the initial and the goal state. ($u_1, \ldots u_k$ representing the arguments of the proposition $p$, $k$ the arity of the proposition).

$$\text{OccursInitial}_p(i) := \left( (\#\,[j \leq i, i = j + k]\ Q_p(j) == 1) \wedge \bigwedge_{m=1}^{k} (\#\,[j \leq i, i = j + m]\ Q_{u_m}(j) == 1) \wedge \text{InInit}(i) \right)$$

$$\text{OccursGoal}_p(i) := \left( (\#\,[j \leq i, i = j + k]\ Q_p(j) == 1) \wedge \bigwedge_{m=1}^{k} (\#\,[j \leq i, i = j + m]\ Q_{u_m}(j) == 1) \wedge \text{InGoal}(i) \right)$$

3. For every ground proposition $p \in P$, we identify the set of ground actions that add it, $TP_p$, where $TP_p = \{a \in A | p \in \text{eff}(a)^+\}$. These sets will be fixed and known a priori, given our fixed objects and domain.

We define the cumulative count of additions for $p$ at step $i$:

$$\text{MadeTrue}_p(i) := (\sum_{a \in TP_p} \#\,[j \leq i]\ \text{OccursAction}_a(j))$$

We do the same for the set of actions that delete it $FP_p = \{a \in A | p \in \text{eff}(a)^-\}$.

$$\text{MadeFalse}_p(i) := (\sum_{a \in FP_p} \#\,[j \leq i]\ \text{OccursAction}_a(j))$$

4. Now, we define the truth value of a proposition $p$ at step $i$.

- **Well-Formed**: Because of well-formedness, the truth value of a proposition toggles (False $\rightarrow$ True $\rightarrow$ False). We compute the net flow (adds minus deletes) for each proposition at every step. If the proposition was in the initial state, then the net flow has to be $0$, and if it was not then this net flow should be $1$.

$$\text{Net}_p(i) := \text{MadeTrue}_p(i) - \text{MadeFalse}_p(i)$$
$$\text{Valid}_p(i) := (\text{OccursInitial}_p(i) \wedge \text{Net}_p(i) == 0) \vee (\neg\text{OccursInitial}_p(i) \wedge \text{Net}_p(i) == 1)$$

- **Delete-Free**: The proposition is true, if it was either already true (as it was in the initial state) or at least one action made it true.

$$\text{Valid}_p(i) := \text{OccursInitial}_p(i) \vee (\text{MadeTrue}_p(i) \geq 1)$$

5. For every ground action a, let $\text{pre}(a)^+$, $\text{pre}(a)^-$ define the list of positive and negative ground propositions for a given action. We need to check that all the preconditions are satisfied *before* the action effect takes place. We check the state of preconditions at $i - 1$ (the state before the effects of the action took place). If any of the preconditions are not met, we mark this position as invalid.

$$\text{PreCondPassed}_a(i) := \bigwedge_{p \in \text{pre}(a)^+} (\#\,[j \leq i, i = j + 1]\ \text{Valid}_p(j) \bigwedge_{p \in \text{pre}(a)^-} (\#\,[j \leq i, i = j + 1]\ \neg\text{Valid}_p(j)$$

$$\text{InvalidAction}_a(i) := \text{OccursAction}_a(i) \wedge \neg\text{PreCondPassed}_a(i)$$

The count of invalid actions by the end of $w$ should be 0.

$$\text{AllActionsValid(i)} := (\sum_{a \in A} \#\,[j \leq i]\ \text{InvalidAction}_a(j) == 0)$$

6. Finally, we need check if every goal condition is satisfied. We iterate over our ground propositions. For every proposition, either it does not appear in the goal or it is valid.

$$\text{GoalSat}(i) := \bigwedge_{p \in P} \neg\text{OccursGoal}_p(i) \vee \text{Valid}_p(i)$$

At the end token, we need to check, that all goals were met.

$$\text{AllGoalsMet}(i) := \text{End}(i) \wedge \text{AllActionsValid}(i) \wedge (\text{\#}\,[j \leq i]\ \neg\text{GoalSat}(i) == 0)$$

Note that, in case negative goal conditions are present, we can check for invalidity of the underlying proposition, however the logic remains the same.

Since $A$ and $P$ are finite, this program has a finite number of lines and operations. At the last position of our sequence, our program outputs True if the given input sequence was valid else not for any possible sequence in $L_{\mathcal{D},O}$. Hence, for both delete-free and well-formed, $L_{\mathcal{D},O} \in$ **C-RASP**[Pos].

$\square$

For our negative results, we will show that there exist instances $\Pi = \langle \mathcal{D}, O, I, G \rangle$, where the language of all valid plans of just that instance $L(\Pi)$ (therefore subsets of the full language $L_{\mathcal{D},O}$) are not contained in **C-RASP**[Pos]. Consequently, the language $L_{\mathcal{D},O}$ itself is not in **C-RASP**[Pos].

*Proof of Statement 2.* Lin & Bercher (2022) showed that $L(\Pi)$ when $\Pi = \langle \mathcal{D}, O, I, G \rangle$ represents a STRIPS instance belongs to a strict subset of star free languages. Here we will show that there could be a STRIPS instance $\Pi$, where $L(\Pi) \notin$ **C-RASP**[pos]. Huang et al. (2025b) had rigorously showed that the star free language $\Sigma^* be^*$ ($\Sigma = \{a, b, e\}$), also known as the flipflop language is not in **C-RASP**[pos]. Thus, we will now create a STRIPS instance $\Pi_{ff}$ where $L(\Pi_{ff}) = \Sigma^* be^*$, which will thus prove our claim.

Let the planning domain be $\mathcal{D}_{ff} = \langle \mathcal{P}, \mathcal{A} \rangle$ where $\mathcal{P} = \{\text{active}\}$, arity(active) = 1 and $\mathcal{A} = \{a_a, a_b, a_e\}$. All three schemas share the argument set args() = $\{x\}$ and empty preconditions pre() = $\emptyset$. Their effects are: eff($a_a$) = $\{\neg\text{active}(x)\}$, eff($a_b$) = $\{\text{active}(x)\}$, eff($a_e$) = $\emptyset$. Now we define the planning instance to be $\Pi_{ff} = \langle \mathcal{D}_{ff}, \{k\}, \emptyset, \text{active}(k) \rangle$. This planning instance is a STRIPS instance, as by construction we did not violate any rules of being in STRIPS. The ground actions correspond directly to the alphabet $\Sigma$: $a = a_a(k)$, $b = a_b(k)$, and $e = a_e(k)$. Note that since the preconditions are empty, all actions are applicable in every state. Let $\pi \in \Sigma^*$ be a sequence of actions $\pi = \sigma_1 \ldots \sigma_n$. We track the state $S_i = \text{succ}(I, \sigma_1 \ldots \sigma_i)$. The ground proposition set is simply $\{\text{active}(k)\}$. Thus, there are only two possible states: $\emptyset$ (which we call OFF) and $\{\text{active}(k)\}$ (which we call ON).

Since $I = \emptyset$, the system starts in the OFF state. Based on the definitions of the ground actions each action regardless of the previous state acts as follows – $a$: eff$(a)^- = \{\text{active}(k)\}$ removes the proposition, setting the state to off, $b$: eff$(b)^+ = \{\text{active}(k)\}$ adds the proposition, setting the state to on, $e$: eff$(e) = \emptyset$ acts as the identity function, leaving the state unchanged.

A plan $\pi$ is valid iff $S_n \models G$, which means active(k) $\in S_n$ (i.e., the final state is on). Consider the last occurrence of a non-$e$ symbol in $w$. If $w$ contains no $a$ or $b$ (i.e., $w \in e^*$), the state remains $I$ (off). The goal is not satisfied. If the last non-$e$ symbol is $a$, the state becomes off at that step. Any subsequent $e$ actions maintain the off state. The goal is not satisfied. If the last non-$e$ symbol is $b$, the state becomes ON at that step. Any subsequent $e$ actions maintain the ON state. The goal is satisfied. Therefore, $w \in L(\Pi_{ff})$ if and only if $w$ contains at least one $b$, and no $a$ occurs after the last $b$. This corresponds exactly to the regular expression $\Sigma^* be^*$. $\square$

*Proof of Statement 3.* We will prove this by constructing a planning instance for a domain with conditional effects, and show that its valid language has PARITY as the the preimage of a length preserving monoid homomorphism, and thus if this language would be either star free or in **C-RASP**[Pos] it would contradict well known prior results.

**Definition A.2** (Lights Out with conditional effects). Let $LO = (V, E)$ be a lights out board represented as a graph where $V = \{v_1, \ldots, v_n\}$ represents the set of cells (vertices) and $E$ defines the connectivity (edges between the vertices). For any $v \in V$, let $N[v] = \{u \in V \mid (v, u) \in E\} \cup \{v\}$ be the closed neighborhood of $v$. Given an initial configuration of cells $I \subseteq V$ that are on, we define the corresponding planning instance $\Pi_{LO} = \langle \mathcal{D}, O, I, G \rangle$ as follows:

- The set of objects is empty, $O = \emptyset$.

- The domain is $\mathcal{D} = \langle \mathcal{P}, \mathcal{A} \rangle$.

    - The predicates are propositional (arity 0), representing the state of each cell: $\mathcal{P} = \{\text{on}_i \mid v_i \in V\}$
    - The action schema correspond to pressing specific cells. Since the logic depends on the fixed graph topology, we define a schema for each cell: $\mathcal{A} = \{a_1, \ldots, a_n\}$ For each cell $v_k \in V$, the schema $a_k$ is defined by:
        * $\text{args}(a_k) = \emptyset$
        * $\text{pre}(a_k) = \emptyset$ (actions are always applicable)
        * The effects $\text{eff}(a_k)$ toggle the cell and its neighbors. For every $v_j \in N[v_k]$, we include the following pair of conditional effects:[8]
        $$\{\text{on}_j\} \rhd \{\neg \text{on}_j\}$$
        $$\{\neg \text{on}_j\} \rhd \{\text{on}_j\}$$

- The initial state is defined by the set of cells that are initially on: $I = \{\text{on}_i \mid v_i \in I\}$

- The goal is to switch all lights off: $G = \{\neg \text{on}_1, \ldots, \neg \text{on}_n\}$

We now use the formal language $\texttt{PARITY} = b^*(ab^*ab^*)^* \subseteq \{a, b\}^*$. Thus, a word belongs to $\texttt{PARITY}$ iff it contains an even number of $a$'s. This language is not star-free, and Huang et al. (2025b) rigorously showed that $\texttt{PARITY} \notin \textbf{C-RASP}[\text{Pos}]$.

We will use a simple coding of words over $\{a, b\}$ as Lights Out action sequences. Formally, this coding is a monoid homomorphism $\psi : \{a, b\}^* \to A_{LO}^*$, where $A_{LO}$ is the finite set of ground Lights Out actions. Since $O = \emptyset$ and all action schemas have empty arguments, every action schema is already ground, so $A_{LO} = \{a_1, \ldots, a_n\}$. Thus $A_{LO}^*$ is the set of all possible finite Lights Out action sequences, and is thus quite distinct from $L(\Pi_{LO})$ which only contains action sequences with valid plans.

A monoid homomorphism is determined by its values on individual letters and is extended to words by concatenation. Hence, if $w = w_1 \cdots w_m$ with each $w_i \in \{a, b\}$, then $\psi(w) = \psi(w_1) \cdots \psi(w_m)$.

We will define $\psi$ so that
$$w \in \texttt{PARITY} \iff \psi(w) \in L(\Pi_{LO}).$$

Equivalently,
$$\psi^{-1}(L(\Pi_{LO})) = \texttt{PARITY},$$

where
$$\psi^{-1}(L(\Pi_{LO})) = \{w \in \{a, b\}^* \mid \psi(w) \in L(\Pi_{LO})\}.$$

This notation denotes the preimage of $L(\Pi_{LO})$ under $\psi$. Thus, we will show that if one could recognize $L(\Pi_{LO})$ either via a star free expression or by a **C-RASP**[Pos] program, then one could also recognize $\texttt{PARITY}$ by first applying $\psi$ and then recognizing $L(\Pi_{LO})$.

More formally, both language classes used in the argument are closed under the appropriate form of preimage. Star-free languages are closed under preimages of monoid homomorphisms. Similarly, Lemma 42 of Huang et al. (2025b) shows that **C-RASP**[Pos] is closed under preimages of homomorphisms that map every input symbol to a word of the same length. Therefore, once we prove
$$\psi^{-1}(L(\Pi_{LO})) = \texttt{PARITY},$$

it follows that membership of $L(\Pi_{LO})$ in either class would imply the corresponding membership of $\texttt{PARITY}$. Since $\texttt{PARITY}$ is known to be neither star-free nor in **C-RASP**[Pos], this gives us the desired contradiction. Now we will construct these homomorphisms.

---

[8]For compactness, we factorize conditions, with the semantics that all conditions that are true in the current state trigger their effects and that consistent sets of conditions must lead to consistent effect sets. This does not affect our results.

**Constructing the Homomorphism.** We will analyze the plan language algebraically, and in turn be able to construct our desired homomorphism. Since the conditional effects toggle truth values, the underlying states (changed due to actions) can be represented as vectors over the field $\mathbb{F}_2 = (\{0, 1\}, +, \cdot)$ Let $n = |V|$, and $\mathfrak{P}$ denote power sets. We can define

$$\phi : \mathfrak{P}(\mathcal{P}) \to \mathbb{F}_2^n$$

so that the $k$-th coordinate of $\phi(S)$ is 1 iff $\text{on}_k \in S$, and 0 otherwise. The initial state maps to $b = \phi(I)$ and the unique state satisfying the all-off goal maps to $\vec{0}$. For each action $a_k \in A_{LO}$, let $e_k \in \mathbb{F}_2^n$ be its effect vector, where the $j$-th coordinate is 1 iff $v_j \in N[v_k]$, and 0 otherwise. Pressing $a_k$ toggles exactly those coordinates, so for every state $S$,

$$\phi(\text{succ}(S, a_k)) = \phi(S) + e_k.$$

With this, we now define the plan homomorphism

$$h : A_{LO}^* \to \mathbb{F}_2^n$$

where

$$h(a_1 \cdots a_m) = e_{a_1} + \cdots + e_{a_m},$$

where each $a_\ell \in A_{LO}$ and $e_{a_\ell}$ is the effect vector of that action. We also set $h(\epsilon) = \vec{0}$. Then $h$ is a monoid homomorphism, for all $w_1, w_2 \in A_{LO}^*$, where $h(w_1 w_2) = h(w_1) + h(w_2)$.

A plan $w \in A_{LO}^*$ is valid exactly when applying its total effect to the initial state reaches the all-off state. Therefore, $b + h(w) = \vec{0}$, and equivalently $h(w) = b$. Thus the valid-plan language is

$$L(\Pi_{LO}) = \{w \in A_{LO}^* \mid h(w) = b\}.$$

**The all-off instance.** Now consider the instance where all lights are initially off, i.e. $I = \emptyset$, then $b = \vec{0}$. Thus, if $\ker(h)$ denotes the kernel of $h$, we get

$$L(\Pi_{LO}) = \{w \in A_{LO}^* \mid h(w) = \vec{0}\} = \ker(h).$$

Choose a Lights Out board with two actions whose effect vectors are distinct; for example, a path on three vertices has this property. Let $\gamma, \beta \in A_{LO}$ be such that $e_\gamma \neq e_\beta$. Equivalently, $d := e_\gamma + e_\beta \neq \vec{0}$. Choose any action $\alpha \in A_{LO}$. We finally can define our desired morphism,

$$\psi : \{a, b\}^* \to A_{LO}^*$$

by

$$\psi(a) = \gamma\beta, \qquad \psi(b) = \alpha\alpha.$$

Both input letters are mapped to action sequences of length 2. The block coding $b$ has no net effect:

$$h(\psi(b)) = h(\alpha\alpha) = e_\alpha + e_\alpha = \vec{0}.$$

The block coding $a$ has nonzero net effect:

$$h(\psi(a)) = h(\gamma\beta) = e_\gamma + e_\beta = d \neq \vec{0}.$$

Since we work over $\mathbb{F}_2$, we also have $d + d = \vec{0}$. Therefore, for every word $w \in \{a, b\}^*$,

$$h(\psi(w)) = \#_a(w) \cdot d,$$

where $\#_a(w)$ denotes the number of a's in the word $w$. Here the coefficient is taken modulo 2. Hence,

$$h(\psi(w)) = \vec{0} \quad \Longleftrightarrow \quad \#_a(w) \text{ is even}.$$

Because $L(\Pi_{LO}) = \ker(h)$, we obtain

$$\psi^{-1}(L(\Pi_{LO})) = \{w \in \{a, b\}^* \mid \#_a(w) \text{ is even}\} = \texttt{PARITY}.$$

Now suppose first that $L(\Pi_{LO})$ were star-free. Star-free languages are closed under preimages of monoid homomorphisms, so $\psi^{-1}(L(\Pi_{LO}))$ would also be star-free. But this preimage is exactly PARITY, which is not star-free. This is a contradiction.

Next suppose that $L(\Pi_{LO}) \in$ **C-RASP**[Pos]. By Lemma 42 of Huang et al. (2025b), **C-RASP**[Pos] is closed under preimages of homomorphisms that map every input symbol to a word of the same length. Since $|\psi(a)| = |\psi(b)| = 2$, we would have $\psi^{-1}(L(\Pi_{LO})) \in$ **C-RASP**[Pos]. Again this preimage is PARITY, contradicting Huang et al. (2025b). Therefore $L(\Pi_{LO})$ is neither star-free nor in **C-RASP**[Pos]. □

## B. Proofs about Generalization over a Planning instances with growing number of objects

We restate Theorem 3.6 and then prove it.

**Theorem B.1.** *Let $\mathcal{D}$ be a planning domain. Consider the language $L_{\mathcal{D}}$ consisting of all possible sequences $\langle I, \pi, G \rangle$ where $\pi$ is a valid plan for some planning instance $\langle \mathcal{D}, O, I, G \rangle$.*

1. *(**Delete-Free / Well-Formed**) If $\mathcal{D}$ is delete-free or well-formed, then $L_{\mathcal{D}} \in$ **C\*-RASP**[Pos]. In particular, generalization from both limited number of objects and plan lengths during training to a increased number of objects and plan lengths is expected.*

2. *(**STRIPS / Conditional Effects**) There exists STRIPS domains as well as domains with conditional effects for which $L_{\mathcal{D}} \notin$ **C\*-RASP**[Pos]. Generalization to instances with longer plans or increased number of objects is thus not expected. In particular, there are domains where $L_{\mathcal{D}}$ subsumes FlipFlop or PARITY.*

We start by proving Statement 1

*Proof.* We represent a planning instance and plan as a sequence of tokens $w$. Along the same lines as Theorem A.1, the sequence is composed of the initial state $I$, and action names obtained by grounding action schemas from the finite set $\mathcal{A}$ and object names $O$ per sample taken from the potentially infinite alphabet $\mathcal{C}$, and the goal $G$.

We will define $\mathfrak{A}$ as follows. Each $\alpha \in \mathfrak{A}$ has the form $n_\alpha(x_1, \ldots x_{\mathrm{arity}(\alpha)})$ where $n_\alpha$ is the name of the action schema, and and the $x_i \in \mathrm{args}(\alpha)$ refer to the placeholder objects. This set can be constructed from the action schemas $\mathcal{A}$ set by ignoring the schemas' preconditions and effects, with a one to one mapping between $\mathcal{A}$ and $\mathfrak{A}$. It should be noted that we are constructing this set just for notational convenience and to avoid ambiguity.

We write it as $w = \$I@\pi@G@.$ where,

$$I = n_{p_1}, po_{1,1}, \ldots, po_{1,pk_1}, \quad p_2, po_{2,1}, \ldots, po_{2,pk_2}, \quad \ldots$$

$$\pi = n_{\alpha_1}, o_{1,1}, \ldots, o_{1,k_1}, \quad \alpha_2, o_{2,1}, \ldots, o_{2,k_2}, \quad \ldots$$

$$G = n_{g_1}, go_{1,1}, \ldots, go_{1,gk_1}, \quad g_2, go_{2,1}, \ldots, go_{2,gk_2}, \quad \ldots$$

In the $I$ block, the token $n_{p_t}$ represents a predicate name, followed by $pk_t$ (arity of the predicate) objects representing its arguments. Thus each $n_{p_t}, po_{t,1} \ldots po_{t,pk_t}$ represents a ground proposition where the proposition name and object structure are taken from $\mathcal{P}$, and each object that has been instantiated is taken from $O$. The key difference here is that $P$, the set of all ground propositions is now infinite, and hence cannot be used. Similarly, in the $\pi$ block, $n_{\alpha_t}$ is an action schema name, followed by $k_t$ object tokens representing its arguments. Therefore, $n_{\alpha_t}, o_{t,1} \ldots o_{t,k_t}$ follows the structure of an $n_{\alpha_t}(x_1, \ldots, x_{\mathrm{arity}(\alpha_t)}) \in \mathfrak{A}$. Finally the $G$ block is also similar to the $I$ block and therefore $n_{g_t}, go_{t,1} \ldots go_{t,gk_t}$. For $n_{\alpha_t}, n_{p_t}$ and $n_{g_t}$, the objects that follow it are taken from the infinite pool of objects $O$. Thus, in addition to the ground propositions, the number of ground actions are also infinite.

We will be constructing **C\*-RASP**[Pos] programs to verify action preconditions and goals for our positive cases. Note that unlike in **C-RASP**, here we will not be able to use object names inside the $Q_\sigma$ operator to check for the presence of a specific object at the current position (owing to the increased number of objects), and will instead rely on the match predicate to help us resolve our issue.

1. *Section Identification:* we must identify which part of the input a token belongs to, using cumulative counts of separator tokens.

$$\text{CountSep}(i) := \# \, [j \leq i] \; Q_@(j)$$

   - $\text{InInit}(i) := (\text{CountSep}(i) == 0)$
   - $\text{InPlan}(i) := (\text{CountSep}(i) == 1)$
   - $\text{InGoal}(i) := (\text{CountSep}(i) == 2)$
   - $\text{End}(i) := (\text{CountSep}(i) == 3)$

   The $==$ syntax is not explicitly listed in **C\*-RASP**[Pos], but it is equivalent to checking $(c(i) \leq 1) \wedge (c(i) \geq 1)$.

2. *Precalculated Sets based on $\mathcal{D}$:* Since our $\mathcal{D}$ is fixed, we can precalculate the following sets for every predicate $p \in \mathcal{P}$ and thus have them be plugged into our program whenever we want:

   - $TP_p$: The subset of action names along with placeholder object arguments $\alpha$ in $\mathfrak{A}$ that have $p$ in their add-effects $(\text{eff}(\alpha)^+)$.
   - $FP_p$: The set of action names along with placeholder object arguments $\alpha \in \mathfrak{A}$ that have $p$ in their delete-effects $(\text{eff}(\alpha)^-)$.
   - $Pre_p$: The set of action names along with placeholder arguments $\alpha \in \mathfrak{A}$ with $p$ in their preconditions.

3. *Check presence of Action/Predicate:* To check the presence of a specific action/ predicate, we can use our local function from $j$.
$$\text{Curr}_\alpha(i) = \# \, [j \leq i, i = j + arity(\alpha)] \; Q_{n_\alpha}(j) \geq 1$$
$$\text{Curr}_p(i) = \# \, [j \leq i, i = j + arity(p)] \; Q_{n_p}(j) \geq 1$$

   We can still use $Q$ here as the number of predicates and action names are taken from our fixed alphabet.

4. *Match predicate:* We have to determine if two different subsequences refer to the same ground proposition, and since our object universe is variable, object names cannot be directly compared against a fixed list. Thus, we compare them exclusively through their positions relative to the object names.

   Consider a mapping function $\mu_{\alpha,\beta,p} : \{1, \ldots, k_\alpha\} \to \{1, \ldots, k_\beta\}$, where $\alpha \in \mathfrak{A}$ has $p \in \mathcal{P}$ listed as a precondition and $\beta \in \mathfrak{A}$ has $p \in \mathcal{P}$ listed in its effects. So $\mu_{\alpha,\beta,p}(m) = n$ implies that we want to map the $m$-th argument of $\alpha$ with the $n$-th argument of $\beta$ to check how each of these map their arguments to $p$. We need this, as the mapping of the arguments of the underlying predicate to $\alpha$ and $\beta$ could be different.

   We define the match predicate $\chi_{\alpha,\beta,p}(i, j)$ to be true if the ground action $n_\alpha, o_1, \ldots, o_{k_\alpha}$ ending at index $i$ and action $n_\beta, o_1, \ldots, o_{k_\beta}$ ending at index $j$ match on the arguments of $p$. Since we only have access to the full argument list at the *end* of the action span due to causal masking, we define the offsets relative to $i$ and $j$:

$$\chi_{\alpha,\beta,p}(i, j) := \text{Curr}_\alpha(i) \wedge \text{Curr}_\beta(j) \wedge \bigwedge_{m=1}^{\text{arity}(p)} (c_{i-k_\alpha+m} = c_{j-k_\beta+n})$$

   where $n = \mu_{\alpha,\beta,p}(m)$. We note that all such offsets $m, n$ pairs will be constant and fixed according to a domain and can thus be plugged into our **C\*-RASP**[Pos] program. We can similarly define $\chi_{\alpha,\text{Init},p}(i, j)$ to be true, if the ground action $n_\alpha, o_1, \ldots, o_{k_\alpha}$ ending at $i$ with a ground proposition $n_p, o_1, \ldots, o_{\text{arity}(p)}$ ending at position $j$ is listed in the initial state.

$$\chi_{\alpha,\text{Init},p}(i, j) := \text{Curr}_\alpha(i) \wedge \text{Curr}_p(j) \wedge \bigwedge_{m=1}^{\text{arity}(p)} (c_{i-k_\alpha+m} = c_{j-k_p+n})$$

   With such match predicates, we will be able to match grounded actions and propositions, irrespective of the exact identities of the underlying objects. Thus, we will henceforth in the proof, be able to directly reason and interact with grounded actions and propositions listed in our $w$. We will be using $\alpha, p$ in our indexing, but those **C\*-RASP** variables will be true for every ground instance of the corresponding $\alpha, p$.

5. *Precondition Check at every action:* For a specific action instance $\alpha$ at $i$ and a specific precondition $p$ of $\alpha$, we compute three values:

   (a) **Provided by Init:** Does the initial state contain the required ground proposition?

   $$V_{init,\alpha,p}(i) := \#\,[j < i]\ (\text{InInit}(j) \wedge \chi_{\alpha,\text{Init},p}(i,j))$$

   The check for being in initial is required, so as to not match with propositions listed in $G$.

   (b) **Provided by Plan (Adds):** How many times has this proposition been added by previous actions?

   $$V_{add,\alpha,p}(i) := \sum_{\beta \in TP_p} \#\,[j < i, \chi_{\alpha,\beta,p}(i,j)]$$

   (c) **Removed by Plan (Deletes):** How many times has this proposition been deleted?

   $$V_{del,\alpha,p}(i) := \sum_{\beta \in FP_p} \#\,[j < i, \chi_{\alpha,\beta,p}(i,j)]$$

We combine these to check if precondition $p$ is satisfied for action $\alpha$ at step $i$:

   • **Case A: Delete-Free.** In a delete-free domain, $FP_p = \emptyset$, so $V_{del}(i) = 0$. The precondition is satisfied if the proposition was ever established:

   $$\text{Satisfied}_{\alpha,p}(i) := (V_{init,\alpha,p}(i) + V_{add,\alpha,p}(i) \geq 1)$$

   • **Case B: Well-Formed.** In a well-formed domain, the truth value of a proposition toggles cleanly. A proposition is true if it was in Init and not net-deleted, or not in Init and net-added. The current truth value is:

   $$\text{CurrentVal}_{\alpha,p}(i) := V_{init,\alpha,p}(i) + V_{add,\alpha,p}(i) - V_{del,\alpha,p}(i)$$

   Since the domain is well-formed, this sum will always be either 0 (False) or 1 (True). Thus:

   $$\text{Satisfied}_{\alpha,p}(i) := (\text{CurrentVal}_{\alpha,p}(i) == 1)$$

   We note that for negative precondition requirements, the satisfaction check above would be changed to the value to be equal to 0. Whether a precondition requirement is positive or negative is known apriori as well, and hence one can aptly set 0 (for negative precondition) or 1 (positive precondition) here.

We mark an action as valid, if all the preconditions are satisfied.

$$\text{Valid}_{\alpha}(i) = \bigwedge_{p \in \text{pre}(\alpha)} \text{Satisfied}_{\alpha,p}(i)$$

We check at the final token of the entire input sequence, that we are indeed at the end and that we found no invalid actions so far.

$$\text{AllActionsValid}(i) := \text{End}(i) \wedge \bigwedge_{\alpha \in \mathcal{A}} (\#\,[j \leq i]\ \neg\text{Valid}_{\alpha}(j) == 0)$$

6. Finally, we check if all goal conditions are met. We iterate over all predicates $p \in \mathcal{P}$. For each predicate $p$, we perform validity checks at every position in the block where the goal proposition is of the type $p$ (each such proposition may have different objects).

   We can construct a similar match predicate as before, $\chi_{\text{Goal},\beta,p}(i,j)$ to match a goal requirement at $i$ with an action $\beta$ at $j$ (where $j \leq i$). We can also construct $\chi_{\text{Goal},\text{Init},p}(i,j)$, here in fact the entire proposition to be matched will be listed in the same way in the goals as in the initial state.

   For a specific goal proposition at index $i$ (the index denotes the end of that goal proposition), we compute its validity:

   (a) **Initial Status:** $G_{init,p}(i) := \#\,[j \leq i, \chi_{\text{Goal},\text{Init},p}(i,j)]$
   (b) **Added:** $G_{add,p}(i) := \sum_{\beta \in TP_p} \#\,[j \leq i, \text{Curr}_{\beta}(j) \wedge \chi_{\text{Goal},\beta,p}(i,j)]$

(c) **Deletes:** $G_{del,p}(i) := \sum_{\beta \in FP_p} \# [j \leq i, \mathrm{Curr}_\beta(j) \wedge \chi_{\mathrm{Goal},\beta,p}(i,j)]$

The goal ending at $i$ is satisfied if:

$$\mathrm{GoalSat}_p(i) := (G_{init,p}(i) + G_{add,p}(i) - G_{del,p}(i) == 1)$$

(Or $\geq 1$ for Delete-Free). If $i$ is not a goal token for $p$, we define $\mathrm{GoalSat}_p(i)$ to be by default true (or ignore it in the final count). We note that for negative goal conditions, the conditions for satisfaction above would be flipped to equating with 0 (for both well-formed and delete-free). We explicitly mark goals that were unsatisfied:

$$\mathrm{UnsatisfiedGoal}(i) := \mathrm{InGoal}(i) \wedge \bigvee_{p \in \mathcal{P}} (\mathrm{Curr}_p(i) \wedge \neg\mathrm{GoalSat}_p(i))$$

At the final token, we verify that the total count of unsatisfied goal tokens is zero.

$$\mathrm{AllGoalsMet}(i) := \mathrm{End}(i) \wedge (\# [j \leq i]\ \mathrm{UnsatisfiedGoal}(j) == 0)$$

The final program output is $\Phi_{valid} := \mathrm{AllActionsValid} \wedge \mathrm{AllGoalsMet}$.

$\square$

We next proceed to proving Statement 2

*Proof.* Statements 2, 3 of Theorem 3.1 already showed that even the set of objects is fixed, universe domains with STRIPS and domains with conditional effects have instances where the language of valid plans is isomorphic to the FlipFlop and Parity language. As Theorem 3.5 showed both FlipFlop and Parity cannot be expressed in **C\*-RASP**. Thus, the current statement falls out as a corollary based on these facts established already. $\square$

# C. Learning Framework

## C.1. Model of the transformer with Extended Alphabet

**Parameterization**   Our parameterization and setup closely follows Huang et al. (2025b), with the crucial distinction being in the way we allow an extended alphabet (to account for the increase of objects during test). We focus on transformers with causal masking. We assume the transformer $T$ is parameterized by the following;

- A finite alphabet $\Sigma$ and a token embedding matrix for the finite alphabet $\boldsymbol{E} \in \mathbb{R}^{|\Sigma| \times d}$.

- Width of the transformer being $d \in \mathbb{N}$.

- A context width $N(T) \in \mathbb{N} \cup \{+\infty\}$.

- Positional encodings $\{\boldsymbol{p}_t \in \mathbb{R}^d : 1 \leq t < N(T) + 1\}$.

- An extended embedding collection $\{\mathbf{c}_k \in \mathbb{R}^d : 1 \leq k < N(T) + 1 \in \mathbb{N}\}$ representing the learnable embeddings for the extended alphabet. Note that while indexed by integers similar to positions, these are distinct parameters from the positional encodings.

- Depth $L$, heads $H$, and layer matrices $\{\boldsymbol{K}_{l,h}, \boldsymbol{Q}_{l,h}, \boldsymbol{V}_{l,h}, \boldsymbol{A}_l, \boldsymbol{B}_l, \boldsymbol{b}_l\}$ as standard.

- An unembedding matrix $\boldsymbol{U} \in \mathbb{R}^{|\Omega| \times d}$

**Input Structure, Embeddings & Positional Embedding**   We define the vocabulary as the disjoint union $\Omega = \Sigma \cup \mathcal{C} \cup \{\$\}$, where $\$$ is a special reserved start symbol. Thus, a input string $w$ is a sequence $w = (w_1, \ldots, w_n)$, where the first token is the special start symbol, $w_1 = \$$, and the rest of the sequences can be either a $\sigma \in \Sigma$ or a symbol $c \in \mathcal{C}$. Similar to Huang et al. (2025b), we study NoPE (No Positional Encoding) and Absolute Positional Encodings (APE) that has learned per-position embedding vectors $\boldsymbol{p}_1, \ldots, \boldsymbol{p}_N$. We encode an input $x$ of length $|x| = k \leq N$ using positional encodings $\boldsymbol{p}_{1+o}, \ldots, \boldsymbol{p}_{k+o}$ where $o$ is an offset such that $k + o \leq N$, and require that the transformer correctly performs the task independently of the offset $o \geq 0$. The offsets try to mimic the fact that language models typically need to solve tasks appearing at aribitrary positions in a long context. Positions outside of the input are considered empty.

**Computation of Initial State**  Let $w$ be an input sequence of length $n$ such that $n \leq N(T)$. We consider two kinds of offset again, where $\{o_p, o_c\} \geq 0$ are the positional offsets and the character offsets such that $n + o_p \leq N(T)$ and $n + o_c \leq N(T)$ as well (ensuring the sequence fits within the context window when shifted).

The input to the first transformer layer, denoted as $\mathbf{y}_i^{(0)} \in \mathbb{R}^d$ for the token at position $i$ (where $1 \leq i \leq n$), is defined as the sum of the token's embedding and its positional encoding:

$$
\mathbf{y}_i^{(0)} = \begin{cases} \mathbf{E}_{w_i} + \mathbf{p}_{i+o_p} & \text{if } w_i \in \Sigma \quad \text{(Finite Token)} \\ \mathbf{c}_{k+o_c} + \mathbf{p}_{i+o_p} & \text{if } w_i \in \mathcal{C} \text{ and } w_i \text{ has value } k \quad \text{(Extended Token)} \end{cases}
\tag{1}
$$

The motivation for training the extended alphabet with offsets is to enable reasoning over the increasing alphabet and still be able to fit everything in the same context window, akin to the motivation of using offsets for absolute positional encodings.

Attention logits, at query position $i$ and key position $j$ are computed as

$$
a_{i,j}^{(l,h)} = (\boldsymbol{y}_j^{(l-1)})^T \boldsymbol{K}_{l,h}^T \boldsymbol{Q}_{l,h} \boldsymbol{y}_i^{(l-1)} \quad \text{for } 1 \leq j \leq i \leq |x|; \; l = 1, \ldots, L; \; h = 1, \ldots, H
\tag{2}
$$

We assume standard softmax attention, but incorporate scaling with $\log|x|$ following prior work finding it necessary to theoretically represent sparse functions and circumvent theoretical limitations of soft attention (Chiang & Cholak, 2022; Edelman et al., 2022):

$$
\boldsymbol{Y}_i^{(l)} := \boldsymbol{y}_i^{(l-1)} + \sum_{h=1}^{H} \frac{\sum_{j=1}^{i} \exp\left(\log|x| \cdot a_{i,j}^{(l,h)}\right) \boldsymbol{V}_{l,h} \boldsymbol{y}_j^{(l-1)}}{\sum_{j=1}^{i} \exp\left(\log|x| \cdot a_{i,j}^{(l,h)}\right)}
\tag{3}
$$

After each attention block, the activations are passed through a one-layer MLP:

$$
\boldsymbol{y}_i^{(l)} := \boldsymbol{Y}_i^{(l)} + \boldsymbol{B}_l \cdot \psi_l(\boldsymbol{A}_l \boldsymbol{Y}_i^{(l)} + \boldsymbol{b}_l)
\tag{4}
$$

where we allow similar to Huang et al. (2025b) the activation function $\psi_l$ to be, in each coordinate, either ReLU or Heaviside. In addition, we also allow the activation function $f(x) = 1/x + \epsilon$, where $\epsilon$ is a small constant to avoid division by 0.

*Remark* C.1 (Use of $f(x) = 1/x + \epsilon$ as an activation function).  We do this to use to have an MLP that could invert values akin to the strategy in Theorem 1 of Kazemnejad et al. (2023). Owing to the universal approximation theorem (Cybenko, 1989), such non-linear activations can be approximated arbitrarily closely by ReLU MLPs. In the idealized learning procedure of Huang et al. (2025b), as well as in the idealized framework we will introduce, Transformers have widths that depend on the context window, as extended alphabets/larger positional embeddings require increasing widths. In principle, such wider transformers can represent non-standard activations (which we use in our proofs) up to the required precision, but in reality such functions can only be approximated for any fixed width. The prevalent use of the Universal Approximation theorem notwithstanding, precisely understanding how such functions are learned on the basis of more commonly used activation functions such as ReLU in MLPs *inside transformers* is largely an open and highly challenging question, which we leave to future work.

To keep our model close to Huang et al. (2025b), we also omit layer norm as they do and also assume an infinite-precision setup for the activations, with the restriction that attention logits (2) and the output of the $\exp(\cdot)$ function are both rounded to $p$ fractional bits of precision before further processing. This mild restriction was put in Huang et al. (2025b) to prevent tiny changes in attention patterns to potentially snowball into large changes in the output due to infinite precision; it is also adopted in Izzo et al. (2026).

A transformer $T$ maps strings $x$ ($|x| \leq N(T)$) to vectors of next-token prediction logits, $T(x, o_p, o_c) \in \mathbb{R}^{|x| \times |\Omega|}$, where $T(x, o_p, o_c)_i = \boldsymbol{U} \boldsymbol{y}_i^{(L)}$ ($i = 1, \ldots, |x|$) for the unembedding matrix $\boldsymbol{U} \in \mathbb{R}^{|\Omega| \times d}$, and $o_p, o_c$ are the offsets. Let $\mathcal{F}(\Omega)$ be the set of all maps $f$ mapping $x$ to $f(x) \in \mathbb{R}^{|x| \times |\Omega|}$.

### C.1.1. PRODUCT FUNCTIONS

Our theory extends the framework for length generalization of transformers with absolute positional encodings, where the width may grow with the input length, to account for cases where transformers also reason based on token identities from an unbounded vocabulary. Similar to Huang et al. (2025b), we cannot view the ground-truth function as realized by a

single transformer. Even if one assigned such a transformer an infinite number of positional encodings, it would effectively only distinguish between a bounded number of positions because the width of the model is bounded. Analogously, even if we equip the transformer with an increasing number of token embeddings, a single finite-width transformer can only act on this infinite alphabet in a limited fashion—primarily by matching identities of token symbols rather than memorizing infinite arbitrary features. Thus, just as Huang et al. (2025b) derived a parameterization to convert sequences of transformers operating on longer sequences into a single limiting object, we derive a similar parameterization to unify sequences of transformers equipped with *increasing alphabet sizes*.

We reuse the key technical idea to reparameterize the transformer in terms of **product functions**—inner products of parameter vectors mediated by parameter matrices. However, in our setting, we explicitly distinguish between pairwise interactions ($\alpha$) and single-site potentials ($\beta$).

The pairwise interactions ($\alpha$) capture operations operations where two kinds of embedding interact (either position-position, position-token, or token-token).For instance in attention scores or with the use of Input-Output predictions (Token-Token):

$$
\begin{aligned}
&\boldsymbol{p}_i^T \boldsymbol{K}_{1,h}^T \boldsymbol{Q}_{1,h} \boldsymbol{p}_j \qquad && \boldsymbol{E}_\sigma^T \boldsymbol{K}_{1,h}^T \boldsymbol{Q}_{1,h} \boldsymbol{E}_\tau \\
&\boldsymbol{p}_i^T \boldsymbol{K}_{2,h}^T \boldsymbol{Q}_{2,h} \boldsymbol{V}_1 \boldsymbol{p}_j \qquad && \boldsymbol{U}_\tau^T \boldsymbol{V}_3 \boldsymbol{V}_1 \boldsymbol{E}_\sigma
\end{aligned}
\tag{5}
$$

Meanwhile, the single-site potentials ($\beta$) capture interactions where such two kinds of embeddings do not interact. For instance while considering MLP operations:

$$
\begin{aligned}
&(\boldsymbol{A}_1)_{s,\cdot}^T \boldsymbol{p}_i \qquad && (\boldsymbol{A}_1)_{s,\cdot}^T \boldsymbol{E}_\sigma \\
&\boldsymbol{U}_\tau^T (\boldsymbol{B}_1)_{\cdot,s} \qquad && \boldsymbol{U}_\tau^T \boldsymbol{V}_2 (\boldsymbol{B}_1)_{\cdot,s}
\end{aligned}
\tag{6}
$$

**Definition C.2** (Product Parameterization)**.** More formally, this parameterization is defined as follows:

For $l = 1, \ldots, L,$:

$$
\begin{aligned}
\mathcal{V}_{\text{start}} &= \{\boldsymbol{p}_i : i\} \cup \{\boldsymbol{E}_\omega : \omega \in \Omega\} \\
\mathcal{VO}_l &= \{(\boldsymbol{B}_l)_{\cdot,s} : s = 1, \ldots, d\} \\
\mathcal{VI}_l &= \{(\boldsymbol{A}_l)_{s,\cdot} : s = 1, \ldots, d\} \\
\mathcal{VU} &= \{\boldsymbol{U}_\omega : \omega \in \Omega\} \\
\mathcal{VO} &= \mathcal{V}_{\text{start}} \cup \bigcup_{l=1}^{L} \mathcal{VO}_l \\
\mathcal{VI} &= \bigcup_{l=1}^{L} \mathcal{VI}_l \\
\mathcal{P} &= \{\{V_{l_1,h_1}, \ldots, V_{l_k,h_k}\} \ : \ 0 \le k \le L; \ l_1 < \cdots < l_k; \ 1 \le h_i \le H\}
\end{aligned}
$$

Given a transformer $T$, define the Pairwise Interaction Potentials ($\alpha$) and Single-Site Potentials ($\beta$):

$$\alpha^{\text{attn}}_{l,h,\mathcal{S}_1,\mathcal{S}_2,\boldsymbol{v},\boldsymbol{w}} := \boldsymbol{v}^T \left( \prod_{S \in \mathcal{S}_1} S \right)^T \boldsymbol{K}^T_{l,h} \boldsymbol{Q}_{l,h} \left( \prod_{S \in \mathcal{S}_2} S \right) \boldsymbol{w} \in \mathbb{R}$$

$$\text{for } 1 \le l \le L; \ \ 1 \le h \le H; \ \ \boldsymbol{v},\boldsymbol{w} \in \mathcal{VO}; \ \ \mathcal{S}_1, \mathcal{S}_2 \in \mathcal{P}$$

$$\alpha^{\text{unembed}}_{\mathcal{S},\boldsymbol{u},\boldsymbol{w}} := \boldsymbol{u}^T \left( \prod_{S \in \mathcal{S}} S \right) \boldsymbol{w} \in \mathbb{R}$$

$$\text{for } \boldsymbol{u} \in \mathcal{VU}; \ \ \boldsymbol{w} \in \mathcal{V}_{\text{start}}; \ \ \mathcal{S} \in \mathcal{P}$$

$$\beta^{\text{mlp}}_{\mathcal{S},\boldsymbol{v},\boldsymbol{w}} := \boldsymbol{v}^T \left( \prod_{S \in \mathcal{S}} S \right) \boldsymbol{w} \in \mathbb{R}$$

$$\text{for } \boldsymbol{v} \in \mathcal{VI}; \ \ \boldsymbol{w} \in \mathcal{VO}; \ \ \mathcal{S} \in \mathcal{P}$$

$$\beta^{\text{unembed}}_{\mathcal{S},\boldsymbol{u},\boldsymbol{w}} := \boldsymbol{u}^T \left( \prod_{S \in \mathcal{S}} S \right) \boldsymbol{w} \in \mathbb{R}$$

$$\text{for } \boldsymbol{u} \in \mathcal{VU}; \ \ \boldsymbol{w} \in \mathcal{VI}; \ \ \mathcal{S} \in \mathcal{P}$$

where the matrix product over a set $\mathcal{S} \in \mathcal{P}$

$$\prod_{S \in \mathcal{S}} S \tag{7}$$

is computed in descending order of layers; with the $S$ associated with the lowest layer at the right. For instance,

$$\prod_{S \in \{V_{1,h}, V_{3,h'}, V_{4,h''}\}} S = V_{4,h''} V_{3,h'} V_{1,h} \tag{8}$$

*Remark* C.3. Here, we exemplify the Product Parameterization.

$$\alpha^{\text{attn}}_{1,h,\emptyset,\emptyset,\boldsymbol{p}_i,\boldsymbol{E}_\sigma} = \boldsymbol{p}_i^T \boldsymbol{K}^T_{1,h} \boldsymbol{Q}_{1,h} \boldsymbol{E}_\sigma$$

$$\alpha^{\text{attn}}_{2,h,\{\boldsymbol{V}_{1,h'}\},\emptyset,\boldsymbol{p}_i,\boldsymbol{p}_j} = \boldsymbol{p}_i^T \boldsymbol{V}^T_{1,h'} \boldsymbol{K}^T_{2,h} \boldsymbol{Q}_{2,h} \boldsymbol{p}_j$$

$$\alpha^{\text{unembed}}_{\{\boldsymbol{V}_{3,h'},\boldsymbol{V}_{1,h}\},\boldsymbol{U}_\tau,\boldsymbol{E}_\sigma} = \boldsymbol{U}^T_\tau \boldsymbol{V}_{3,h'} \boldsymbol{V}_{1,h} \boldsymbol{E}_\sigma$$

$$\beta^{\text{mlp}}_{\emptyset,(\boldsymbol{A}_1)_{s,\cdot},\boldsymbol{p}_i} = (\boldsymbol{A}_1)^T_{s,\cdot} \boldsymbol{p}_i$$

$$\beta^{\text{mlp}}_{\{\boldsymbol{V}_{1,h}\},(\boldsymbol{A}_3)_{s,\cdot},\boldsymbol{E}_\sigma} = (\boldsymbol{A}_3)^T_{s,\cdot} \boldsymbol{V}_{1,h} \boldsymbol{E}_\sigma$$

$$\beta^{\text{unembed}}_{\{\boldsymbol{V}_{3,h'},\boldsymbol{V}_{2,h}\},\boldsymbol{U}_\tau,(\boldsymbol{B}_1)_{\cdot,s}} = \boldsymbol{U}^T_\tau \boldsymbol{V}_{3,h'} \boldsymbol{V}_{2,h} (\boldsymbol{B}_1)_{\cdot,s}$$

*Remark* C.4. For ease of notation, we have not restricted the layers from which different vector parameters are taken in the definition of $\alpha$ and $\beta$; hence, they will also include products that are not relevant to actual computations, such as

$$\boldsymbol{p}_i^T \boldsymbol{V}^T_{2,h} \boldsymbol{K}^T_{1,h'} \boldsymbol{Q}_{1,h''} \boldsymbol{V}_{3,h'''} \boldsymbol{p}_j \tag{9}$$

where a vector of the form $V_{3,h'''} \boldsymbol{p}_j$ cannot actually feed into the computation of queries in the first layer. This is simply for simplicity of notation; such products will not impact results and is consistent with the parameterization used in Huang et al. (2025b).

### C.1.2. SYMBOLIC LIMIT TRANSFORMERS

We define locality and translation invariance by generalizing the notions from (Huang et al., 2025b) beyond positional encodings to also cover token embeddings:

**Definition C.5** (Locality & Translation Invariance). For any embedding types $\mathbf{x}, \mathbf{y} \in \{\boldsymbol{p}, \boldsymbol{c}\}$, if the product functions obey the following, then the transformer is said to be local and translation invariant:

1. **Locality & Translation Invariance for Related Types:** When inputs share a type (Pos-Pos or Tok-Tok), then their interactions vanish if the distance exceeds a bandwidth $\Delta \in \mathbb{N}$ [9]:

$$|k - m| > \Delta \implies \alpha_{...,\mathbf{x}_k,\mathbf{x}_m} = 0 \tag{10}$$

Translation Invariance implies that for all shifts $\delta$, the behaviour remains the same:

$$\alpha_{...,\mathbf{x}_k,\mathbf{x}_m} = \alpha_{...,\mathbf{x}_{k+\delta},\mathbf{x}_{m+\delta}} \tag{11}$$

2. **Translation Invariance for Unrelated Types:** When inputs differ in type (Pos-Tok) or act singly (MLP-Pos), the interaction cannot pick out specific absolute indices from a structure it is unrelated to. Thus, the interaction collapses to a **Constant**. For any $\mathbf{x}_k$, if the other term is of a different type or fixed:

$$\beta_{...,\mathbf{x}_k} = C \quad \text{and} \quad \alpha_{...,\mathbf{x}_k,\mathbf{y}_m} = C \tag{12}$$

We will use the parameterization just defined to translate sequences $T_1, T_2, T_3, \ldots$ of transformers running on inputs of length $1, 2, 3, \ldots$ to limiting transformer-like objects that are applicable at all input lengths, while keeping width $d$ bounded even if the widths of $T_n$, and thus the allowed number of tokens diverge to infinity.

We will define this limit transformer like object as follows:

**Definition C.6.** A *Symbolic Limit Transformer* is a transformer $T$ where:

1. $N(T) = +\infty$

2. Positional encodings $\{\boldsymbol{p}_t\}_{t \in \mathbb{N}}$ and extended alphabet embeddings $\{\boldsymbol{c}_k\}_{k \in \mathcal{C}}$ have globally bounded norms in $\mathcal{H}$. There exists $C_{\text{emb}} > 0$ such that:
$$\sup_{t \in \mathbb{N}} \|\boldsymbol{p}_t\| \leq C_{\text{emb}} \quad \text{and} \quad \sup_{k \in \mathcal{C}} \|\boldsymbol{c}_k\| \leq C_{\text{emb}} \tag{13}$$

3. All weight matrices $\boldsymbol{W} \in \{\boldsymbol{K}_{l,h}, \boldsymbol{Q}_{l,h}, \boldsymbol{V}_{l,h}, \boldsymbol{A}_l, \boldsymbol{B}_l, \boldsymbol{U}\}$ are bounded linear operators on $\mathcal{H}$, where their spectral norms are bounded.

4. For every layer $l$, the MLP projects to a finite-dimensional subspace. There exists $d_{\text{ff}} \in \mathbb{N}$ such that $\boldsymbol{A}_l : \mathcal{H} \to \mathbb{R}^{d_{\text{ff}}}$ and $\boldsymbol{B}_l : \mathbb{R}^{d_{\text{ff}}} \to \mathcal{H}$.

5. The behavior of this transformer is governed by the set of scalar product functions defined earlier (Definition C.2). Each product function must evaluate to a number in $p$-bit precision, for some fixed $p \in \mathbb{N}$. Crucially, these functions must satisfy the **Locality and Translation Invariance Constraints** defined in Definition C.5

Length generalization will be linked to expressibility by Symbolic Limit Transformers. A Limit Transformer, as defined in Huang et al. (2025b) could use positional information through bounded-width and bounded-precision positional encodings $\boldsymbol{p}_i$, and additionally through potentially more complicated functions $\phi_{l,h}$, where a function $f : \mathbb{N} \times \mathbb{N} \to \mathbb{R}$ would be "translation-invariant" if $f(i,j) = f(i+\tau, j+\tau), \forall i \leq j, \forall \tau \geq 0$, and "local" if there is $\tau$ such that $f(i,j) = 0$ when $j > i + \tau$.

Symbolic Limit Transformers use positional information in the same way. However since now they have access to an infinite character vocabulary, we put some restrictions in the ways in which these character tokens can interact with each other. Thus we can perform operations like checking the equality of character tokens, or check the equality of character tokens that lie near each other in their embedding spaces. The other points in the definition are simply to keep the outputs of each of the operations be finite, and representable in finite precision.d

The parameterization in terms of inner products permits a translation from a transformer $T$ to a bounded-width Symbolic Limit Transformer (Statement 3 of Proposition C.13 in the Appendix).

---

[9] $\ldots$ implies that the rest of the parameters are the same on both sides

## C.2. Definition of Inference Procedure

To define the inference procedure, we specify the following hypothesis class at each input length $n$:

**Definition C.7** (Hypothesis Class). For each $n = 1, 2, 3, \ldots$, define the hypothesis class $\Theta_n$ as the set of transformers $T$ (as defined in Section C.1) where (1) $N(T) = n$, (2) each parameter vector and matrix of $T$ is represented at $p$ bits of precision, for some $p \in \mathbb{N}$, (3) each product function involving only positional encodings is translation-invariant. (4) each product function involving only the extended character alphabet is also translation-invariant.

Note that the width $d$ of the transformers $T \in \Theta_n$ is unconstrained. We keep the requirement of Huang et al. (2025b) of wanting the contributions of positional encodings $\boldsymbol{p}_i$ (that vary with position) to the transformer's computations to be offset-independent. This is a stronger requirement than for the input-output behavior to be offset-independent: we ask for the transformer's "algorithm" itself to be the same across offsets. We extend this logic to the character alphabet in requirement (4). Here, the goal is to prevent the model from overfitting to specific absolute values of the extended tokens. By enforcing translation invariance on the embeddings $\boldsymbol{c}_k$, we ensure that the model only ever uses *relative* differences between token indices (to do operations like local matching or exact equality checks). This effectively restricts the transformer to learning algorithms based on pattern matching—such as identifying when a current token matches one from the past—without allowing it to attach arbitrary semantics to specific tokens in the infinite sequence. Our inference procedure will use a regularizer $\mathcal{R}$ favoring simpler hypotheses. It should be noted that while our extension of Limit Transformers is more complex than in Huang et al. (2025b), the regularizer we will use would be simpler. It should be noted that this trades off a complexity in a theoretical mathematical construct – that is just a tool for proving statements about standard transformers – with an increase in simplicity in the conditions of the regularizer we use in the idealized model of learning. The following will be sufficient:

**Definition C.8** (Regularizer). Let $T \in \Theta_n$, thus $N(T) = n$. Define $\mathcal{R}(T)$ as the sum of (1) $L + H$; (2) the precision $p$ used in Definition C.7; the precision $p$ used for rounding attention logits and the output of $\exp(\cdot)$ (Section C.1); (3) $\max_{l,h} \|\boldsymbol{K}_{l,h}^T \boldsymbol{Q}_{l,h}\|$; $\max_{l,h} \|\boldsymbol{V}_{l,h}\|$; $\max_l \|\boldsymbol{A}_l\|_F$, $\|\boldsymbol{B}_l\|_F$; $\|\boldsymbol{U}\|$; (4) $\max_i \|\boldsymbol{p}_i\|_2$, $\max_\sigma \|\boldsymbol{E}_\sigma\|_2$, $\max_l \|\boldsymbol{b}_l\|_2$; (5) the term:

$$\zeta(T) = \sum_{l=1}^{L} \sum_{h=1}^{H} \sum_{\mathcal{S}_1, \mathcal{S}_2 \in \mathcal{P}} \left( \sum_{j=1}^{N(T)} \left| \alpha_{l,h,\mathcal{S}_1,\mathcal{S}_2,\boldsymbol{p}_1,\boldsymbol{p}_j}^{\text{Attn}} \right|^2 + \sum_{k=1}^{|\Omega|} \left| \alpha_{l,h,\mathcal{S}_1,\mathcal{S}_2,\boldsymbol{E}_1,\boldsymbol{E}_k}^{\text{Attn}} \right|^2 \right) + \sum_{\mathcal{S} \in \mathcal{P}} \sum_{k=1}^{|\Omega|} \left| \alpha_{\mathcal{S},\boldsymbol{U}_1,\boldsymbol{E}_k}^{\text{unembed}} \right|^2 \tag{14}$$

We note that this regularizer omits a term $rank(\boldsymbol{V}_{l,h})$ used in Huang et al. (2025b); we do not need it and can thus simplify by removing it.

As in Huang et al. (2025b), the idea of (14) is to discourage accidental attention between far-away positions and interactions between character token embeddings that do not appear together during training, which could hamper length generalization. These form a subset of the product functions formally defined earlier (Definition C.2). Due to translation invariance, such a term entails a bound on products for all pairs $\boldsymbol{p}_i, \boldsymbol{p}_j$ ($i \leq j$) and $\boldsymbol{c}_i, \boldsymbol{c}_j$ entering causal attention. While such a regularizer is not part of standard training, standard initialization tends to lead to bounded values for (14) when $d$ is large. As shown in Huang et al. (2025b), it thus captures an implicit bias of standard initialization and training. Additionally, by removing the bounded rank constraint on the regularizer, we enforce lesser restrictions on it, making it a more plausible estimate of standard initialization and training compared to what was used in Huang et al. (2025b). Importantly, the width $d$ does not explicitly enter $\mathcal{R}$; as a consequence, for any sufficiently large $C$, the number of transformers $T_n \in \Theta_n$ with $\mathcal{R}(T_n) \leq C$ is infinite, simply because $d$ is not constrained. Nonetheless, this regularizer will be sufficient for identification under our idealized inference procedure, which observes the input-output behavior of the target function $f$ on inputs of length $\leq \frac{n}{2}$ and selects a transformer $T$ with maximal context window $n$, $T \in \Theta_n$ that exactly fits that input-output behavior while minimizing the regularizer $\mathcal{R}(T)$. At inference time, when the transformer $T_n$ processes input lengths up to $n$, it may encounter extended tokens $\boldsymbol{c}_k$ that were effectively unseen during the "training" phase (which is restricted to smaller indices). This can be thought of as adding random embeddings at test time. However, these tokens would be constrained by the translation invariance and locality conditions of the hypothesis class $\Theta_n$, and the algorithm learned by the transformer would not be able to rely on the specific identity of these new tokens, and instead would have to use the same relative algorithms learned on the smaller alphabet (such as checking for equality $\boldsymbol{c}_i = \boldsymbol{c}_j$ or local proximity). This setup models the transformer's ability to generalize simpler algorithmic primitives (like pattern matching) to completely novel vocabulary items. The inference procedure is then defined in analogy to Huang et al. (2025b):

**Definition C.9** (Inference Procedure). Given a function $f \in \mathcal{F}(\Omega)$, the *Inference Procedure* obtains a sequence of

transformers $T_1 \in \Theta_1, T_2 \in \Theta_2, \ldots$ as follows. Define $U_n$ as the set of $T \in \Theta_n$ matching the behavior of $f$ on the restricted domains of inputs constrained in the following ways relative to $n$.

1. The sequence length of any input $x$ is $|x| \leq \frac{n}{2}$.

2. The extended character tokens $\{c_k\}$ appearing in $x$ fall within a range of size at most $\frac{n}{2}$. Specifically, if $K_x = \{k \in \mathbb{N} \wedge k \in [1, n] \mid c_k \text{ appears in } x\}$ is the set of indices in input $x$, then $\max(K_x) - \min(K_x) \leq \frac{n}{2}$.

Then choose $T_n \in U_n$ such that

$$\mathcal{R}(T_n) \leq \frac{1}{n} + \inf_{T \in U_n} \mathcal{R}(T) \tag{15}$$

Importantly, we only ask $T_n$ to match the behavior of $f$ up to length $\frac{n}{2}$ with context window sizes of the character embeddings involved also restricted to $\frac{n}{2}$, formalizing the idea of training on shorter inputs and testing on longer ones; our identifiability guarantee will provide conditions under which $T_n$ will end up matching $f$ correctly up to length $n$ – representing length generalization. In (15), we do not simply ask for minimizing the regularizer, as the set of elements of $U_n$ with $\mathcal{R}(T)$ smaller than a given value need not be finite and thus a minimum need not be attained by any $T_n$. As in (Huang et al., 2025b), we take the testing length to be twice the training length, but the analysis works whenever the training length diverges to infinity.

### C.3. Main Result: Convergence of Inference Procedure

Our main result asymptotically characterizes length generalization under the inference procedure from Definition C.9. For functions representable by Symbolic Limit Transformers, we guarantee that *any* run of the Inference Procedure will ultimately achieve length generalization, so that transformers with context length $n$ and access to a vocabulary range of $n$ from $\mathcal{C}$ chosen to fit the target function on inputs with length $\leq \frac{n}{2}$ and vocabulary range of $n/2$ from $\mathcal{C}$ will, when $n$ is sufficiently large, also perform correctly at all lengths $\leq n$. Formally, we obtain the following analogue of Theorem 7 in Huang et al. (2025b):

**Theorem C.10** (Guaranteed Length Generalization in the Limit). *Let $f \in \mathcal{F}(\Omega)$. Then the following are equivalent:*

1. *$f$ is expressible by a Symbolic Limit Transformer.*

2. *(Guaranteed Length Generalization) Applying the Inference Procedure from Definition C.9 to $f$ generates a sequence $T_1, T_2, \ldots$ with $\sup_{n=1,2,3,\ldots} \mathcal{R}(T_n) < \infty$, for which there is some $N_0$ such that, for all $m > N_0$, $T_m$ matches $f$ on all inputs of any length $k \leq m$.*

*Remark* C.11. We note that a Symbolic limit transformer $T_\infty$ representing $f$ need not itself be offset-invariant. It is sufficient to have

$$T_\infty(x, 0) = f(x) \tag{16}$$

Statement 2 of Proposition C.13 shows that such a function has a sequence of transformers $T_n \in \Theta_n$ which are offset-invariant, even without assuming $T_\infty$ to be offset-invariant.

**High-Level Proof Sketch** Our proof here tracks the proof of Theorem 7 of Huang et al. (2025b) really closely. The core logic of both ours and theirs is the same, and the difference primarily lies in the fact that our learning setting has different definitions for the regularizer, our limiting object, inference procedure etc. The core idea is that if $f$ is expressible by a Symbolic Limit Transformer satisfying then, even though the Inference Procedure produces infinitely many distinct *transformers* $T_1, T_2, \ldots$ (with increasing numbers of position and token embeddings), these can only traverse a finite set of underlying *algorithms*, each described by some Symbolic Limit Transformer. The property of translation invariance and locality ensure that the parameter count effectively remains finite, as its position-related parameters as well as the extended character related parameters can be fully specified in terms of finite product functions. The regularizer bounds the complexity of the symbolic Limit Transformers as well, thus keeping the set of algorithms traversed finite. For $1 \Rightarrow 2$, given a sequence generated by the Inference Procedure, we show that $\mathcal{R}$ stays bounded and use the compactness property to show that a subsequence exhibits behavior equivalent to $f$. To show that, in fact, *all* possible sequences $T_n$ generated by the Inference Procedure ultimately exhibit behavior equivalent to $f$, when $n$ is large, we show that subsequences failing to length-generalize would exhibit increasing attention scores between far-away positions and between characters whose embeddings are far apart, as input length increases. Due to the penalty on such scores in $\mathcal{R}$, any such sequence would, for

large $n$, need to have a higher value of $\mathcal{R}$ than sequences avoiding such an increase. For 2⇒1, we obtain the Symbolic Limit Transformer from the compactness property applied to the sequence generated by the Inference Procedure. The penalty on pairwise interactions enforces that the transformer satisfy the locality property, and as noted in the Remark C.11, offset invariance is not necessary for $T_\infty$.

**Preliminaries and Formal Proof**  For our formal proof, we will set up some definitions that will help provide translations between ordinary transformers and Symbolic Limit Transformers.

It will be useful to define a complexity metric applicable to Symbolic Limit Transformers.

**Definition C.12.** For a Limit Transformer $T_\infty$, define $\mathcal{R}_\infty(T_\infty)$ as the sum of

1. The number of layers and heads $(L + H)$

2. $d_{MLP}$

3. the precision $p$ used for expressing the output of any product function, and the precision $p$ used for rounding attention logits and the output of $\exp(\cdot)$ (Section C.1).

4. The total L2 norm of the pairwise product functions. This corresponds to the limit $(N(T) \to \infty)$ of the $\zeta(T)$ term in Definition C.8:

$$\zeta(T_\infty) = \sum_{l=1}^{L}\sum_{h=1}^{H}\sum_{\mathcal{S}_1,\mathcal{S}_2 \in \mathcal{P}} \left( \sum_{j=1}^{\infty}\left|\alpha_{l,h,\mathcal{S}_1,\mathcal{S}_2,\boldsymbol{p}_1,\boldsymbol{p}_j}^{\text{Attn}}\right|^2 + \sum_{k=1}^{\infty}\left|\alpha_{l,h,\mathcal{S}_1,\mathcal{S}_2,\boldsymbol{E}_1,\boldsymbol{E}_k}^{\text{Attn}}\right|^2 \right)$$
$$+ \sum_{\mathcal{S} \in \mathcal{P}}\sum_{k=1}^{\infty}\left|\alpha_{\mathcal{S},\boldsymbol{U}_1,\boldsymbol{E}_k}^{\text{unembed}}\right|^2$$

**Proposition C.13.** *The following are true.*

1. *Specification by Product functions: The input-output behavior of a (symbolic-limit or normal) transformer is fully specified by its product functions. Thus, for any choice of product functions that respects translation invariance and locality, there is a corresponding symbolic-limit transformer, and also (restricting to any $N(T)$) a corresponding transformer implementing these.*

2. *Symbolic Limit Transformer to a standard Transformer: For any symbolic-limit transformer $T_\infty$ and any context window length $N_0$, there is a corresponding transformer $T \in \Theta_{N_0}$ with $N(T) = N_0$ that exactly matches the product functions of $T_\infty$ restricted to inputs up to size $N_0$. Furthermore, $\mathcal{R}(T) \leq \mathcal{R}_\infty(T_\infty)$.*

3. *Standard Transformer to a Symbolic Limit Transformer: For any transformer $T$, there is a corresponding symbolic limit transformer $T_\infty$ that has the same product functions up to $N(T)$, and $\mathcal{R}_\infty(T_\infty)$ is bounded in terms of $R(T)$.*

4. *Let $C > 0$ be a constant. There is a finite set of symbolic limit transformers denoted $\mathcal{Q}_C$, such that any $T_\infty$ with $R_\infty(T_\infty) < C$ agrees with an element of $\mathcal{Q}_C$ on all product functions except possibly those representing pairwise interactions between embeddings of related types (Position-Position, Token-Token, etc.).*

*Proof.* 1. This directly follows from the definition of our Product Parameterization (Defintion C.2). Since the output of the model is a composition of scalar products, specifying those products specifies the model.

2. We can simply construct $T$ to evaluate the product functions of the Symbolic Limit Transformer for indices $i, j \leq N_0$. Since $\mathcal{R}_\infty(T_\infty)$ is the sum of squared norms over the infinite domain, and $\mathcal{R}(T)$ is the sum over the finite domain $N_0$, non-negativity of the terms involved implies $\mathcal{R}(T) \leq \mathcal{R}_\infty(T_\infty)$.

3. We construct $T_\infty$ from $T$ as follows:

- For the product functions terms involving unrelated types or involving just a single position or single character from the extended set, we extend the values observed in $T$ constantly to infinity (because of our assumption of locality).

- For **pairwise** terms (related types), since the product functions are translation invariant, we take all available distances from $T$ and use that to fill in the values of our product functions wherever the relative distance between the terms is smaller than the available distances, and set the other product functions to $0$ i.e. when the relative distance was not observed in $T$. This extension thus, also ensures the infinite sum of squares converges, keeping $\mathcal{R}_\infty(T_\infty)$ bounded by the finite sum $\mathcal{R}(T)$.

4. We focus on the product functions excluded from our exception. More specifically the single-site $\beta$ terms, or the $\alpha$ terms where Position-Token interactions happen.

By the definition of the Symbolic Limit Transformer, these functions must satisfy the Locality constraint. Thus, they must evaluate to a constant value $K$ everywhere to ensure the limit object is well-defined. Additionally, since $\mathcal{R}_\infty(T_\infty) < C$:

1. The contribution of such product functions to the regularizer would be $|K|^2$. Thus, $|K|^2 \leq \mathcal{R}_\infty(T_\infty) < C$, which implies $|K| < \sqrt{C}$.

2. Since the precision $p$ is fixed, $K$ must be a multiple of $2^{-p}$.

There are only finitely many discrete multiples of $2^{-p}$ in the bounded interval $[-\sqrt{C}, \sqrt{C}]$. Since the transformer has a fixed, finite number of layers and heads, there are only finitely many such constant product functions to assign. Thus, the set of possible configurations for these components, $\mathcal{Q}_C$, is finite.

$\square$

The following definitions will be used:

**Definition C.14.** If $T \in \Theta_i$, then define $\mathcal{R}_-(T)$ to be $\mathcal{R}(T)$ minus the term in Eq. (14). That is,

$$\mathcal{R}(T) = \mathcal{R}_-(T) + \zeta(T) \tag{17}$$

where

$$\zeta(T) = \sum_{l=1}^{L} \sum_{h=1}^{H} \sum_{\mathcal{S}_1, \mathcal{S}_2 \in \mathcal{P}} \left[ \sum_{j=1}^{N(T)} \left| \alpha_{l,h,\mathcal{S}_1,\mathcal{S}_2,\boldsymbol{p}_1,\boldsymbol{p}_j}^{\text{Attn}} \right|^2 + \sum_{k=1}^{|\Omega|} \left| \alpha_{l,h,\mathcal{S}_1,\mathcal{S}_2,\boldsymbol{E}_1,\boldsymbol{E}_k}^{\text{Attn}} \right|^2 \right] + \sum_{\mathcal{S} \in \mathcal{P}} \sum_{k=1}^{|\Omega|} \left| \alpha_{\mathcal{S},\boldsymbol{U}_1,\boldsymbol{E}_k}^{\text{unembed}} \right|^2$$

The following lemma will be used for both directions of our main theorem C.10, and is an adaptation of Lemma 17 from Huang et al. (2025b). The proof technique is the same, the main thing that changes is that the terms in the derivation refer to our framework: in paticular, the treatment of positional encodings from Huang et al. (2025b) is transferred to also apply to extended alphabet characters. We intentionally keep our notation and argumentation similar to theirs to enable quick readability for a reader familiar with Huang et al. (2025b). For readers unfamiliar with Huang et al. (2025b), the following set of proofs are still self contained, and our remarks are meant to simply acknowledge that the following proof argument style is not novel, and is simply an application of it in our context.

**Lemma C.15.** *Let $T_1, T_2, \ldots$, where $T_n \in \Theta_n$, be a sequence generated by the Inference Procedure based on the functional behavior of a function $f \in \mathcal{F}$, and such that*

$$\sup_{n=1,2,3,\ldots} \mathcal{R}(T_n) < \infty \tag{18}$$

*Then $f$ is expressible by a Symbolic Limit Transformer and there is some $N_0$ such that, for all $m > N_0$, $T_m$ matches $f$ on all inputs of length $k \leq m$.*

*Proof.* We will refer to all sets of pariwise functions $\alpha$ with related types with $\Phi_{\text{related}}$. We will use a superscript $\Phi_{\text{related}}^{(T)}$, to indicate the symbolic limit transformer whose relevant product functions of this type are being talked about.

We can use statement 3 of Proposition C.13 to get Limit Transformers $\tilde{T}_1, \tilde{T}_2, \ldots$ such that $\sup_i \mathcal{R}_\infty(\tilde{T}_i) < \infty$ for the sequence of transformers $T_1, T_2, \ldots$ generated by our Inference Procedure. In similar vein to Huang et al. (2025b), we thus get,

$$\tilde{T}_i(x, o) = T_i(x, o), \quad \forall i, o, x; |x| + o \leq i \tag{19}$$

and, each $\tilde{T}_i$ possesss the same set of pairwise product functions as $T_i$ up to length $i$, which means $\Phi_{\text{related}}^{(\tilde{T}_i)} = \Phi_{\text{related}}^{(T_i)}$

Each $\tilde{T}_i$ has (i) the collection of its constant product functions (single site potentials and cross-type interactions) (ii) the collection of pairwise decaying product functions. We will write $\mathfrak{P}(\tilde{T}_i)$ for the collection of the constant product functions $\tilde{T}_i$. Let $A := \sup_i \mathcal{R}_\infty(\tilde{T}_i) < \infty$. Then $\{\mathfrak{P}(\tilde{T}) : \mathcal{R}_\infty(\tilde{T}) \leq A\}$ and by Proposition C.13 (Statement 4) only a finite number of Symbolic Limit Transformer parameter settings $\mathfrak{P}(\tilde{T}_i)$ will be traversed as $i \to \infty$.

If each product function involved in $\Phi_{\text{related}}^{(\tilde{T}_i)}$ also traversed a finite set of distinct functions as $i \to \infty$, then the number of $\tilde{T}_i$ itself is finite would have been finite. Each of these functions are local according to definition; however, a priori, they might not be local for any single finite $\tau$ (where $\tau$ is the distance beyond which product functions evaluate to 0) across the different $\tilde{T}_i$. We will now show that all such functions are in fact local for a single finite $\tau$.

For any given $n$,

$$\mathcal{R}(T_n) \in \left[ \inf_{T \in U_n} (\mathcal{R}(T)), \frac{1}{n} + \inf_{T \in U_n} (\mathcal{R}(T)) \right]$$

Again similar to Huang et al. (2025b), here, our $\inf_{T \in U_n} \mathcal{R}(T)$ is bounded and monotonically increasing in $n$, and thus it converges to some limit, say $\tilde{R}$. As $1/n \to 0$, the interval width tends to 0, and therefore the squeeze theorem gives $\mathcal{R}(T_n) \to \tilde{R}$.

For each $\tau$ and each $n$, we consider

$$D_n(\tau) = \sum_{f \in \Phi_{\text{related}}^{(\tilde{T}_n)}} \sum_{i=1}^{\min(n,\tau)} |f^{(\tilde{T}_n)}(1,i)|^2 \leq \mathcal{R}(T_n)$$

The $f^{(\tilde{T}_n)}(1,i)$ refers to every single product function, where the interactions happen between a term $i$ and a term $1$. Therefore for positional attention terms $(p_i^T \ldots p_j)$, this implies interactions between $p_1$ and $p_i$, for extended alphabet attention terms it implies interaction between $c_1$ and $c_i$, and for unembedding terms it implies interactions between some $U_1$ and $E_i$.

Each such $f$ has precision bounded in terms of $\mathcal{R}(T_n)$, and thus $\{D_n(\tau) : n \in \mathbb{N}\}$ is a discrete set. Consequently, every accumulation point of the sequence $(D_n(\tau))_{n \in \mathbb{N}}$ must be attained for infinitely many values of $n$.

Having established this, we now look at $\mathcal{R}_-(T_n)$ from Equation 17. The following derivation (Equation 20 to Equation 26) are verbatim the same as Equations 11–17 in Huang et al. (2025b), with the key difference being in the interpretation of the terms here. Let

$$R_0 := \liminf_{n \to \infty} \mathcal{R}_-(T_n) \tag{20}$$

and let $\nu_1, \nu_2, \nu_3, \ldots$ be such that

$$\lim_{i \to \infty} \mathcal{R}_-(T_{\nu_i}) = R_0 \tag{21}$$

Then, for some $D_0$,

$$\lim_{i \to \infty} D_{\nu_i}(\nu_i) = D_0 \tag{22}$$

and

$$\tilde{R} = \lim_{n \to \infty} \mathcal{R}(T_n) = \lim_{i \to \infty} \mathcal{R}(T_{\nu_i}) = R_0 + D_0 \tag{23}$$

Indeed,

$$D_0 = \limsup_{n \to \infty} D_n(n) \tag{24}$$

because[10]

$$D_0 + R_0 = \lim_{n \to \infty} (\mathcal{R}_-(T_n) + D_n(n)) = \liminf_{n \to \infty} \mathcal{R}_-(T_n) + \limsup_{n \to \infty} D_n(n) \tag{25}$$

---

[10]As noted by Huang et al. (2025b), if $a_n + b_n$ converges and $a_n, b_n$ are bounded, then the limit $\lim(a_n + b_n)$ equals $\limsup a_n + \liminf b_n$. For, assume $\limsup a_n + \liminf b_n > \lim(a_n + b_n)$ (similar if $>$ is replaced by $<$). Then let $i(n)$ be a subsequence such that $a_{i(n)} \to \limsup a_n$. Then $\lim(a_n + b_n) = \lim(a_{i(n)} + b_{i(n)}) = \limsup a_n + \lim b_{i(n)} \geq \limsup a_n + \liminf b_n > \lim(a_n + b_n)$, contradiction.

Define, for each $\tau \in \mathbb{N}$,

$$D_\infty(\tau) = \liminf_{i \to \infty} D_{\nu_i}(\tau) \tag{26}$$

Because all functions in $\Phi_{\text{related}}$ have bounded precision in terms of $\mathcal{R}(T_n)$, and our function above is monotonically increasing, there must be $\tau_\infty$ such that $D_\infty(\tau_\infty) = \lim_{\tau \to \infty} D_\infty(\tau)$.

We now define a sequence $T'_n$ where for each $n$,

$$\liminf_{j \to \infty} D_{\nu_j}(n) = D_\infty(n) \leq D_\infty(\tau_\infty)$$

As all functions in $\Phi_{\text{related}}$ have bounded precision, there are infinitely many $\nu_i$ such that $D_{\nu_i}(n) = \liminf_{j \to \infty} D_{\nu_j}(n)$. Hence, we can select $i(n) \in \mathbb{N}$ such that $\nu_{i(n)} \geq n$ and

$$D_{\nu_{i(n)}}(n) = \liminf_{j \to \infty} D_{\nu_j}(n)$$

For each $n$, let $T'_n$ be the restriction of $T_{\nu_{i(n)}}$ to positions up to $n$. Since $T_{\nu_{i(n)}}$ agrees with $f$ up to length $\nu_{i(n)}/2 \geq n/2$, it follows that $T'_n$ also agrees with $f$ up to length $n/2$. Once again, the derivation from here till the point where we show that $D_\infty(\tau_\infty) = D_0$ are verbatim the same as [Huang et al. (2025b)](#). Then

$$\begin{aligned}
\limsup_{n \to \infty} \mathcal{R}(T'_n) &= \limsup_{n \to \infty} \mathcal{R}_-(T'_n) + D_{\nu_{i(n)}}(n) \\
&= \limsup_{n \to \infty} \mathcal{R}_-(T_{\nu_{i(n)}}) + D_\infty(\tau_\infty) \\
&= R_0 + D_\infty(\tau_\infty)
\end{aligned}$$

Since $T_n$ was created by the Inference Procedure, we have

$$\limsup_{n \to \infty} \mathcal{R}(T'_n) \geq \lim_{n \to \infty} \mathcal{R}(T_n) \tag{27}$$

On the other hand, since $\mathcal{R}(T'_n) \leq \mathcal{R}(T_{\nu_{i(n)}})$, we also have

$$\limsup_{n \to \infty} \mathcal{R}(T'_n) \leq \lim_{n \to \infty} \mathcal{R}(T_n) \tag{28}$$

giving

$$\limsup_{n \to \infty} \mathcal{R}(T'_n) = \lim_{n \to \infty} \mathcal{R}(T_n) = D_0 + R_0 \tag{29}$$

Hence,

$$\begin{aligned}
R_0 + D_\infty(\tau_\infty) &= \limsup_{n \to \infty} \mathcal{R}(T'_n) \\
&= \lim_{n \to \infty} \mathcal{R}(T_n) \\
&= R_0 + D_0
\end{aligned}$$

and $D_\infty(\tau_\infty) = D_0$. Now assume there are infinitely many $n$ such that the functions in as each function in $\Phi_{\text{related}}^{(T_n)}$ are not $\tau_\infty$-local, hence, infinitely many $n$ such that $D_n(n) \geq D_n(\tau_\infty) + 2^{-2p}$. Then:

$$\begin{aligned}
D_0 &= \limsup_{n \to \infty} D_n(n) \\
&\geq \limsup_{n \to \infty} D_n(\tau_\infty) + 2^{-2p} \\
&\geq \liminf_{i \to \infty} D_{\nu_i}(\tau_\infty) + 2^{-2p} \\
&= D_0 + 2^{-2p}
\end{aligned}$$

This is a contradiction. The first inequality follows from $D_n(n) \geq D_n(\tau_\infty)$ whenever $n \geq \tau_\infty$, as for every $n$, $D_n(\cdot)$ is monotonically increasing. The second holds because $(\nu_i)_{i \in \mathbb{N}}$ is a subsequence of $(n)_{n \in \mathbb{N}}$; hence a $\limsup$ over the larger

sequence gives an upper bound of $\lim\inf$ over the subsequence. Thus with this, we can claim that all the functions in $\Phi_{\text{related}}^{(\tilde{T}_n)}$ have to be local for a uniform $\tau_\infty$.

The remainder of the proof also follows Huang et al. (2025b); we repeat it here for self-containedness. The attention product functions involving only positions or only extended characters have products bounded by $(R_0)^4$, and are bounded in absolute value by $\leq \|\boldsymbol{p}_i\|_2 \|\boldsymbol{K}_{l,h}^T\|\|\boldsymbol{Q}_{l,h}\|\|\boldsymbol{p}_j\|_2 \leq (R_0)^4$ or $\leq \|\boldsymbol{c}_i\|_2 \|\boldsymbol{K}_{l,h}^T\|\|\boldsymbol{Q}_{l,h}\|\|\boldsymbol{c}_j\|_2 \leq (R_0)^4$. The unembedding terms are in fact bounded by a smaller polynomial $\|U_i\|\|E_j\| \leq (R_0)^2$, therefore each function value is expressed at precision bounded by a polynomial in $R_0$. There are only a finite set of functions that satisfy these properties and are are local for this $\tau_\infty$.

We had already shown that the constant components of $\tilde{T}$ are from a finite set, and now we have also shown that the pairwise product functions are also from a finite set. Hence, we know that, $\mathcal{Q} := \{\tilde{T}_i : i \in \mathbb{N}\}$ is finite. Let $\mathcal{Q}_\infty \subseteq \mathcal{Q}$ be the set of Limit Transformers that equal $\tilde{T}_i$ for infinitely many different $i$. By definition of the Inference Procedure, every element of $\mathcal{Q}_\infty$ is functionally equivalent to $f$ at all input lengths. Because $\mathcal{Q}$ is finite, there is $N_0$ such that $\tilde{T}_i \in \mathcal{Q}_\infty$ for each $i \geq N_0$. Hence, $T_i$ is functionally equivalent to $f$ at all lengths $\leq i$ as soon as $i$ exceeds the threshold $N_0$.

$\square$

We now prove the theorem.

*Proof of the Theorem.* With Lemma C.15, now both directions of the theorem are just corollaries.

**2$\Rightarrow$1:** This directly follows from Lemma C.15.

**1$\Rightarrow$2:** By Statement 2 of Proposition C.13, there exists a standard normal transformer $\widehat{T}_i \in \theta_i$ which can be derived from the ground truth Symbolic Limit Transformer $\tilde{T}_\infty$ for each $i = 1, 2, 3, \ldots$, such that $\mathcal{R}(\widehat{T}_i)$ is uniformly bounded, i.e.

$$\limsup_{i \to \infty} \mathcal{R}(T_i) \leq \limsup_{i \to \infty} \mathcal{R}(\widehat{T}_i) < \infty \tag{30}$$

where $T_i$ refers to the sequence generated by Inference Procedure in Lemma C.15. Lemma C.15 now provides a threshold $N_0 > 0$ such that the sequence of inferred transformers stablise to some Symbolic Limit Transformer, which we define as computing a function $g$. Now for all $m > N_0$, $T_m$ computes $g$ on inputs of length $\leq m$. However, by construction of the inference procedure, $T_m$ is also constrained to match the ground truth function $f$ on inputs of length $\leq m$. Therefore for all $m > N_0$, and all inputs x, such that $|x| \leq m$, we have $g(x) = T_m(x) = f(x)$. Since this holds for arbitrarily large m, we conclude that $g \equiv f$. Thus, for all $m > N_0$, the inferred transformer $T_m$ matches f.

$\square$

## C.4. Result for NoPE Transformers

As in (Huang et al., 2025b), we obtain an analogous result for NoPE transformers:

**Corollary C.16.** *For ease of the reader, we mark the differences to Theorem C.10 in green font.*

*Let $f \in \mathcal{F}(\Omega)$. Then the following are equivalent:*

1. *$f$ is expressible by a Symbolic Limit Transformer where all $\boldsymbol{p}_i \equiv 0$*

2. *(Guaranteed Length Generalization) Consider the inference procedure from Definition C.9 applied to $f$ with $\mathcal{R}$ while constraining all $\boldsymbol{p}_i \equiv 0$, generating a sequence $T_1, T_2, \ldots$. For any such sequence, there is some $N_0$ such that, for all $m > N_0$, $T_m$ matches $f$ on all inputs of any length $k \leq m$, and $\sup_{n=1,2,3,\ldots} \mathcal{R}(T_n) < \infty$.*

*Proof.* The statement 2 of Proposition C.13 still stands, but instead when translating a Symbolic Limit Transformer to an ordinary transformer, since the positional encodings are taken to be zero, $\boldsymbol{p}_i \equiv 0$, the product functions involving these positions become 0. Similarly in the statement 3 of Proposition C.13 also stands, but when $\boldsymbol{p}_i \equiv 0$ in a transformer, the resulting Symbolic Limit Transformer also has zero positional encodings and zero outputs for all associated positional product functions. Specifically, the pairwise product functions restricted to position-position interactions will be identically zero, and the single site potentials or the cross related types will have fewer product functions. The finiteness and

compactness arguments in Lemma C.15 then rely solely on the interactions between extended alphabet characters (Token-Token interactions) and the remaining constant components. Since these are a subset of the components bounded in the general proof, the convergence and finiteness arguments apply equally. The proof of Theorem C.10 thus applies directly to show Corollary C.16.

$\square$

### C.5. Statement of Main Theorem for Arbitrary Training Lengths

Analogous to how Huang et al. (2025b) extended their generalization results from length $n/2$ to $n$ to arbitrary scaling of train vs test lengths, we do the same.

**Definition C.17.** A *training length* is a function $t : \mathbb{N} \to \mathbb{N}$ satisfying $\lim_{t \to \infty} t(n) = +\infty$ and $t(n) \leq n$ for all $n$.

If $t(n)$ is a training length, then the $t(n)$-Inference Procedure determines $T_n \in \Theta(n)$ to match $f$ at all inputs of lengths $\leq t(n)$ while minimizing $\mathcal{R}(T_n)$ up to $\frac{1}{n}$.

The special case of $t(n) = \frac{n}{2}$ is the Inference Procedure from Definition C.9.

The definition above is an adaptation of Definition 22 from Huang et al. (2025b), except the inference procedure and regularizer mentioned in the definition will refer to the versions we defined earlier. With this, we can state the following.

**Theorem C.18.** *Let $f \in \mathcal{F}(\Omega)$. The following are equivalent:*

1. *$f$ is expressible by a Symbolic Limit Transformer.*

2. *Let $t(n)$ be any training length. Then the $t(n)$-Inference Procedure will output solutions $T_1, T_2, \ldots$ such that, for some $N_0$, for all $m > N_0$, $T_m$ matches $f$ at all lengths $\leq m$.*

   Intuitively, this says that, when selected to fit the behavior of $f$ on sufficiently long inputs of length $t(n)$, the output of the Inference Procedure will generalize to unboundedly longer inputs of length $n$, where $n$ can be arbitrarily larger than $t(n)$.

**Corollary C.19.** *Assume $f \in \mathcal{F}(\Omega)$ is not expressible by a Symbolic Limit Transformer. Then, for some training length $t(n)$, the $t(n)$-Inference Procedure outputs a sequence $T_n$ where infinitely many $T_n$ fail to match $f$ at length $n$.*

The proof here tracks the proof of Theorem 23, Corollary 24 & Remark 25 from Huang et al. (2025b) which are applicable directly here, with the difference being that the terms refer to our learning model. We restate it for completeness.

*Remark* C.20. Theorem C.18 differs from Theorem C.10 in the following aspects.

- Here, we talk about length generalization for all arbitrary training lengths $t(n)$, not specifically $\frac{n}{2}$.

- We do not ask for $\sup_i \mathcal{R}(T_i) < \infty$, but ask for $T_n$ to ultimately length generalize.

*Proof of Theorem C.18.* 1$\Rightarrow$2 The proof of Theorem C.10 remains valid in this direction without any changes, as we never used the fact that training length was half the context size.

2$\Rightarrow$1 We will show that if $f$ is not expressible by a Symbolic Limit Transformer, then generalization would not happen. Thus, assume $f$ is not expressible by a Symbolic Limit Transformer. We use the same arguments as in Lemma C.15. Consider any sequence $T_n \in \Theta_n$ that matches $f$ and has $\liminf_{n \to \infty} \mathcal{R}(T_n) < \infty$. This sequence need not be generated by our Inference procedure. If we translating each of these to a Symbolic Limit Transformer we once again will get a sequence where, except the pairwise product functions, only a finite number of settings will be traversed.

Now, as in the proof of Lemma C.15, we use $D_\infty(\tau)$ to construct a sequence of Symbolic Limit Transformers that are local for a single $\tau$. Now since $f$ is not expressible by a Symbolic Limit transformer, $\liminf_{n \to \infty} \mathcal{R}(T_n) = \infty$ (†) for any sequence $T_n \in \Theta_n$ that matches $f$. Because if this was not true, then by Lemma C.15, we would get a Symbolic Limit Transformer.

We will now construct a sequence of failure points $n_k$ and and a training length $t(n)$. Let $k \in \mathbb{N}$ represent a target training length. For any such fixed $k$, it is possible to match $f$ on inputs of length $\leq k$ with a finite regularization cost. Let $U_k$ be

an upper bound on this regularization cost required to fit $f$ on inputs of length $k$, considering any transformer of any size $n > k$. We set $U_k$ such that for any $n > k$, there exists a transformer in $\theta_n$, matching $f$ on the prefix $k$ with cost $\leq U_k$.

Because the cost to match $f$ at length $n$ diverges to infinity (as established above), for any fixed $k$, we can find a $n_k$ sufficiently large such that any transformer $T \in \theta_{n_k}$ that matches $f$ at the full length $n_k$ must have a regularization cost strictly greater than $U_k + 1$. We construct this sequence $n_k$ inductively ensuring $n_k > n_{k-1}$.

We now define the training length function $t(n)$ based on these intervals: $t(n) = \max\{k : n_k \leq n\}$. This function is a step function, for $n \in [n_k, n_{k+1})$, the training length is constant at $t(n) = k$. Note that $t(n) \to \infty$ as $n \to \infty$.

Consider the behavior of the Inference Procedure at specific test lengths $n = n_k$. The procedure receives a training constraint of length $t(n_k) = k$. It seeks to minimize $\mathcal{R}(T)$ subject to matching $f$ on length $k$. By definition of $U_k$, there exists a transformer $T^* \in \Theta_{n_k}$ that matches $f$ on length $k$ with $\mathcal{R}(T^*) \leq U_k + \epsilon$. However, by our choice of $n_k$, any transformer that matches $f$ on the full length $n_k$ requires a cost $> U_k + 1$. Since the Inference Procedure minimizes cost, it will prefer $T^*$ (cost $\approx U_k$) over any solution that generalizes to $n_k$ (cost $> U_k + 1$). Thus, the inferred transformer $T_{n_k}$ will match the training data (length $k$) but fail to match the target function $f$ at the test length $n_k$. Since this occurs for infinitely many $k$, the inference procedure fails to length generalize

$\square$

## C.6. Length Generalization for C-RASP

Any **C\*-RASP** or **C\*-RASP**[Pos] program can be translated to a Symbolic Limit Transformer. If the positional functions $\psi(i, j)$ and all the constants of the type $\delta_k, \gamma_k$ from our Match predicate (definition 3.3) are not used ($\tau_k$ can still be used), then then length generalization will be guaranteed even without positional embeddings. If any of them are required to be used, then length generalization is only guaranteed with the use of absolute positional embeddings. We say a Symbolic Limit Transformer $T$ *accepts* an input if the value in the last dimension in the last position of the output is greater than $0$, and rejects otherwise.

**Theorem C.21.** *For every **C\*-RASP** program $P$ with local functions $\Psi$, there exists a Symbolic Limit Transformer $T_\infty$ such that for all $w \in \Sigma^*$, $P$ accepts $w$ iff $T_\infty$ accepts $\$w$. Furthermore:*

1. *If $P$ uses neither local positional relations $\psi(i, j)$ nor spatial offsets in Match Predicates (i.e., $\delta_k = \gamma_k = 0$ for all matches, though $\tau_k \neq 0$ is allowed), then $T_\infty$ requires no positional encodings (NoPE), and length generalization is guaranteed for NoPE transformers.*

2. *If $P$ uses local positional relations $\psi(i, j)$ or non-zero spatial offsets ($\delta_k \neq 0$ or $\gamma_k \neq 0$), then $T_\infty$ utilizes absolute positional encodings, and length generalization is guaranteed for APE transformers.*

As a consequence, the Inference Procedure will ultimately length-generalize on inputs from a function $f$ expressible by a **C\*-RASP**[Pos] program. If the **C-RASP** program requires no positional functions as said above, then length generalization will succeed even with NoPE transformers.

*Remark* C.22. We note that the Symbolic Limit Transformer $T_\infty$ provided by the proof of Theorem C.21 emulates the **C-RASP** program $P$ at zero offset: That is, $P$ accepts $w$ iff a predetermined entry in the last output dimension of $T_\infty(\$w, 0)$ is above some threshold. In principle, its computations may not be offset-invariant, i.e., for the constructed $T_\infty$, the output $T_\infty(\$w, o)$ may depend on $o$. Importantly, the proof of Theorem C.10 does not require a Symbolic Limit Transformer computing $f$ to be offset-invariant, but just requires it to compute $f$ when the offset is zero. This is because Statement 2 of Proposition C.13 ensures that, for any Symbolic Limit Transformer $T_\infty$, even if it is not offset-invariant, there are transformers $T_n \in \Theta_n$ whose behavior matches $T_\infty(\cdot, 0)$.

*Proof.* We will also show that a Symbolic Limit Transformer $T_\infty$ simulates a **C\*-RASP** program $P$ if for every operation $P_k$ of $P$ there is a dimension $d_k$ in $T$ such that when $P_k(i)$ when run on $w$ is true iff $T_\infty(\$w)_{i+1,d_k} = 1$ (and $0$ otherwise) for Boolean operation and $P_k(i) = c$ iff $T_\infty(\$w)_{i+1,k} = \frac{c}{i+1}$ for count operations.

We will use induction on the length of $P$. All boolean cases and all but one counting case are identical to Huang et al. (2025b). The only distinct operation which still needs proving is the match predicate.

We restate the proof of the case when $C(i) := \#\,[\,j \leq i, \psi(i, j)\,]\ P(j)$ from Huang et al. (2025b) when $\psi$ is a local function of the following form

$$\psi(i,j) = \begin{cases} 1 & j = i - \ell \\ 0 & \text{else} \end{cases}$$

We are doing this, as we will re-use a key idea here to implement our match predicate. Thus, in this $C(i)$ will either be 1 or 0 depending if $P(i - \ell)$ is true or false. If we set the query and key matrices to 0 we get

$$s_{ij} = \log N \cdot \psi(i,j)$$

We assume the $\log$ is base 2, but the argument is similar for others. Then we can have attention compute

$$c_{i,k} = \frac{\displaystyle\sum_{j \leq i} \exp\left(\log N \cdot \psi(i,j)\right) \cdot P(j)}{\displaystyle\sum_{j \leq i} \exp\left(\log N \cdot \psi(i,j)\right)} = \frac{\displaystyle\sum_{j \leq i} N^{\left(\frac{\psi(i,j)}{\ln 2}\right)} \cdot P(j)}{\displaystyle\sum_{j \leq i} N^{\left(\frac{\psi(i,j)}{\ln 2}\right)}}$$

If $P(i - \ell)$ and $\neg P(j)$ for $j \neq i - \ell$, then we have a lower bound:

$$\frac{N^{\left(\frac{1}{\ln 2}\right)}}{N^{\left(\frac{1}{\ln 2}\right)} + i - 1} \leq c_{i,k}$$

If $\neg P(i - \ell)$ and $P(j)$ for $j \neq i - \ell$ then we have an upper bound:

$$c_{i,k} \leq \frac{i - 1}{N^{\left(\frac{1}{\ln 2}\right)} + i - 1}$$

Since $N^{\frac{1}{\ln 2}} \geq i$, and we know that $P(i - \ell) \iff c_{i,k} \geq \frac{1}{2}$, we can construct an MLP that computes the correct value. It will output either $\frac{0}{i+1}$ or $\frac{1}{i+1}$, in the dimension reserved for $P_{k+1}(i)$, for instance by using a conditional operation (allowed in the syntax of the language) that checks that the output of the attention layer $c_{i+1,k} \geq \frac{1}{2}$.

Now, when $C(i) := \#\left[j \leq i, \chi(i,j)\right]$, where $\chi(i,j) \equiv \bigwedge_{k=1}^{K}(\boldsymbol{c}_{j-\delta_k} = \boldsymbol{c}_{i-\gamma_k} + \tau_k)$. The $\boldsymbol{c}$ here refers to integers representing the elements of the extended alphabet, whereas everywhere else before in this proof, we were referring to $c$ as counts. Similarly $k$ here refers to the constants in this specific condition. Just for this proof, we will use $\sigma$ to refer to characters to avoid confusion, and $r$. Hence, we want to prove that $C(i) := \#\left[j \leq i, \chi(i,j)\right]$, where $\chi(i,j) \equiv \bigwedge_{r=1}^{R}(\sigma_{j-\delta_r} = \sigma_{i-\gamma_r} + \tau_r)$. Note that all the $\gamma_r, \tau_r, \delta_r$ are constants and help in looking around in the neighborhood, and the total number of such conditions $R$ is also bounded. We reserve $2R$ spots in our residual stream.

We use the same process defined earlier in computing our local $\psi(i,j)$ functions, except in the attention computation, we use the value matrices to pass the value of the character embeddings at offsets to be transferred to the current position. We take the value matrix to be the identity matrix $I$ [11]. Therefore, the only change would be that we use $\sigma_j$ instead of P(j) in our derivation above. Thus, $2R$ spots out of all will now store each of our values of $\sigma_{i-\gamma_r}, \sigma_{i-\delta_r}$.

We can assume that the $\sigma_i$ here are be one-hot vectors representing the corresponding integers. Adding a constant $\tau$ corresponds to a linear shift such that $\mathbf{R}_\tau \sigma_i = \sigma_i + \tau$. So, $\mathbf{R}_\tau$ can be implemented as a shifted-diagonal matrix. Note that this is just one way in which this could happen. There could be other ways of this being true. We just want to be able to go from one character's embedding to another with a linear transformation. Thus, we can hardcode these shifts in the KQ parameter matrices doing the retrievals of $\sigma_{i-\gamma_r}$. Thus, instead of $KQ = 0$, they implement such a shift. Therefore we will already have the values of $\sigma_{i-\gamma_r} + \tau_r$ in $R$ positions.

Thus, with these many attention heads, now we have all the information in our residual stream, and we need to actually compute the equality and the $\wedge$ condition. We construct an attention head for this. We use the key and query matrices to concatenate the respective slots that hold all the R $\sigma_{i-\delta_r}$ vectors and all the R $\sigma_{i-\gamma_r} + \tau_r$ vectors respectively, and the

---

[11] Allowed now, as the rank constraint on V matrices is no longer there, unlike in Huang et al. (2025b)

product between these will be maximum when all the sets of $R$ vectors match, which is what we want. Additionally, the key query matrices also scale up all the one hot vectors by a factor of 2. Thus now, we note that if $\chi(i,j)$ is true, then the score is $2R$, and $\leq 2(R-1)$ if even one condition fails. Therefore to verify a full match (and avoid confusion with partial matches), the attention score to the \$ is always $2R$. The value matrix at \$ holds the value 1, and the value matrix at all other positions hold the value 0. Let $M_i = \{j \leq i | \chi(i,j) = True\}$ and $m = |M_i|$ be the count we wish to compute. Based on our construction of $Q$ and $K$, the attention scores $s_{ij}$ satisfy:

$$s_{ij} = \begin{cases} 2R & \text{if } j = \$ \text{ (Start Token)} \\ 2R & \text{if } j \text{ satisfies the match requirements)} \\ \leq 2(R-1) & \text{if } j \text{ Mismatch} \end{cases} \tag{31}$$

The output of the attention head at position $i$ therefore

$$h_i = \frac{\sum_{j \leq i} \exp(\log N \cdot s_{ij}) V(j)}{\sum_{j \leq i} \exp(\log N \cdot s_{ij})}$$

$$= \frac{\exp(\log N \cdot s_{i\$}) \cdot 1 + \sum_{j \neq \$} \exp(\log N \cdot s_{ij}) \cdot 0}{\exp(\log N \cdot s_{i\$}) + \sum_{j \in M_i} \exp(\log N \cdot s_{ij}) + \sum_{j \notin M_i \cup \{\$\}} \exp(\log N \cdot s_{ij})}$$

Dividing both the numerator and the denominator by $\exp(\log N \cdot 2R)$, and substituting the score values, we obtain:

$$h_i = \frac{1}{1 + \sum_{j \in M_i} \frac{\exp(\log N \cdot 2R)}{\exp(\log N \cdot 2R)} + \sum_{j \notin M_i \cup \{\$\}} \frac{\exp(\log N \cdot s_{ij})}{\exp(\log N \cdot 2R)}}$$

$$= \frac{1}{1 + m \cdot 1 + \sum_{j \notin M_i \cup \{\$\}} \exp(s_{ij} - \log N \cdot 2R)}$$

We analyze the error term caused by mismatches. Since $s_{ij} \leq \log N \cdot 2(R-1)$ for mismatches, the exponent is bounded by $-2 \log N$. There are at most $i$ such mismatching terms (bounded by context length $N$). Thus:

$$\sum_{j \notin M_i \cup \{\$\}} \exp(s_{ij} - \log N \cdot 2R) \leq N \cdot \exp(-2 \log N) = N \cdot N^{-2} = \frac{1}{N}$$

Therefore, the output of the attention head is:

$$h_i = \frac{1}{1 + m + O(N^{-1})} \tag{32}$$

As Izzo et al. (2026) argued, because of the restriction that attention logits and the output of the $\exp(\cdot)$ operation be rounded p fractional bits of precision, the $O(1/N)$ terms gets rounded to 0. Thus our value converges strictly to $\frac{1}{1+m}$. This is f As $m$ is an integer bounded by $N$, the values $\{\frac{1}{1}, \frac{1}{2}, \ldots, \frac{1}{N+1}\}$ are well-separated. We use our activation function $f(x) = 1/x$ to invert this value to recover $m$ (or compute $\frac{m}{i+1}$ as required by the count operation) with arbitrary precision.

$\square$

**Lemma C.23.** *C\*-RASP*[Pos] *where the input alphabet* $\Omega$ *is finite, is equivalent to* *C-RASP*[Pos].

*Proof.* The syntax of **C\*-RASP**[Pos] closely tracks the syntax of **C-RASP**. The only additional operation in **C\*-RASP** is that of the match predicate $\chi(i,j)$.

The class **C\*-RASP** extends **C-RASP** solely through the addition of the **Match** operation. We show that when $\Sigma$ is finite, any function defined by a Match operation can be simulated by a finite sequence of standard **C-RASP** operations.

Recall the definition of the Match operation for a specific neighborhood structure defined by constants vectors $\delta, \gamma \in \mathbb{N}^K$ and $\tau \in \mathbb{Z}^K$:

$$C_{\text{match}}(i) := \#[j \leq i, \chi(i,j)] \tag{33}$$

where $\chi(i,j) \equiv \bigwedge_{k=1}^{K} (c_{j-\delta_k} = c_{i-\gamma_k} + \tau_k)$.

Since $\Sigma$ is finite, we know all pairs of $(\sigma, \sigma + \tau)$ we are looking for. Let us say that we enumerate each of these as $(a_k, b_k)$, where $k \in [1, K]$. The number of such pairs is equal to the size of the alphabet $\Sigma$, as every character from $\Sigma$ can only be at a fixed distance $\tau$ to one other character from $\Sigma$. Thus, the whole match predicate boils down to a couple of steps. The first step is finding a $j$ in the past, which has the symbol $b_k$ stored at $\delta_k$ distance away from it. This local operation can be encoded as following in **C-RASP**[local].

$$P_{\text{past}}(i) := \bigwedge_{k=1}^{K} (\#\,[j \leq i, j + \delta_k = i]\ Q_{b_k}(j) > 0)$$

The second step is to check whether the current position also has symbols $a_k$ stored at a distance of $\gamma_k$ distance from it. This is fairly similar to the previous check, and can use similar commands.

$$P_{\text{current}}(i) := \bigwedge_{k=1}^{K} (\#\,[j \leq i, j + \gamma_k = i]\ Q_{a_k}(j) > 0)$$

The final step is counting the number of past positions where step 1 holds, and also making sure that step 2 holds at the current position. Thus,

$$C_{match}(i) := \#\,[j \leq i]\ P_{\text{past}}(j) \wedge P_{\text{current}}(i)$$

Although once we know either of $a_k$ or $b_k$, we can get the other character in the pair. Apriori, we do not know the identity of the current position. In case of the infinite alphabet, the key was that the program could perform the matching without relying on the exact identity of the symbol. To simulate the same, instead of the predicates $Q_{a_k}$ or $Q_{b_k}$, we need to replicate the whole 3 steps for all possible pairs in $\Sigma$ at every step $k$. Thus, the 3 lines of program are repeated $|\Sigma|$ times, each time with a different $(a_k, b_k)$ combination assigned to each $k$. Since, we can simulate the only extra command in **C\*-RASP**[Pos] using commands of **C-RASP**[local], when the alphabet is finite, the two are equivalent.

$\square$

We restate Theorem 3.5 and prove it.

**Theorem C.24.** *Consider the alphabet* $\Sigma = \{a, b, e\}$. *Then,* $PARITY := b^*(ab^*ab^*)^* \notin$ **C-RASP**$[\infty, local]$. *and Flip flop* $:= \Sigma^* b e^* \notin$ **C-RASP**$[\infty, local]$.

*Proof.* The proof of both of the languages above is analogous. We represent the corresponding language as $L$, and both $PARITY$ or Flip-Flop can be substitued in place of $L$ in our forthcoming arguments. We assume that there exists a program **C\*-RASP**[Pos] , $P$ for $L$. Since, $P$ works for $L$, it can work for $L$ with bounded alphabet as well, as the $\Sigma$ of $L$ is simply $\Sigma = \{a, b, e\}$. However as per theorem C.23, such a $P$ can be simulated in **C-RASP**[local]. That would imply that $L \in$ **C-RASP**[periodic, local], which was proven to be impossible in Lemma 11 of Huang et al. (2025b). Thus, our initial assumption that such a program exists must be wrong. Hence, the corollary stands. $\square$

**Remark on Periodic Relations:** Huang et al. (2025b) introduce **C-RASP**[periodic,local], which includes both local relations ($\psi$ in the counting operation), and periodic relations $\phi(i, j)$ checking if $i \equiv k \pmod{m}$. In our new framework, we omit such periodic relations. In Huang et al. (2025b), accounting for these relations necessitated the assumption that the regularizer penalizes the *rank* of $V$ matrices. In our present framework, we instead only penalize the spectral *norm* of these matrices, for two reasons: (i) first, because a norm penalty is better motivated as a proxy for standard learning methods than a rank penalty, (ii) second, because penalizing the rank of $V$ matrices would preclude moving information about object names across adjacent positions, which in fact is important for verifying plan validity.

In Huang et al. (2025b), periodic relations were important for explaining APE length generalization on certain formal languages such as $(aa)^*$, which are definable in **FO**$[Reg]$ but not **FO**$[<]$. Such languages are not directly relevant to our present results, as they are not star-free but empirically easier for transformers than PARITY/FlipFlop. In particular, including such relations would not change Theorems 3.1 and 3.6. We leave to future work to better understand the role of periodic relations in length generalization.

**Remark on Unique Copying Task:** Unique copying is the task of copying a string that is composed of distinct tokens. Huang et al. (2025b) showed that that task of unique copying is in **C-RASP**. However such kind of copying would require an increase of vocabulary during test time, and thus actually creates a challenge for rigorously treating the task using the results of Huang et al. (2025b), which assumes a fixed alphabet. Huang et al. (2025b) addressed this problem by formalizing Unique Copy as repeating a string unboundedly often with a fixed alphabet. However, formalizing the task in our new framework becomes much more principled and more true to the original formulation of the task. Essentially, the following lines of program are enough to detect that unique copying was successful:

$$\text{OnlyOneMatch}(i) := (\# \left[ j \leq i, c_{j-1} = c_i \right] \ == 1)$$

$$\text{UniqueCopySuccessful}(i) := (\# \left[ j \leq i \right] \ \neg\text{OnlyOneMatch}(i) == 0)$$

# D. Experimental Details

## D.1. Domains and Dataset Generation

For the generation of our training and test datasets, we use custom generators instead of existing problem generators and symbolic planners in order to control the lengths of the generated plans and get more variation. The generation of correct plans is mostly based on randomly selecting applicable action sequences but varies slightly depending on the domain. We first describe the shared set-up and then provide the domain-specific details below.

Table 2 provides an overview over the characteristics of the six datasets we generate (see Table 3 for information about size and train/test splits). Note that the initial state of the Colors planning instances is fixed in the sense that all bags are empty initially. However, the part of the initial state that defines which objects are colors and bags varies depending on $O$.

All datasets contain the same number of valid and invalid plans. In particular, we generate a set of planning instances $\Pi_i = \langle \mathcal{D}, O_i, I_i, G_i \rangle$ and for each $\Pi_i$ we create a plan $\pi = [a_1, \ldots, a_k, \ldots, a_n]$ such that $\pi$ induces a state sequence from $s_1 = I$ to $s_{n+1}$ such that $s_{n+1} \models G$, i.e. $\pi$ is a valid plan. Additionally, we generate an invalid plan $\pi'$ for each $\Pi_i$ and add $\langle \Pi_i, \pi \rangle$ and $\langle \Pi_i, \pi' \rangle$ to the dataset.

We consider two ways of constructing invalid plans: converting the valid plan into an *incomplete* plan or a *non-executable* plan. To obtain the incomplete plans, the valid plan $\pi = [a_1, \ldots, a_k, \ldots, a_n]$ is converted into an invalid plan $\pi' = [a_1, \ldots, a'_k, \ldots, a'_m]$ inducing a sequence of states $s_1, \ldots, s'_{m+1}$ where $s_1 = I$ and $s'_{m+1} \not\models G$. The incomplete plans are created such that exactly one (in the case of Heavy Grippers, Colors) or at least one (Lights Out) of the literals in the goal is not satisfied. For Lights Out and Colors, $\pi'$ contains exactly the same number of actions as $\pi$. For Heavy Grippers, the number of actions is similar but some of the invalid plans are slightly shorter.

For Heavy Grippers, we additionally generate a dataset with non-executable plans. To obtain a non-executable plan $\pi'$, the valid plan is converted into an action sequence $\pi' = [a_1, \ldots, a_{m-1}, a'_m]$ such that $[a_1, \ldots, a_{m-1}]$ induces a sequence of states $s_1, \ldots, s'_m$ and $s'_m \not\models \text{pre}(a'_m)$. We then combine the datasets with the incomplete plans and non-executable plans into the final dataset. In the STRIPS Colors domain and the Lights Out domain with conditional effects all actions are always applicable. Therefore, we cannot generate a dataset with non-executable plans here and only consider incomplete ones.

The valid and invalid plans for the Colors and Lights Out domains are minimal pairs, differing only by single action. The same applies to the valid and non-executable plans in the Heavy Grippers datasets. However, some of the incomplete plans differ slightly more due to the specific characteristics of the domain and might have a slightly different plan length than their valid counterparts.

**Heavy Grippers.** The Gripper domain is a well-known planning domain from the IPC benchmarks. Here, we create a variant of it that introduces two new aspects to the domain, namely that balls can be heavy and that the robot has a charge that affects how it can pick and drop balls (see Section 2.1 and Figure 2).

We create a well-formed and a delete-free variant of our Heavy Grippers domain shown in Figure 2. First, we eliminate the conditional effects by separating pick into two actions, $pick$ and $pickHeavy$, and remove charged from the preconditions of $pick$. Second, we create the well-formed variant by extending the preconditions and the delete-free variant by removing the delete effects, resulting in the two domains shown in Figure 7.

For the generation of the planning instances we fix the number of grippers to 2 (the standard number for the IPC Gripper variant), but vary the number of balls and rooms. In particular, when generating a plan of length $N$, we sample the

| Domain | Domain variant | Object set O | Fixed states | Invalid plans |
|---|---|---|---|---|
| Heavy Grippers | well-formed | varying number and names | none | incomplete + non-executable |
| Heavy Grippers | delete-free | varying number and names | none | incomplete + non-executable |
| Colors | well-formed | varying number and names | initial I | incomplete |
| Colors | STRIPS | varying number and names | initial I | incomplete |
| Lights Out | well-formed | fixed | goal G | incomplete |
| Lights Out | conditional effects | fixed | goal G | incomplete |

*Table 2.* Overview over the datasets generated for our experiments.

---

**Predicates:** $room(r), ball(b), gripper(g), free(g), heavy(b)$, $charged, atRobby(r), at(b, r), carry(b, g)$

**Action** $move(r_1, r_2)$:
  **Pre:** $\{room(r_1), room(r_2), atRobby(r_1), \neg atRobby(r_2), \neg charged\}$
  **Eff:** $\{charged, atRobby(r_2), \neg atRobby(r_1)\}$

**Action** $pick(b, r, g)$:
  **Pre:** $\{ball(b), room(r), \neg heavy(b), gripper(g), atRobby(r),$
    $free(g), at(b, r), \neg carry(b, g)\}$
  **Eff:** $\{carry(b, g), \neg free(g), \neg at(b, r)\}$

**Action** $pickHeavy(b, r, g)$:
  **Pre:** $\{ball(b), room(r), heavy(b), gripper(g), atRobby(r),$
    $free(g), charged, at(b, r), \neg carry(b, g)\}$
  **Eff:** $\{carry(b, g), \neg free(g), \neg at(b, r), \neg charged\}$

**Action** $drop(b, r, g)$:
  **Pre:** $\{ball(b), room(r), gripper(g), atRobby(r), carry(b, g),$
    $charged, \neg at(b, r), \neg free(g)\}$
  **Eff:** $\{at(b, r), free(g), \neg carry(b, g), \neg charged\}$

---

**Predicates:** $room(r), ball(b), gripper(g), free(g), heavy(b)$, $charged, atRobby(r), at(b, r), carry(b, g)$

**Action** $move(r_1, r_2)$:
  **Pre:** $\{room(r_1), room(r_2), atRobby(r_1)\}$
  **Eff:** $\{charged, atRobby(r_2)\}$

**Action** $pick(b, r, g)$:
  **Pre:** $\{ball(b), room(r), gripper(g), atRobby(r), at(b, r),$
    $free(g), \neg heavy(b)\}$
  **Eff:** $\{carry(b, g)\}$

**Action** $pickHeavy(b, r, g)$:
  **Pre:** $\{ball(b), room(r), gripper(g), atRobby(r), at(b, r),$
    $free(g), charged, heavy(b)\}$
  **Eff:** $\{carry(b, g)\}$

**Action** $drop(b, r, g)$:
  **Pre:** $\{ball(b), room(r), gripper(g), atRobby(r), carry(b, g)\}$
  **Eff:** $\{at(b, r), free(g)\}$

---

*Figure 7.* The well-formed (left) and delete-free (right) variants of the Heavy Grippers domain used in our experiments. Variants were obtained from the variant shown in Figure 2.

number of balls $n_b$ from the range $[0.6 * N, 0.85 * N]$. The number of heavy balls and rooms is selected by sampling from $[0.45 * n_b, 0.85 * n_b]$ and range $[0.2 * n_b, 0.5 * n_b]$ respectively. This sampling strategy results in different combinations of plan lengths and number of objects while modeling that more objects tend to correlate with larger plans, which is the standard in planning datasets. We randomly assign a name to each object sampled from a fixed set of possible names. This set contains object names ranging from Object0 to ObjectM, where M is at least as large as the maximum number of objects within one instance in our dataset.

For the initial state $I$, the initial locations of the robot and all balls are randomly assigned. We then sample a sequence of applicable actions until reaching the target length and make sure that in the resulting state each ball is located in a room, i.e. not carried anymore. The goal $G$ consists of the locations of all balls obtained after applying the sampled sequence of applicable actions. For the delete-free variant, part of the actions of the sequence are sampled from all actions applicable under the delete-free variant and some are sampled from the smaller set of actions applicable under the well-formed variant. This mixture increases the probability that balls are actually dropped again which is important for getting reasonable goal states.

The non-executable plans are obtained by replacing the last action of the valid plan with an action for which the preconditions are not valid at that step in the action sequence. This action is sampled at random from the non-executable move and drop actions. Pick actions are not considered because valid plans never end with pick actions, and this type of mistake could be easily identified without actually reasoning over the planning instance and plan.

For the generation of incomplete plans, it is not possible to simply replace a single action. We select one ball and modify the plan such that the last step at which that ball is dropped does not drop the ball in the goal room anymore, which requires also changing the movements of the robot. Figure 8 shows a small example planning instance from the Heavy Grippers domain, with plans that are valid ($\pi$), incomplete ($\pi'_1$) and non-executable ($\pi'_2$) under the well-formed domain variant. Under the

$O = \{$object_237, object_223, object_100, object_154, object_280, object_113, object_94, object_7, object_76$\}$
$I = \{at\text{-}robby($object_280$), gripper($object_237$), gripper($object_223$), free($object_237$), free($object_223$),$
$\qquad room($object_100$), room($object_154$), room($object_280$), room($object_113$),$
$\qquad ball($object_94$), ball($object_7$), ball($object_76$), heavy($object_94$),$
$\qquad at($object_94, object_100$), at($object_7, object_154$), at($object_76, object_280$)\}$
$G = \{at($object_94, object_280$), at($object_7, object_154$), at($object_76, object_154$)\}$

$\pi = [pick($object_76, object_280, object_223$), move($object_280, object_100$), pick\_heavy($object_94, object_100, object_237$),$
$\qquad move($object_100, object_154$), drop($object_76, object_154, object_223$), move($object_154, object_280$),$
$\qquad drop($object_94, object_280, object_237$), move($object_280, object_113$)]$

$\pi'_1 = [pick($object_76, object_280, object_223$), move($object_280, object_100$), pick\_heavy($object_94, object_100, object_237$),$
$\qquad move($object_100, object_154$), drop($object_76, object_154, object_223$), \underline{move($object_154, object_113$)},$
$\qquad \underline{drop($object_94, object_113, object_237$)}]$

$\pi'_2 = [pick($object_76, object_280, object_223$), move($object_280, object_100$), pick\_heavy($object_94, object_100, object_237$),$
$\qquad move($object_100, object_154$), drop($object_76, object_154, object_223$), move($object_154, object_280$),$
$\qquad drop($object_94, object_280, object_237$), \underline{drop($object_76, object_280, object_223$)}]$

*Figure 8.* An example planning instance from the well-formed Heavy Grippers domain with a valid plan $\pi$, an incomplete plan $\pi'_1$ and a non-executable plan $\pi'_2$. The actions by which the invalid plans differ from the valid one are underlined.

---

**Predicates:** bag$(b)$, color$(c)$, hasColor$(b, c)$

**Action** remove$(c, b)$:
  **Pre:** $\{$bag$(b)$, color$(c)\}$
  **Eff:** $\{\neg$hasColor$(b, c)\}$

**Action** add$(c, b)$:
  **Pre:** $\{$bag$(b)$, color$(c)\}$
  **Eff:** $\{$hasColor$(b, c)\}$

---

**Predicates:** bag$(b)$, color$(c)$, hasColor$(b, c)$

**Action** remove$(c, b)$:
  **Pre:** $\{$bag$(b)$, color$(c)$, hasColor$(b, c)\}$
  **Eff:** $\{\neg$hasColor$(b, c)\}$

**Action** add$(c, b)$:
  **Pre:** $\{$bag$(b)$, color$(c)$, $\neg$hasColor$(b, c)\}$
  **Eff:** $\{$hasColor$(b, c)\}$

---

*Figure 9.* The standard (left) and well-formed (right) variants of the Colors domain.

delete-free variant, both $\pi$ and $\pi'_2$ are valid plans, but $\pi'_1$ is also incomplete.

**Colors.**  This domain involves placing colored balls into bags, where actions add or remove a color from a specific bag, and goals specify which colors each bag should contain. We design a standard STRIPS and a well-formed variant of the domain as shown in Figure 9. The variants differ in how they handle redundant operations, i.e. adding a color that is already in a bag or removing one that is not. The well-formed variant enforces that a color can be added or removed only if it is absent or present in the bag, respectively, while the standard STRIPS does not have this restriction.

For all planning instances, we define the initial state as one in which no color is in any bag. We choose the number of colors and bags as a function of the target plan length: we set the total number of bag–color pairs (i.e., #colors $\times$ #bags) to approximately one quarter of the plan length, and we make the numbers of colors and bags as close as possible. We randomly assign a name to each object sampled from a fixed set of possible names. This set contains object names ranging from Object0 to ObjectM, where M is at least as large as the maximum number of objects within one instance in our dataset. Valid plans are generated by randomly sampling applicable actions until reaching the target length. The goal is then defined as the set of all hasColor facts that hold in the resulting state.

In order to obtain the incomplete plans, we randomly substitute an action in the plan with another applicable action, and verify that this substitution leads to a different end state, failing the original goals. Figure 10 shows a small example planning instance from the Colors domain where the goal is to have color object3 in bag object5 and color object8 in bag object6. The figure shows a valid and an incomplete plan for the well-formed variant and a valid and an incomplete plan for the standard STRIPS. For both cases, the invalid plans are obtained by replacing a single action that has the effect that object3 will not be in object5 in the end. Note that the valid plan for the STRIPS variant is invalid under the well-formed variant but the well-formed valid plan would also be valid under the standard STRIPS domain.

---

**Well-formed variant**
$O = \{\text{object\_5}, \text{object\_6}, \text{object\_3}, \text{object\_8}\}$
$I = \{bag(\text{object\_5}), bag(\text{object\_6}), color(\text{object\_3}), color(\text{object\_8})\}$
$G = \{hasColor(\text{object\_5}, \text{object\_3}), hasColor(\text{object\_6}, \text{object\_8})\}$

$\pi_1 = [add(\text{object\_3}, \text{object\_5}), add(\text{object\_8}, \text{object\_5}), remove(\text{object\_3}, \text{object\_5}),$
$\qquad add(\text{object\_8}, \text{object\_6}), \underline{add(\text{object\_3}, \text{object\_5})}, remove(\text{object\_8}, \text{object\_5})]$
$\pi_1' = [add(\text{object\_3}, \text{object\_5}), \overline{add(\text{object\_8}, \text{object\_5})}, remove(\text{object\_3}, \text{object\_5}),$
$\qquad add(\text{object\_8}, \text{object\_6}), \underline{add(\text{object\_3}, \text{object\_6})}, remove(\text{object\_8}, \text{object\_5})]$

**Standard variant**
$O = \{\text{object\_5}, \text{object\_6}, \text{object\_3}, \text{object\_8}\}$
$I = \{bag(\text{object\_5}), bag(\text{object\_6}), color(\text{object\_3}), color(\text{object\_8})\}$
$G = \{hasColor(\text{object\_5}, \text{object\_3}), hasColor(\text{object\_6}, \text{object\_8})\}$

$\pi_2 = [remove(\text{object\_3}, \text{object\_5}), add(\text{object\_8}, \text{object\_5}), add(\text{object\_8}, \text{object\_5}),$
$\qquad add(\text{object\_8}, \text{object\_6}), remove(\text{object\_8}, \text{object\_5}), remove(\text{object\_3}, \text{object\_6})]$
$\pi_2' = [\overline{remove(\text{object\_3}, \text{object\_5})}, add(\text{object\_8}, \text{object\_5}), add(\text{object\_8}, \text{object\_5}),$
$\qquad \underline{remove(\text{object\_8}, \text{object\_5})}, remove(\text{object\_8}, \text{object\_5}), remove(\text{object\_3}, \text{object\_6})]$

*Figure 10.* A small planning instance of the Colors domain with a valid plan $\pi_1$ and an incomplete plan $(\pi_1')$ for the well-formed variant and a valid $(\pi_2)$ and incomplete $(\pi_2')$ plan for the standard STRIPS. The actions by which the valid and invalid plans differ are underlined.

**Lights Out.** While the well formed variants of Heavy Grippers and Colors domains serve as positive cases to show generalization over variable objects, we will use Lights Out to show negative results. It is important to note that this restriction does not undermine the validity of our evaluation. On the contrary, since conditional effects of Lights Out represents a negative result, the failure to generalize *even* for a *fixed* set of objects matches our predictions from Theorem 3.1. Lights Out is a puzzle consisting of lights arranged in a square grid. Each light is either on or off, and pressing a light toggles the state of itself and of each of the adjacent lights. The goal is to have all the lights off. We consider two formalizations of this domain, one with conditional effects and one that is well-formed. We use the standard 5x5 in all our Lights Out datasets with lights named $L_{00}, L_{01}, ... L_{44}$ according to their positions in the grid.

The variant with conditional effects includes only a single action schema for pressing a light as shown in Figure 11 (left). There are no preconditions for pressing a light and the effect on each of the affected lights (i.e. the pressed light and the adjacent lights) is conditioned on the current status of that light.

The well-formed variant has separate actions for pressing each light and for each possible combination of the state of that light and the adjacent lights. Figure 11 (right) shows two action schemas from the well-formed domain. The shown action schemas are two of the schemas for pressing the light in the upper left corner ($L_{00}$) which has two adjacent lights, $L_{10}$ and $L_{01}$. The other action schemas for pressing $L_{00}$ cover all the remaining possible $2^3$ on/off-combinations of $L_{00}$, $L_{10}$ and $L_{01}$. Overall, the domain consists of 512 action schemas. There could be many other ways of defining this, we choose one for our convenience.

All Lights Out planning instances have the same goal, i.e. all lights need to be off. In order to generate the initial states and valid plans, we start from the goal state and sample actions up to the target plan length. The resulting state becomes then the initial state and the valid plan is obtained by inverting the action sequence. The incomplete plans are obtained by replacing the last action with a different, applicable action which is guaranteed to result in a state where at least one light is turned on.

### D.2. Training and evaluation

**Setup.** Unless otherwise specified, we use a decoder-only Transformer based on the GPT-2 architecture, with 8 layers, hidden size 768, and 12 attention heads. We use trainable absolute positional embeddings (APE) and pre-layer normalization, and optimize it using the standard causal language modeling objective.

Inputs are tokenized with a domain-specific vocabulary that includes delimiter tokens, action tokens, argument and object tokens, the verdict tokens, along with other special tokens. Each example is constructed using delimiter tokens that separate

<table>
<tr><td>

**Predicates:** $on(l), out(l), adj(l_1, l_2)$

**Action** press$(l_1)$:
  **Pre:** $\emptyset$
  **Eff:**
$$\{out(l_1)\} \rhd \{on(l_1), \neg out(l_1)\}$$
$$\{on(l_1)\} \rhd \{out(l_1), \neg on(l_1)\}$$
$$\forall l_2.\{adj(l_1, l_2), on(l_2)\} \rhd \{out(l_2), \neg on(l_2)\}$$
$$\forall l_2.\{adj(l_1, l_2), out(l_2)\} \rhd \{on(l_2), \neg out(l_2)\}$$

</td><td>

**Constants:** $L_{00}, L_{01}, L_{02}, L_{03}, L_{04}, L_{10}, L_{11}, \ldots$

**Predicates:** $on(l), out(l)$

**Action** press-00-1:
  **Pre:** $\{out(L_{00}), out(L_{01}), on(L_{10}), \neg on(L_{00}), \neg on(L_{01}), \neg out(L_{10})\}$
  **Eff:** $\{on(L_{00}), on(L_{01}), out(L_{10}), \neg out(L_{00}), \neg out(L_{01}), \neg on(L_{10})\}$

**Action** press-00-2:
  **Pre:** $\{on(L_{00}), out(L_{01}), on(L_{10}), \neg out(L_{00}), \neg on(L_{01}), \neg out(L_{10})\}$
  **Eff:** $\{out(L_{00}), on(L_{01}), out(L_{10}), \neg on(L_{00}), \neg out(L_{01}), \neg on(L_{10})\}$

</td></tr>
</table>

*Figure 11.* Left: the Lights Out domain with conditional effects and universal quantification over adjacent lights. Right: an excerpt of the well-formed variant of Lights Out. The complete well-formed domain consists of 512 action schemas.

| Dataset | Variant | Train | Val-ID | Val-OOD | Test-ID | Test-OOD |
|---|---|---|---|---|---|---|
| Color Bags | Well-formed | 1,720,000 | 12,000 | 13,333 | 24,000 | 26,667 |
| Color Bags | STRIPS | 1,720,000 | 12,000 | 13,333 | 24,000 | 26,667 |
| Grippers | Well-formed | 360,000 | 18,000 | 20,000 | 18,000 | 20,000 |
| Grippers | Delete-free | 360,000 | 18,000 | 20,000 | 18,000 | 20,000 |
| Lights-Out | Well-formed | 180,000 | 6,000 | 6,666 | 12,000 | 13,334 |
| Lights-Out | Conditional effects | 180,000 | 6,000 | 6,666 | 12,000 | 13,334 |

*Table 3.* Number of instances per dataset, variant, and split.

the initial conditions $I$, the plan action sequence $\pi$, the goal $G$, and a final verdict token $V \in \{\texttt{correct}, \texttt{incorrect}\}$:

$$\texttt{<init>}\, I\, \texttt{<plan>}\, \pi\, \texttt{<goal>}\, G\, \texttt{<verdict>}\, V.$$

**Tokenization.** We tokenize plans at the level of action primitives and their arguments, with domain-specific schemes: (i) Lights Out (conditional-effect version): actions of the form $\texttt{press}(l_{ij})$ are tokenized as `<press>  <j>`. (ii) Lights Out (well-formed version): there are multiple distinct actions for pressing a specific light; we therefore tokenize these as `<press>  <j> <k>`. (iii) Colors and Heavy Grippers: actions such as $\texttt{add}(object_i, object_j)$ and $\texttt{pick}(object_i, object_j, object_k)$ are tokenized as `<add> <object_i> <object_j>` and `<pick> <object_i> <object_j> <object_k>`. Moreover, we only include tokens that are different for different instances. For example, for Colors, because all instances share the same initial state, i.e., all bags are empty, we omit the initial state from the input sequence.

**Data splits.** For all three domains, we generate planning instances and correct plans of lengths 11 up to 200, and their invalid counterparts. We train only on the dataset instances with a plan length of up to 100 and test on the complete range of plans. All datasets are balanced, with the same number of valid and invalid plans.

**Evaluation.** We train on plans of length 11–100 actions (ID) and validate on both 11–100 actions (ID) and 101–200 actions (OOD) datasets; here, "length" refers to $|\pi|$, instead of the total number of input tokens, which also increases monotonically with $|\pi|$. We run four random seeds and pick the checkpoint with the best validation performance.[12] Then we report both ID and OOD accuracy scores on held-out test sets. At inference time, we provide the prefix up to `<verdict>` and compare only the logits of the two verdict tokens at the last prompt position; the prediction is the one with the larger logit.

**Instance packing.** To improve training throughput while preserving instance boundaries, we concatenate multiple variable-length instances into fixed-length blocks of length $B = 4096$ tokens, without splitting any instance across blocks. If an instance does not fit in the remaining space, we start a new block. For Colors and Heavy Grippers allowing cross-instance attention leads to unstable optimization with large loss spikes, so we prevent tokens from different instances in the same block from attending to each other, i.e., attention is restricted to the tokens from the same instance, in addition to the standard causal mask. Instances are shuffled between each epoch to avoid memorization.

---

[12]For the not-well-formed variant of Colors, we exclude the results of a random seed, because it failed to converge to a high ID performance. The phenomenon is consistent with our observation regarding the in-distribution learnability analysis in Figure 12.

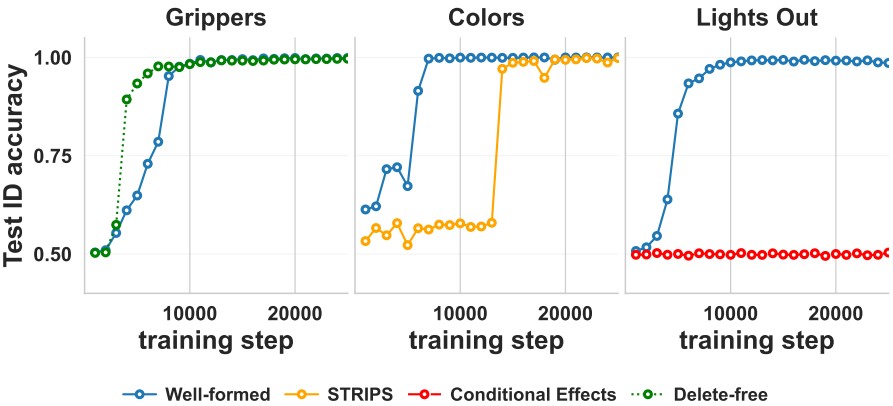

*Figure 12.* In-distribution test accuracy during the learning process (seed 0).

**Optimization.** Across all domains, we use a per-device batch size of $8$ blocks. Our pilot shows that a larger block, e.g., 16 or 32, achieves similar performance; we therefore stick to 8 for efficiency. We train for $50{,}000$ steps for all runs, and validate them every 1000 steps. We use AdamW with a linear learning-rate schedule, with warmup over the first $10\%$ of training steps, weight decay $0.1$, and gradient clipping to max norm $0.5$. We train in bfloat16 on a single A100 or H100 GPU per run.

We performed a learning-rate sweep over $\{2 \times 10^{-5}, 5 \times 10^{-5}, 1 \times 10^{-4}, 3 \times 10^{-4}, 6 \times 10^{-4}, 1 \times 10^{-3}, 2 \times 10^{-3}\}$ and selected the best setting per domain; this yields the following fixed learning rates: (i) Lights Out $2 \times 10^{-5}$, (ii) Colors $3 \times 10^{-4}$, (iii) Heavy Grippers $6 \times 10^{-4}$.

## E. Additional Empirical Results

**Extended main empirical results.** We first present an extended version of our main empirical results in Figure 13, i.e., the generalization accuracy across different plan lengths. Results are bucketed into 10-length intervals; for example, the point at $x = 20$ reports the accuracy over lengths 11–20. Besides the average accuracy scores, we also show the best seed and the standard deviation across all four seeds.

**Disentangling object-count and plan-length shifts.** In the main experiments for Colors and Heavy Grippers, the OOD setting increases both the number of objects and the plan length, reflecting the usual correlation between larger planning instances and longer plans. To better understand the separate effects of object-count growth and plan-length growth on generalization, we additionally evaluate two controlled settings: one where $|\pi|$ is fixed while $|O|$ increases, and one where $|O|$ is fixed while $|\pi|$ increases. We do not include Lights Out here because it already uses a fixed object universe.

The results show that the generalizable problem formulations continue to perform well under both settings. In Colors, the well-formed formulation remains near-perfect when either $|O|$ or $|\pi|$ increases, while the STRIPS formulation degrades in both settings. In Heavy Grippers, both the well-formed and delete-free formulations remain highly accurate. Although the delete-free formulation shows a mild decline as $|O|$ increases, performance remains strong overall. These results indicate that the observed variable-universe trends persist when object-count growth and plan-length growth are examined separately.

**In-distribution learnability.** We also compare the ID learnability of different planning variants, by plotting the curve of ID test accuracy during the training process in Figure 12. We use one random seed as an example here, and our observations are consistent across different seeds.

*Model failure on Lights Out with conditional effects.* Lights Out with conditional effects is the only setting in which models remain at chance-level accuracy. while others achieve near-perfect ID accuracy. This suggests that conditional-effect Lights Out is substantially more challenging, and highlights the massive benefit of the well-formed reformulation, which introduces an exponential (in grid size) number of actions and yields near-perfect accuracy for both ID and OOD lengths.

*Well-formed is easier than STRIPS.* For Colors, although both the well-formed and STRIPS variants achieve near-perfect ID accuracy, the well-formed variant is learned substantially earlier, again suggesting the benefit of well-formed actions.

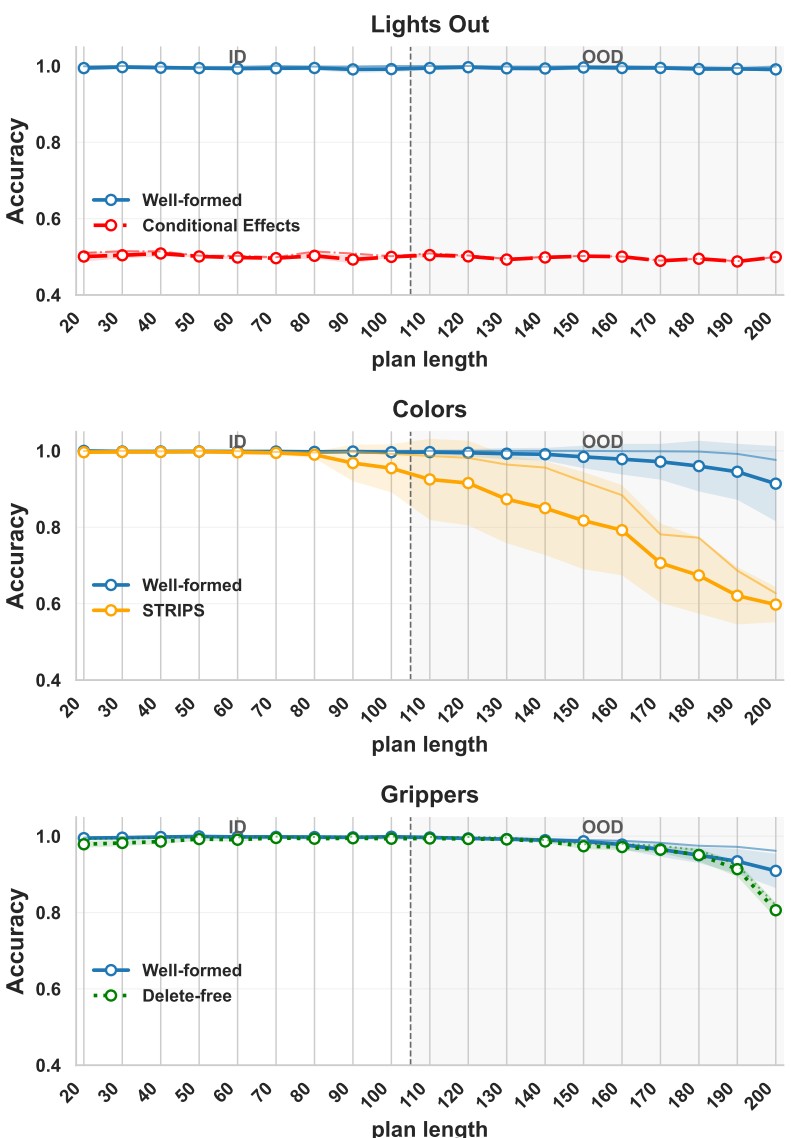

*Figure 13.* Accuracy scores for each 10-length bucket. This is an extended version of results in Figure 5.

*Delete-free vs well-formed.* For Heavy Grippers, the delete-free variant is learned slightly earlier than the well-formed variant, but variants do eventually converge.

## F. Manual Assessment of Well-formedness

Checking whether a domain is well-formed is hard if a domain is not syntactically well-formed. In order to still get a sense of the frequency of well-formed domains in existing benchmarks, we manually checked the 10 domains from the IPC Learning track 2023 (Taitler et al., 2024). For all domains, we first went through all action definitions in the domain definition file and checked for syntactic well-formedness. If some actions were not syntactically well-formed, we assessed their well-formedness based on the action definition together with the respective IPC problem generator [13]. We now illustrate our reasoning for the manual evaluation for one well-formed domain (Ferry) and one not well-formed domain (Satellite).

---

[13]https://github.com/ipc2023-learning/benchmarks/tree/main

| More objects | | | Longer plans | | |
|---|---|---|---|---|---|
| Setting | WF | STRIPS | Setting | WF | STRIPS |
| Colors, $|O| = 13$ | 0.9998 | 0.8021 | Colors, $|\pi| = 141$–$160$ | 0.9991 | 0.6714 |
| Colors, $|O| = 14$ | 0.9998 | 0.7174 | Colors, $|\pi| = 161$–$180$ | 0.9977 | 0.6699 |
| Colors, $|O| = 15$ | 0.9954 | 0.6663 | Colors, $|\pi| = 181$–$200$ | 0.9934 | 0.6351 |
| Setting | WF | DF | Setting | WF | DF |
| Heavy Grippers, $|O| = 81$–$100$ | 1.0000 | 0.9948 | Heavy Grippers, $|\pi| = 140$–$159$ | 1.0000 | 1.0000 |
| Heavy Grippers, $|O| = 101$–$120$ | 1.0000 | 0.9824 | Heavy Grippers, $|\pi| = 160$–$179$ | 1.0000 | 0.9832 |
| Heavy Grippers, $|O| = 121$–$140$ | 1.0000 | 0.9615 | Heavy Grippers, $|\pi| = 180$–$199$ | 0.9812 | 0.9012 |

*Table 4.* Controlled OOD evaluations separating object-count growth from plan-length growth. Left: $|\pi| = 100$, while $|O|$ increases. Right: $|O|$ is fixed, with 15 objects for Colors and 140 objects for Heavy Grippers, while $|\pi|$ increases.

**Ferry.** The Ferry domain defines three actions: "sail" for sailing between locations, "board" for moving a car onto the ferry and "debark" for moving a car from the ferry to a location. The "sail" action is syntactically well-formed, i.e. for each of the two effects their negation is one of the preconditions.

The action "board" has three effects: on($car$), ¬at($car, loc$), ¬empty-ferry(). The preconditions include empty-ferry() and at($car, loc$) but for on($car$), the negation is not part of the preconditions. Therefore, the well-formedness of the action depends on whether it is possible that on($car$) is true together with the preconditions of the "board" action. The problem generator for Ferry only generates problem instances where for each car C, there is exactly one location L such that at(C, L) is true initially, i.e. each car is at a specific location in the initial state. At the same time, there is no predicate on($car$) for any car in the initial state. The only action that makes on(C) true for a car C is the action "board" itself which makes at(C, loc) false. Additionally, the only action that makes at(C, loc) true for some location is the "debark" action which makes on(C) false. Therefore, it is not possible that for any car C and any location L it is true that on(C) ∧ at(C, L), and therefore "board" is well-formed.

The reasoning for the action "debark" is similar. The action has three effects: at($car, loc$), empty-ferry(), ¬on($car$). The preconditions are on($car$) and at-ferry($loc$) but they do not include ¬empty-ferry() nor ¬at($car, loc$). As described above, on(C) ∧ at(C, loc) can never be true for any car in any reachable state and therefore ¬at($car, loc$) is always true (for any location) when the precondition on($car$) holds. Regarding empty-ferry(), the problem generator always generates instances where empty-ferry() is true initially. The only action that makes this false is "board" which makes on($car$) true. Therefore, ¬empty-ferry() is exactly then true when there is some car C for which on(C) is true.

**Satellite.** The Satellite domain defines 5 actions: "turn_to" for turning the satellite into a specific direction, "switch_on" and "switch_off" for switching the power of a an instrument on and off, "calibrate" for calibrating an instrument and "take_image" for using an instrument to take an image. "turn_to" is syntactically well-formed and the other actions are not.

The action "take_image" has only one effect, namely have_image($dir, mode$) which specifies that an image has been taken in a specific mode and direction. There is no other action that has have_image($dir, mode$) as part of its effect. In particular, this means that there is no action that has ¬have_image($dir, mode$) as an effect and therefore have_image(D, M) will always stay true for any direction D and mode M if it was true once. At the same time, the preconditions of "take_image" do not include ¬have_image($dir, mode$). Therefore, it is possible to apply take_image(D, M) even if have_image(D, M) is already true. This makes "take_image" a not well-formed action and therefore Satellite is not well-formed.

