# OpenReview forum: "On the Ability of Transformers to Verify Plans"
_ICML.cc/2026/Conference — ICML 2026 regular_

### Official Review · Reviewer_62cv · 2026-03-11

**Soundness:** 3
**Presentation:** 3
**Significance:** 3
**Originality:** 2
**Overall Recommendation:** 4
**Confidence:** 2

**Summary:**

The paper analyzes the length generalization of decoder-only transformers on plan verification. Introduces C*-RASP, extending C-RASP to handle growing vocabularies (more objects at test time). Proves plan verification is in C*-RASP[Pos] for delete-free and well-formed domains, but not for general STRIPS or conditional effects (via reductions to FlipFlop/PARITY). Experiments confirm the theoretical predictions.

**Compliance With Llm Reviewing Policy:**

Affirmed.

**Key Questions For Authors:**

See above.

**Limitations:**

Section 5.3 discusses several limitations but does not discuss the gap between the idealized learning model and actual training, the difficulty of verifying well-formedness for novel domains, or the translation vs. permutation invariance issue.

**Strengths And Weaknesses:**

C*-RASP is a genuine contribution with implications beyond planning. Handling growing alphabets via the match predicate is non-trivial and addresses a real gap in length-generalization theory. The 4-way classification maps planning subclasses cleanly onto C*-RASP membership. The key insight that well-formed propositions toggle predictably, enabling counting-based verification, is elegant. Experiments are well-designed minimal pairs: Lights Out with conditional effects at chance level, well-formed variant near-perfect, despite the latter having exponentially more actions.

Several concerns. The idealized learning model assumes exact target matching and hard-constrained translation invariance. Actual experiments use standard GPT-2 with AdamW. The paper does not implement the offset trick for extended alphabets that the theory requires. This gap between the formal framework adn actual training should be discussed more explicitly.

The construction in Theorem C.20 uses and activation that is justified by the universal approximation theorem. But the UAT guarantees approximation by arbitrarily wide single-layer networks, not fixed-width transformer MLPs. Authors should clarify what precision guarantees hold for bounded-width architectures.

Well-formedness is defined semantically, quantifying over all states in valid plans. Checking this is PSPACE-hard. The claim that 70\% of IPC-style domains are well-formed is important but not substantiated with methodology. Authors should clarify how this was determined and whether efficient sufficient conditions exist?

Theorem 3.6 states membership in C*-RASP, but the constructions in Appendix B.1 use positional relations, which is C*-RASP[Pos] machinery. Authors should clarify whether the correct statement is C*-RASP[Pos] membership, as this matters for the APE vs NoPE conclusions.

The STRIPS negative result uses an instance with empty preconditions. While formally valid, most practical STRIPS domains have rich precondition structure. The negative result may overstate difficulty for realistic domains that are not well-formed but have other regularity.

The match predicate assumes object identifiers have numeric structure. Standard planning treats object names as arbitrary identifiers where the natural symmetry is permutation, not translation. It is unclear how broadly the translation-invariance assumption applies.

The experimental scope is limited to three domains. No standard IPC benchmarks tested. No ablation on the offset trick for extended alphabets.

---

> ### Author Rebuttal · Authors · 2026-03-28
>
> We thank the reviewer for recognizing that **C\*-RASP is a genuine contribution with implications beyond planning** & **Handling growing alphabets is non-trivial & addresses a real gap in length-generalizaton theory**. We also thank the reviewer for their kind words, calling our insights **elegant** & our minimal pair experiments **well designed**.
>
> We will try to address the concerns raised.
>
> > The idealized ... more explicitly.
>
> We agree our learning model is different from what people use empirically. Our motivation in using GPT-2 with AdamW was to test whether our theory is useful in predicting what is actually seen in more realistic settings. Our point here was that in actual training, length generalization (LG) happens when we predict it to, & doesn't when we predict it not to. We will make this more clear in the main paper.
>
> The offset trick has the implication of helping make full use of the match predicate & using the $\tau_k$ shifts in object embeddings (OE) to match not just $c_{i}$ with another $c_{i}$ in context, but also some $c_{i}$ & some $c_{i+\tau_k}$. If the predicate ends up not using $\tau_k$, then using offsets is equivalent to training OE randomly. For results on plan verification, this shift is not required. The mention of these offsets is intended to make our framework be accommodating for future work. We conjecture that having the ability to do such a shift can explain why index hints help in generalization (going from index $i$ to $i+\tau_k$ would help disambiguate where retrieval should happen). Here, the offset trick would bring about the intended invariance.
>
> > Theorem C.20  ... architectures.
>
> We would like to clarify that the transformers learned by the idealized inference procedure have width depending on the context window (accommodating test inputs, but not unboundedly longer inputs), because extended alphabet & larger positional embeddings likely require increasing width. Thus, in principle, they might represent this function up to the precision needed for the context window, given sufficient width as provided by the UAT. The same idea underlies the use of the Heaviside function in [2], which also can only be approximated at any fixed width. We will be more explicit about the inclusion of these functions in the main paper.
>
> > Well-formedness ...  exist?
>
> Indeed, in general it is hard to check whether a planning instance is well-formed (WF). There are cases, when the syntactic definition holds where it's easy, but this is restrictive. From a practical standpoint, a domain modeler, writing a PDDL model & a generator, would typically be able to tell. Thus, we asked a domain expert to manually go through all 10 domains in the IPC Learning track, and verify this property of the problem generator. We will include an appendix section where we document this expert's reasoning as applied to two of the domains (one WF, one not WF). Note that the hardness of ascertaining WFness does not invalidate our results.
>
> > membership in C*-RASP, ... conclusions.
>
> We thank the reviewer for catching our typo. While in Fig 1 and in our proofs we say that the +ve results hold for C*-RASP[Pos], in an unfortunate typo, we don't write it correctly in Theorem 3.6. We will correct this.
>
> > The STRIPS negative ... regularity.
>
> Practical domains do feature rich precondition structure. Our construction is an *intentional minimal example* to cleanly show what could lead to problems.
> When the preconditions are richer, the Flipflop signature (core cause) might get obscured. We do not claim that every single NWF STRIPS domain would not be in C-RASP[POS]; our claim is that, when such a FlipFlop signature exists, transformers won't generalize well (as shown through our Colors NWF Variant, which suffers from this).
>
> > The match  ..assumption applies.
>
> We would like to refer to our responses to reviewer TeCR & LSBR re choice of numeric structure & offsets.
> Translation invariance (TI) has seen to be beneficial for LG, as a transformers not showing TI can fail to generalize even on the induction head task [2] (for which there is enough evidence of success in trained transformers [3,4 inter-alia]). [2] also empirically showed that trained transformers often exhibit TI, both when trained from scratch on algorithmic problems, and in GPT-2. Thus, we think that TI should apply generally.
>
> > limited to three ... tested.
>
> We refer to our response to reviewer TeCR (due to lack of space).
>
> > Section 5.3... invariance issue
>
> We thank the reviewer for highlighting this, we will make this more clear.
>
> [1] Kazemnejad et al. "The impact of positional encoding on length generalization in transformers." NeurIPS 2023
>
> [2] Huang et al. "A Formal Framework for Understanding Length Generalization in Transformers",ICLR 2025.
>
> [3] Olsson et al. "In-context learning and induction heads." arXiv:2209.11895 (2022)
>
> [4] Elhage et al. "A mathematical framework for transformer circuits." Transformer Circuits Thread 1.1 (2021)

---

> > ### Author Rebuttal · Reviewer_62cv · 2026-04-01
> >
> > The authors confirmed the C*-RASP vs C*-RASP[Pos] typo in Theorem 3.6 and will correct it. The explanation that GPT-2 experiments test the predictive power of the theory, rather than implement it literally, is reasonable. The domain expert verification of well-formedness is adequate. I maintain my score.

---

> > > ### Author Response · Authors · 2026-04-02
> > >
> > > We are glad our responses addressed the reviewer’s concerns adequately and thank the reviewer once again for their feedback and catching our typo. We will indeed correct it, and make the other changes as promised.

---

### Official Review · Reviewer_LSBR · 2026-03-11

**Soundness:** 3
**Presentation:** 2
**Significance:** 3
**Originality:** 3
**Overall Recommendation:** 5
**Confidence:** 4

**Summary:**

The paper analyzes the expressivity of transformers by defining a range of tasks related to plan verification, and generalization in length and number of objects in the the planning instance.The main results are all captured in a table in Figure 1: basically, two large class of STRIPS problems have plans that can be verified by transformers via the equivalence to C and C*-RASP programs, while general STRIPS and STRIPS extended with conditional effects have plans that can't be verified by transformers. So there are positive and negative results, which build on the RASP family of languages and the properties of STRIPS plans and variations, shown elsewhere to correspond to a subset of star-free regular languages.

**Compliance With Llm Reviewing Policy:**

Affirmed.

**Final Justification:**

The detailed rebuttal addressed my concerns and I thus adjusted my (positive) recommendation upwards. Nice work.

**Key Questions For Authors:**

1. Why C-RASP[pos]? Why not other RASP fragments, like B-RASP? Where do the numbers get into the picture?
2. Simple argument for why not-well formed STRIPS is not in C-RASP[Pos]? I know that this is in the paper, but looks that this should be straightforward but I didn't fully get it from the text.

**Limitations:**

Yes

**Strengths And Weaknesses:**

Strenghts: Shed some light on the capabilities and limitations  of transformers in the setting of (classical) planning, addressing not just length generalization but "number of objects generalization"

Weakness: 1) Some of the result directly follow form earlier results on RASP and transformers, and on plans and formal languages. 2) The presentation could be improved -- it should be easier to grasp the results in the table (Figure 1), in particular, the counter-examples I think that too many things have been left unsaid in body of the paper. Too many points to be connected by the reader, in results that are not too subtle or deep, but definitely useful.

---

> ### Author Rebuttal · Authors · 2026-03-28
>
> We thank the reviewer for their postive evaluation of our work. We would like to address the concerns raised by the reviewer.
>
> > Some of the result directly follow form earlier results on RASP and transformers, and on plans and formal languages.
>
> We would like to gently disagree with the reviewer on this point. We do not believe that our results follow so easily from prior results on RASP and transformers. We would be very curious to know exactly which results follow from past work, as we would like to attribute the result to the correct prior work in case we had missed it.
>
> According to our knowledge, there has been extremely little work in connecting classical planning to transformer theory (specifically RASP), with the key past paper being by Núñez-Molina et al [1], that looked at the connection between B-RASP and Transformers (and we discuss the difference between their work and ours). In case we have missed some crucial related work, which anticipates our results, we would kindly request the reviewer to share it.
>
> > 2) The presentation could be improved ... definitely useful.
>
> We will strive to improve the presentation in the main paper. As suggested by reviewer XgB6 as well, we will change Figure 1 to be graphical in nature, and change Figure 3 to instead show some intuition about about our proof techniques for showing both positive and negative examples, and we hope that those figures will be able to provide sufficient clarity on the counter examples through such figures. We thank the reviewer again for pointing this out, and we promise to take serious steps to address this.
>
>
> > Why C-RASP[pos]? Why not other RASP fragments, like B-RASP? Where do the numbers get into the picture?
>
> The RASP variants address slightly different formal models of transformers and also are useful for provng results for Transformers of a very different kind. Specifically B-RASP is about transformers with Hard Attention (attention is 1 on one token, and 0 on all others). Additionally, B-RASP is only intended for providing *expressivity* results, i.e. whether such a formal model of Transformer can in principle perform a task, and the weights of the model could be hard-coded. Therefore, a positive result in B-RASP has limited scope and does not imply *learnability* or *generalization*, as that particular task might still not be learnable for Transformers with limited data (especially so in a length generalizable way). To date, C-RASP and C-RASP[Pos] are the **most successful** RASP frameworks for **understanding learnability** and **length generalization** observed in Transformers, with the added advantage that they give guarantees for standard softmax Transformes. While C-RASP[Pos] gives guarantees for Transformers with Absolute Positional Encodings, C-RASP does so for No Positional Encodings. The use of C-RASP[Pos] becomes essential as the arguments of actions may have a large arity, and thus distinguishing between argument i and argument i+1 would be difficult without the use of the [Pos] operations.
>
> We assume that the reviewer refers to the numbers used to refer to the alphabet in C*-RASP. In case, we have misunderstood the question, we ask the reviewer to kindly clarify which numbers they meant.
> The numbers are used in the alphabet in C*-RASP for two reasons.
> * To give an ordering to the extended alphabet to enable learning of the alpahbet through offsets and also to enable the match predicate to be more powerful.
> * All the RASP variants are essentially formal logic models, and hence do not operate at an embedding level, but on a logic level -- with boolean components and numerical components. This is why choosing numbering for the extended alphabet is a matter of convenience.
>
> > Simple argument for why not-well formed STRIPS is not in C-RASP[Pos]? I know that this is in the paper, but looks that this should be straightforward but I didn't fully get it from the text.
>
> Because of the less restrictive nature of Not well formed STRIPS, one can construct planning domains where the valid plans would look like the FlipFlop language (intuitively this can be thought of as the language of retrieving the last written bit from 1 bit cache memory.) Such a language is not in C-RASP[Pos], and provably not length generalizably learnable via Transformers (with extensive empirical evidence). Thus, for some domains in Not well formed STRIPS, such a FlipFlop signature appears in the language of valid plans, hence it's not in C-RASP[Pos].
>
>
> [1] Núñez-Molina, Carlos, Vicenç Gómez, and Hector Geffner. "From Next Token Prediction to (STRIPS) World Models--Preliminary Results." arXiv preprint arXiv:2509.13389 (2025).

---

> > ### Author Rebuttal · Reviewer_LSBR · 2026-04-01
> >
> > The rebuttal addressed my questions well. Brief follow up:
> >
> > -- there are papers in planning exploring the relation between STRIPS plans and star-free languages (On the Expressive Power of Planning Formalisms in Conjunction with LTL
> > Songtuan Lin, Pascal Bercher, 2022), while the relation between star-free languages and transformers is studied in the B-RASP paper. This is the couple of papers that I had in mind when I mentioned that some of the results in your paper, may follow from these two papers -- but I may be missing some relevant technical details (e.g. don't recall well-formed assumption there, but there may be a similar STRIPS restriction)
> >
> > -- About generalization: the paper by Nuñez Molina et al that you mention shows how positive and negative action traces in STRIPS, of any length, can be distinguished by a B-RASP program -- this result may be related to those in the paper too.
> >
> > -- Your well-formed assumption in STRIPS seems to refine the one in the recent SIFT paper by Gösgens et al., In that paper, the assumption is exploited by the resulting SIFT algorithm. Perhaps there is a relation between this model learning algorithm and the similar restriction needed to verify STRIPS, and something interesting can be said about this too.

---

> > > ### Author Response · Authors · 2026-04-02
> > >
> > > We are glad that our response addressed some of reviewer’s concerns, and we appreciate the opportunity to clarify further.
> > >
> > > > papers in planning ... STRIPS restriction.
> > >
> > > We agree that there are important connections, and we do acknowledge them explicitly in the paper. In particular, we cite Lin & Bercher (2022) in Lines 165–166, and Lines 735–737. We also discuss Nuñez Molina et al. at the end of Section 5.2. That said, we believe our results are not implied by these. The relation is limited to **one statement of one theorem**, viz. Statement 3 of Theorem 3.1, and even there our **conclusion is different in kind**.
> > >
> > > Specifically, Lin & Bercher show that STRIPS plan languages is a subset of star-free languages. We do use that fact, but only to emphasize Statement 3 of Theorem 3.1. Our contribution is to show that, even within this subset, there are STRIPS domains whose plan languages are not in C-RASP[Pos]! The FlipFlop language is the concrete separating example: it is star-free, hence falls within B-RASP, but since it is not in C-RASP[Pos], this has implications.
> > >
> > > > About generalization:  .... paper too.
> > >
> > > The same point also clarifies the relation to Nuñez Molina et al. Their result shows that +ve and -ve STRIPS traces of arbitrary length can be distinguished by a B-RASP program. As mentioned at the end of Section 5.2, this is indeed related, but there are clear differences. First, their results apply only in the fixed-vocab (grounded) setting, whereas we study generalization not only in sequence length but also under vocab growth. This distinction is central for learning in planning, where the aim is typically to learn from small grounded problems and generalize to larger ones with many more objects. Second, B-RASP characterizes what can be expressed by a transformer with hard/rightmost attention, whereas our analysis concerns what standard softmax-based transformers can length-generalize.
> > >
> > > This difference is not only formal, but also empirically meaningful. Our experiments support exactly this distinction: although B-RASP-style expressivity suggests such languages are representable in principle, standard transformers fail in the way as **predicted by C-RASP[Pos]**. Thus our statement about FlipFlop is not merely a theoretical separation, but one whose implications are borne out experimentally. While their result and ours are certainly related, ours is not implied by it: rather, it establishes a limitation that **remains invisible from the B-RASP perspective**.
> > >
> > > To summarize:
> > > 1. Lin & Bercher: STRIPS is a subset of star-free languages.
> > > 2. B-RASP is equivalent to star-freeness.
> > > 3. Nuñez Molina et al. use these ideas to argue that the entirety of grounded STRIPS is therefore in B-RASP.
> > > 4. We use the Lin&Bercher connection for Statement 3 of Theorem 3.1, and then show that even within this star-free/STRIPS regime there are languages (FlipFlop), that are in B-RASP but not in C-RASP[Pos].
> > > 5. This leads to a different conclusion: while such languages may be representable in principle, standard transformers should not be expected to length-generalize on them.
> > > 6. Our experiments supports that this distinction has practical consequences.
> > >
> > > Rest of our theory — including well-formedness (WF), conditional effects, delete-freeness, the fixed- vs. variable-universe, and most importantly about C\*RASP and the concerning theoretical guarantees — do not follow from Lin & Bercher or Nuñez Molina et al. We hope this clarifies both the connection and the difference. If the reviewer feels it would be helpful, we would be happy to expand this discussion in the final version so it's more clear for readers.
> > >
> > > > Your well-formed ...  this.
> > >
> > > Indeed, as mentioned in Section 5.1, our WF condition weakens the syntactic WF condition in the SIFT paper (the syntactic condition is also known in the literature as "one-safety" or "toggling" [1]).  We use the weaker definition as it allows us to show that transformers should generalise under both sequence length & vocab size growth for a much wider class of problems than could have been concluded if we had used the syntactic definition (e.g. far more in planning benchmarks).
> > > If we interpret your question as "can the weaker assumption also be used within SIFT", then the 2-SAT feature consistency procedure underlying SIFT should still be valid under the weaker assumption. However, SIFT will still output a domain that is WF in the stronger syntactic sense.
> > >
> > > If we interpret your question more broadly as being about the connections between the learnability of plan verification and that of domain models then yes this is a very interesting and we do not know!  For instance, we think it is unknown whether STRIPS domains without the WF restriction are learnable and how that relates to learning the non-deleted preconditions. But we are not experts here.
> > >
> > > [1] Sarah L. Hickmott et al:  Optimality Properties of Planning Via Petri Net Unfolding: A Formal Analysis. ICAPS 2009.

---

### Official Review · Reviewer_XgB6 · 2026-03-13

**Soundness:** 3
**Presentation:** 2
**Significance:** 3
**Originality:** 3
**Overall Recommendation:** 5
**Confidence:** 3

**Summary:**

This theory-driven paper analyzes the plan verification abilities of transformers, i.e., given a planning instance and a plan, verifying whether the plan correctly solves the instance. They categorize the planning instances by imposing incremental constraints: STRIPS (no conditional effects), well-formed (each effect changes state), and delete-free (positive effects only). They determine whether these problem classes belong to the C-RASP[Pos] (Counting Restricted Access Sequence Processing) language. They also consider generalization to larger object vocabularies, and develop a new variant of C-RASP for this purpose: C*-RASP. Similarly, the Limit Transformer concept from prior work is extended to Symbolic Limit Transformer for the same purpose.

**Compliance With Llm Reviewing Policy:**

Affirmed.

**Final Justification:**

The rebuttal addressed the questions I raised in my initial review. After that, our discussion with the authors mostly focused on the vocabulary extension aspect.

Both the rebuttal and other reviews are aligned with my initial judgement. Therefore, I will keep the same recommendation.

**Key Questions For Authors:**

1. How do you interpret the falling accuracy in well-formed Colors in Figure 5? In particular, I would appreciate more discussion regarding the different performance degradation rates between the theoretically generalizable problems.
2. The delete-free variant of Heavy Grippers exhibits a sharper drop in accuracy than the well-formed variant in Figure 5, despite the fact that delete-free is more restrictive. Why?

**Limitations:**

Yes. Section 5.3 discusses the limitations in sufficient detail.

**Strengths And Weaknesses:**

The paper presents an important and relevant theoretical analysis. The results have potential implications beyond just planning since the length generalization and vocabulary generalization are studied jointly, which also differentiates the paper from prior theoretical studies.

The paper presents the theorems concisely in the main text with proof sketches, deferring the actual proofs to the appendix. Most proofs are constructive in nature and as a result, rather verbose. I checked some of the proofs (primarily the proof of statement 1 in both Theorems A.1 and B.1), didn’t identify any errors, and found them more straightforward than expected. But I couldn’t check all of them in detail due to the sheer length and my limited familiarity with prior work (other proofs depend heavily on theoretical results from previous papers).

The main text is very dense. In hindsight, it’s actually harder to understand than some of the appendices. It’s clear that the authors put a lot of effort into the paper but its theoretical density makes it hard to present the results with both brevity and clarity. Some suggestions:
1. I don’t think Figure 3 is helpful in the main text. The heavy grippers domain was one of the easier parts to understand and Figure 2 was sufficient.
2. The main text could potentially benefit from figures that explain the intuition behind the proofs. I’m under the impression that the main idea behind proof of statement 1 in Thm. A.1 could be distilled into a figure.
3. Similarly, a more intuitive and graphical overview figure (to replace Figure 1) could improve comprehension for the broader ML audience.

As recognized by the authors, some of the simplifying assumptions (e.g., CoT and RoPE are not considered) may limit the practical implications. However, the paper is already heavy on the theory, and such considerations are best left for future work.

### Vocabulary Extension

To enable post-training vocabulary extension, the authors construct an extended embedding collection and use offsets to expose the model to all tokens during training (Appendix C.1). This operation corresponds to alpha-renaming, and implementing extensible vocabularies in this way has been previously studied in the literature [1] under the name “alpha-renaming baseline”.
1. Consider citing [1] to relate your contribution to vocabulary extension methods. Since vocabulary generalization is a central focus in your paper, [1] is highly relevant.
2. The proposed formulation never modifies the order of extended tokens (which are called interchangeable tokens in [1]) due to dependence on offset. For example, $c_0$ and $c_1$ may transform to $c_1$ and $c_2$, but never $c_1$ and $c_0$, even though the latter transformation would also retain the same semantic meaning within the planning instance. Consider using the complete definition of alpha-renaming (a mapping from $[1, \ldots, N(T)]$ to a random permutation) for theoretical rigor. I’m also curious about how this offset-based alpha-renaming affects the empirical results compared to a fully random alpha-renaming, but I understand that this is somewhat out-of-scope.

### Minor issues
1. Comma missing after $a \in A$ in line 135, left column.
2. Period missing at the end of the sentence in line 380, left column.

## References
* [1] Işık, İlker, et al. “Interchangeable Token Embeddings for Extendable Vocabulary and Alpha-Equivalence.” 2025, Forty-second International Conference on Machine Learning.

---

> ### Author Rebuttal · Authors · 2026-03-28
>
> We thank the reviewer for their positive evaluation, terming it **important and relevant theoretical analysis**. We are also happy that the proofs that the reviewer checked were straightforward to follow and the reviewer **didn’t identify any errors**. We also thank them for  their kind words viz. **It’s clear that the authors put a lot of effort into the paper**.
>
> We would like to now address the reviewer's concerns .
>
> > Main text is dense. ...
> >> I don’t think Figure 3 is ........ distilled into a figure.
> >> More intuitive ...  ML audience.
>
> We acknowledge that the main text is dense as our paper addresses formal planning and combines it with frameworks for length generalization in Transformers. As it combines fields which not been well studied in tandem, the text gets dense. We are grateful for the feedback on improving readability. Our motivation behind Figure 3 was to show what a valid plan would look like, as that might not be obvious to many who are not well versed with planning but are with Transformer theory. In case the reviewer feels that the previous section already communicates this well, we will move the figure to the appendix. We will strive to illustrate our proof idea and our broader results through images, and graphical figures. We agree that it would be help in enhancing readability.
>
> > Consider citing [1] ...  that this is somewhat out-of-scope.
>
> We thank the reviewer for sharing this paper. We would like to clarify that there is a very clear motivation behind not giving random permutations and giving explicit ordering to characters. While our current results for plan verification do not require it, such an ordering is key to having a very general definition of the match predicate. This predicate allows matching not just 2 new symbols in context, but also if 1 of the symbols is a small shift away from the other in the embedding subspace. We hypothesize that using our framework, it could be possible to, for instance explain why index hints help in generalization as seen in prior work [1]. One would be able to shift from 1 index to the other (due to the consistent local ordering between characters, which would be lost in a random permutation) and not lose track of the positions with index hints providing local context for disambiguation. Our offset trick to enable generalization over a larger vocabulary by only having access to a shorter one is also based on the fact that the ordering exists. The paper that the reviewer has shared has an orthogonal idea for improving the vocabulary generalization and could be potentially used in conjunction with ours. We will cite the paper and add this point as a discussion in our paper to clarify the need for the ordering of the extended character set.
>
> > Minor Issues : Comma missing ...
>
> We thank the reviewer for catching these typos, we will fix them!
>
> > Q. falling accuracy in well-formed ... generalizable problems.
>
> As the reviewer noted, our theoretical guarantees are given via an idealized learning setup. When experiments are done in actual setups, we tend to see slight degradations at lengths that are close to the maximum lengths for which we train our positional encodings with offset. In most cases, this can be attributed to the fact that the absolute longest positional encodings are sometimes left undertrained as  the no. of data samples for which such positions are achieved via our random seeds are low.
> While we can hypothesize such post hoc explanations, a solid theoretical understanding of this doesn't exist. However, the key thing that our framework can shed light on is the fact that **there will exist a clear difference in the performance curves** between algorithms that are friendly for the Transformer architecture (like the operations in C & C*-RASP) and those that are not. Such differences get best highlighted in the presence of minimal pairs (like we see in our Colors Domain). Additionally, the presence of such a difference between the curves is predictable and explainable with our frameworks.
>
> > The delete-free variant...  restrictive. Why?
>
> We would like to clarify here that Delete-free is not a subset of Well-formed. Consider a planning instance with initial state {p,q}, goal {r}, and a single action with precondition {q} and effect {p,r}. This is delete-free (no negative effect) but not well-formed as the action is applicable in the initial state where one of its effect already holds.
> The drop in accuracy here again is only towards the very end (after generalization of x1.9 or more), and we believe that the reason for that is once again similar to what we see in the case of Colors. Training more and more positional encodings would lead to generalization upto to a higher point.
>
> [1] Zhou, Hattie, et al. "What Algorithms can Transformers Learn? A Study in Length Generalization." ICLR'24

---

> > ### Author Rebuttal · Reviewer_XgB6 · 2026-04-03
> >
> > Thank you for the rebuttal. I appreciate the additional discussion on the experimental results. I have a few follow-up questions:
> > 1. I understand that character indices are semantically significant for a general definition of match predicate. But is character ordering required for the domains considered in the experiments? In my current understanding, the index hints in [1] serve a very different function than the symbolic characters used in, e.g., Fig. 2 (Heavy Grippers), where the indices of characters (not in terms of position, but index within the alphabet) don't seem meaningful.
> > 2. Does the paper depend on the character ordering assumption for any theoretical claim? In other words, if we relaxed the character ordering assumption, how would this affect the theoretical results?
> >
> > [1] Zhou, Hattie, et al. "What Algorithms can Transformers Learn? A Study in Length Generalization." ICLR'24

---

> > > ### Author Response · Authors · 2026-04-04
> > >
> > > We are happy to have resolved some of the concerns of the reviewer, and are happy to answer further. The answers to both questions are slightly related, so we will start by answering the second question first.
> > >
> > > > Does the paper depend on the character ordering assumption for any theoretical claim? In other words, if we relaxed the character ordering assumption, how would this affect the theoretical results?
> > >
> > > If we remove the character ordering assumption, none of the theorems about planning change at all. The only effect on the theory is that C*-RASP, as a formalism, becomes slightly less expressive than it is when such an ordering is included.
> > >
> > > The translation from C*-RASP to Symbolic Limit Transformers is one-way rather than an equivalence (the same asymmetry exists between C-RASP and Limit Transformers). Any C*-RASP program can be translated into a Symbolic Limit Transformer, which is enough for us to establish positive length-generalization guarantees for any task expressible in C*-RASP. However, the converse does not hold. That is, even if a task cannot be shown to be expressible in C*-RASP, it may still be expressible by a Symbolic Limit Transformer and therefore still admit length generalization.
> > >
> > > If character ordering is removed, C*-RASP simply becomes weaker. The planning results remain unchanged, but our framework would be able to prove fewer positive results in the future, and the gap between Symbolic Limit Transformers and our more human-interpretable and practically usable formalism would become slightly larger.
> > >
> > > Put differently, without character ordering one can more easily construct tasks that are length-generalizable and expressible by Symbolic Limit Transformers, yet not expressible in C*-RASP. Index-hint-type constructions are a natural example: they require nonlocal jumps in indices, which fall outside what C*-RASP can express without such ordering.
> > > For C-RASP, so far,  the community has not found a task that provably cannot have a program in this formalism and still length generalizes (implicitly proving that this task then was expressible in Limit Transformers). Without character ordering, one would be able to construct such a task quite easily (index hints) and show that it is not in C*-RASP (because of the jumps required in indices) and yet length generalizable (as seen in prior work).
> > >
> > > > I understand that character indices are semantically significant for a general definition of match predicate. But is character ordering required for the domains considered in the experiments? In my current understanding, the index hints in [1] serve a very different function than the symbolic characters used in, e.g., Fig. 2 (Heavy Grippers), where the indices of characters (not in terms of position, but index within the alphabet) don't seem meaningful.
> > >
> > > Exactly, the reviewer's intuition is exactly right. The domains we consider both theoretically and experimentally (including Heavy Grippers) do not require any character ordering of any sort.  There is an ordering over arguments imposed by the structure of the domain, but this can be handled using local positional relations, without any need for character-level ordering relations. This makes our setting quite different from the index-hint setting in [1], and therefore would be not affected by the removal of the character ordering assumption at all.
> > >
> > > We hope this clarifies the point, and we would be happy to elaborate further if useful.

---

### Official Review · Reviewer_TEcr · 2026-03-13

**Soundness:** 3
**Presentation:** 3
**Significance:** 3
**Originality:** 4
**Overall Recommendation:** 4
**Confidence:** 4

**Summary:**

This paper analyzes the ability of decoder-only transformers to verify plans: determine if a candidate plan meets the requirements of an instance and produces a valid solution to the desired goal. The primary contribution is a revised version of the C-RASP approach, called C-RASP*, which extends C-RASP to handle growing vocabulary size during the testing phase as the number of objects increases. The central theoretical discovery is that there is a clean separation between transformers that can provably perform length-generalization (in delete-free domains) and those that cannot do so for more general STRIPS or conditional-effect verification domains. Results from the Lights Out World, Colors, and Heavy Grippers experiments, like the theoretical results, show that domain types with well-formed/delete-free properties generalize very well, while domains with STRIP or conditional-effect characteristics generally do not.

**Compliance With Llm Reviewing Policy:**

Affirmed.

**Final Justification:**

Based on initial comments and rebuttal.

**Key Questions For Authors:**

- To what extent does the idealized inference model contribute to the positive guarantees compared to serving primarily as a proof device?
- Can you separate cursing growth from sequence-length growth in the variable-universe experiment with an ablation?
- Do the key findings hold across an expanded set of traditional planning benchmarks rather than simply within custom synthetic domains?

**Limitations:**

Yes. The paper addresses limitations related to its emphasis on verification instead of generation and the extent of its theoretical framework. In my opinion, the author properly addresses the limitations of the paper.

**Strengths And Weaknesses:**

Strengths
- The C*-RASP model extends previous research on length-generalization theories in terms of variable universes.
- The conditions for plan verification learned by transformers are interpretable.
- The experimental data backs up theoretical assumptions.

Weaknesses
- Learning is still relatively abstract compared to standard SGD trainers, as the training model is abstract.
- Unfortunately, the experimental study is limited in that it would not lend itself to replicating studies done with more general data sets than those utilized.
- Finally, the experiments conducted in the variable universe are not conducted in controlled circumstances, so establishing the influence of new objects is difficult due to the relationship between length and vocabulary growth.

---

> ### Author Rebuttal · Authors · 2026-03-28
>
> We thank the reviewer for their positive evaluation. We would like to address the reviewers' concerns.
>
> > *W*: Learning is still relatively abstract ... abstract.
> > *Q*: To what extent does the idealized .... proof device?
>
> We agree that the learning model we use & standard SGD trainers are different in their formulation. However guarantees considering actual training dynamics have been challenging, restricted to 1/2 layers & on toy tasks [2,3,inter-alia]. We use the learning model of [1] for all of our C-RASP results, as it has so far been empirically predictive across scale [4,5]. In fact we reduce restrictions on the regularizer compared to [1] for C*-RASP. Additionally, as the reviewer noted, the experiments do track the theory. We believe that, till the time optimization theory advances to deeper transformers, such idealized learning models can be valuable in building intuition by proving both -ve and +ve results.
>
> > *W*: the experimental study is limited .... utilized.
> > *Q*: Do key findings .... synthetic domains?
>
>
> We believe that our theory should apply to the IPC planning domains. There are many datasets for planning but very few for plan verification (PV). PlanBench and ACPBench [6,7] have been used for testing PV using SOTA LLMs but have too few instances for training.
> Existing classical planners can be used to produce examples of good quality valid plans for a given instance, or a small set of plans that are "diverse" wrt a specific diversity metric (e.g. top K). Such planners have been used to train transformers for plan generation [e.g. 8]. However for PV we need both valid and invalid plans that are diverse in a much broader sense. Our data generation achieves this without requiring specific diversity metrics.
> Creating apt datasets with enough +ve and -ve plans for each IPC domain such that plan instances don't have patterns that allow shortcuts is an open problem. We would be eager to see if in future work someone evaluates our predictions on a wider range of IPC domains on a bigger scale.
>
> > *W*: experiments conducted .... vocabulary growth.
> > *Q* separate cursing growth ... ablation?
>
> We study a fixed universe, (fixed objects, longer plans) and a variable universe (more objects and longer plan). This is why -- we tried to show empirical agreement of our +ve claims in the most general case. Our motivation in studying these -- both theoretically and empirically -- was due to the fact that in many cases, more objects tend to lead to longer plans. We agree with the reviewer that studying just an increase in the no. of objects, with fixed length would also be interesting. As per our current framework, if the plan length does not change, the increased no. of objects gets limited by the total sequence length during training, and thus any plan in such a case would be considered in-distribution.
>
> We do however run ablations as per the request of the reviewer for Colors and Grippers, in this setting. Lights Out experiments were already done in a fixed universe setting and so it is not included in the ablation. We see that indeed the model generalizes to just more objects (same length as training) & to longer plans (same number of objects).
>
> * Colors
>
>     * Length=100, more objects
>
>     No. Objects | WF | NWF
>     ---- | ---- | ---- |
>     12| 1.0| 0.8389|
>     13| 0.9998| 0.8021|
>     14| 0.9998| 0.7174|
>     15| 0.9954| 0.6663|
>
>     * Objects=15, longer plans
>
>     Length | WF | NWF
>     ------ | ----- | ------
>     141-160| 0.9991| 0.6714
>     161-180| 0.9977| 0.6699
>     181-200| 0.9934| 0.6351
>
> * Grippers
>
>     * Length=100, more objects
>
>     No. Objects | WF | DF
>     ----- | ------ | ------
>     81-100| 1.0 | 0.99
>     101-120| 1.0| 0.98
>     120-140| 1.0| 0.96
>
>     * Objects=140, longer plans
>
>     | Length | WF | DF
>     | ------ | ----| -----
>     | 140-159| 1.0| 1.0
>     | 160-179| 1.0| 0.9832
>     | 180-199| 0.9812| 0.9012
>
>
> 1. Huang et al. "A Formal Framework for Understanding Length Generalization in Transformers",ICLR 2025.
>
> 2. Gopalani et al. Global convergence of SGD on two layer neural nets. Information and Inference: A Journal of the IMA, 14(1), 2025.
>
> 3. Ren et al. Emergence and scaling laws in SGD learning of shallow neural networks, NeurIPS 2025.
>
> 4. Yang et al. "Knee-Deep in C-RASP: A Transformer Depth Hierarchy, NeurIPS 2025.
>
> 5. Jobanputra et al. Born a Transformer--Always a Transformer? On the Effect of Pretraining on Architectural Abilities, NeurIPS 2025
>
> 6. Valmeekam et al. "PlanBench: An Extensible Benchmark for Evaluating Large Language Models on Planning and Reasoning about Change." NeurIPS 2023
>
> 7. Kokel et al. "ACPBench: Reasoning About Action, Change, and Planning", AAAI 2025
>
> 8. Rossetti et al. "Learning General Policies for Planning through GPT Models.", ICAPS 2024

---

> > ### Author Rebuttal · Reviewer_TEcr · 2026-04-03
> >
> > Thank you for the detailed rebuttal. I maintain my score.

---

### Decision · Program_Chairs · 2026-04-30

**Decision:**

Accept (regular)

**Comment:**

This paper formally extends the C-RASP framework to analyze transformer length and vocabulary generalization in plan verification. It theoretically proves and empirically validates that transformers can generalize on well-formed/delete-free domains, but struggle with general STRIPS or conditional effects.

Reviewers praised the elegant theoretical analysis and the C*-RASP framework as a non-trivial contribution extending beyond planning (Reviewer XgB6, Reviewer 62cv). The theoretical boundaries established between domains are well-supported by carefully designed minimal pair experiments (Reviewer TEcr, Reviewer 62cv). Initial concerns regarding the dense presentation (Reviewer XgB6, Reviewer LSBR), the gap between theoretical models and standard training (Reviewer TEcr, Reviewer 62cv), and the need to isolate vocabulary growth (Reviewer TEcr) were successfully resolved. The authors addressed these by providing new ablations isolating length and vocabulary scaling (Rebuttal to TEcr), clarifying distinctions from prior star-free language research (Rebuttal to LSBR), and justifying their character ordering assumptions (Rebuttals to XgB6 and 62cv).

Remaining limitations include the reliance on an idealized learning model rather than standard training dynamics, evaluation restricted to custom synthetic domains, the omission of modern architectures like RoPE and Chain-of-Thought, a strict focus on plan verification over generation, and the computational hardness of verifying domain "well-formedness."

All reviewers gave positive scores. Hence, I recommend Accept.